# Rethinking GCNs for the Traveling Salesman Problem: Are We Encoding Effectively?

## Abstract

Graph Convolutional Networks (GCNs) have demonstrated strong potential to address the Traveling Salesman Problem (TSP). However, existing GCN-based TSP solvers still struggle with limited generalization, overfitting, and extending to asymmetric TSPs. To address these challenges, we rethink how to enable models to learn unified and generalizable representations for TSPs. Specifically, we introduce three encoding strategies: global node embedding for **Input Unification**, Min-Max Scaling for universal **Edge Normalization**, and layer-wise expanding views for **Aggregation Enhancement**. These designs culminate in **UNE-GCN**, a model that learns generalizable TSP representations with strong robustness, favorable learning dynamics, and linear scalability. Extensive experiments show that UNE-GCN can guide LKH-3 to perform more efficient searches, and the two-stage framework of UNE-GCN + LKH-3 achieves superior solutions with less search time on both symmetric and asymmetric TSPs. To demonstrate the effectiveness of our encoding strategies, we apply experimental comparison and discuss different encoding schemes to unveil and validate the critical roles of **U-N-E** in the advancement of GCN-based TSP solvers. Experimental results demonstrate that UNE-GCN achieves state-of-the-art performance, with up to a **0.60%** improvement in Gap over plain LKH-3 on large-scale ATSPs and an **83%** reduction in error metric compared to the original GCN backbone on large-scale STSPs, providing insights for the design of more effective graph encoders.

## 1 Introduction

The Traveling Salesman Problem (TSP), as a classical combinatorial optimization and routing problem, has been extensively studied. Among advanced classical algorithms, there are the exact solver Concorde (Applegate et al., 2006) and the heuristic solver Lin-Kernighan-Helsgaun (LKH) (Helsgaun, 2000). While these classical solvers have demonstrated strong performance, they rely heavily on hand-crafted heuristics. Recently, numerous neural TSP solvers based on graph neural networks (GNNs) have emerged and achieved promising results. Among various neural solving paradigms, the two-stage solving framework has been widely adopted. Specifically, it first employs graph convolutional networks (GCNs) to generate heatmaps, and then guides post-search algorithms based on candidate sets indicated by the heatmaps. An edge-aware GCN (Bresson & Laurent, 2018) is widely adopted to encode the structural properties of TSP. Commonly used post-search algorithms include Monte Carlo Tree Search (Fu et al., 2021), LKH-3 (Helsgaun, 2017), and RBS (Huang et al., 2025), which can leverage heatmaps to perform searches.

Despite the proliferation of GNN-based TSP solvers, a critical question remains insufficiently addressed: *Are we encoding TSPs effectively?* There is a growing consensus that neural TSP solvers should prioritize generalization—the capacity to generalize effectively from small-scale to large-scale TSPs is essential for practical utility and cost-effectiveness. However, we observe that existing graph encoders lack effective designs for generalization. Instead, they introduce features detrimental to generalization, causing GCNs to overfit these noisy signals and hindering the learning of more universal representations. Meanwhile, the flexibility of encoding also remains to be improved, as existing GCNs struggle to solve symmetric and asymmetric TSPs in a unified manner. We argue that such improvements of encoding should be fundamentally rooted in the graph input and encoding layers. Existing approaches, such as employing Diffusion models as decoders (Sun & Yang, 2023), are not intrinsically designed for graph encoding and further introduce additional costs during both

training and inference. Furthermore, previous graph encoders do not exhibit favorable scaling-up behavior; instead, they tend to amplify overfitting and degrade generalization when scaling up model parameters.

To address these challenges, we propose UNE-GCN, a unified and generalizable framework that integrates three encoding strategies to enhance graph representation learning for the Traveling Salesman Problem (TSP). Our main contributions are summarized as follows:

- We explore three critical aspects of graph encoding: **Input Unification**, **Edge Normalization**, and **Aggregation Enhancement**. Specifically, we replace node coordinate embeddings with a global node embedding, apply Min-Max Scaling to normalize the sparse distance matrix, and introduce layer-wise expanding views for multi-view learning. We have demonstrated the effectiveness and generalization of **U-N-E** in solving general TSPs.

- We design **UNE-GCN**, a unified and effective framework that combines **U-N-E** to achieve strong generalization across both symmetric and asymmetric TSPs. It exhibits robust learning dynamics, including resistance to overfitting, stable convergence, and favorable scaling behavior.

- Extensive experiments demonstrate that UNE-GCN achieves state-of-the-art performance across a wide range of TSP benchmarks. Notably, UNE-GCN trained on small-scale instances can effectively guide LKH-3 to solve large-scale problems, particularly in the asymmetric setting, contributing new insights into the scaling behavior of GNN-based solvers.

## 2 RELATED WORK

**Typical GNN Backbones for TSPs**  The commonly used backbones of GNN-based TSP solvers include Graph Convolutional Networks (GCNs), and Graph Attention Networks (GATs). Moreover, GCNs (Joshi et al., 2019; Fu et al., 2021; Xin et al., 2021; Sun & Yang, 2023; Li et al., 2024; Huang et al., 2025; Pan et al., 2025) adopt a non-autoregressive approach to generate indicative heatmaps, while GATs (Kwon et al., 2021; Pan et al., 2025) employ an autoregressive approach to sequentially predict the next node. Built upon a GCN backbone, our framework focuses on enhancing input encoding to improve generalization and representational capacity.

**Generalization for Symmetric TSPs**  With respect to generalization of GCNs for symmetric TSPs, Joshi et al. (2021) and Lischka et al. (2024) demonstrated that graph sparsification by $k$-Nearest Neighbors ($k$-NN) can improve generalization to some extent. Moreover, Fu et al. (2021) proposed decomposing large-scale problems into smaller sub-problems that the model is more adept at solving, and introduced Monte Carlo Tree Search (MCTS) to merge these sub-problems. In addition, Sun & Yang (2023); Li et al. (2024) proposed employing diffusion models as decoders and generating multiple heatmaps with different random seeds, which improved the diversity of solutions. Recently, related studies (Fang et al., 2024; Huang et al., 2025) have shown that normalization of scale-dependent features can significantly improve generalization. Specifically, they introduced a coordinate-based edge normalization, which eliminates cross-scale differences in edge lengths. Building upon these insights, we unify the ideas of graph sparsification and edge normalization, and propose more general strategies that facilitate better generalization to symmetric TSP instances.

**Unified Modeling for General TSPs**  To extend to asymmetric TSPs, Kwon et al. (2021) proposed a matrix encoding network (MatNet) based on GAT, which introduces dual node representations consisting of zero vector and one-hot encoding. To address the scalability issue of one-hot encoding, Pan et al. (2025) proposed an extendable pseudo one-hot encoding. Furthermore, Pan et al. (2025) proposed pseudo-coordinates and a dual GCN to accommodate the coordinate-free and asymmetric nature. Despite these efforts, existing approaches still face limitations in terms of generalization. Specifically, coordinate-based edge normalization is inapplicable to asymmetric TSPs, and current node embedding strategies fail to offer a unified representation for both symmetric and asymmetric TSPs. Furthermore, the learning dynamics of these neural encoders have not been thoroughly investigated, and the overfitting issue has not been thoroughly explored or addressed. To this end, we aim to develop a unified and generalizable encoding framework capable of handling both symmetric and asymmetric TSPs. Our work focuses on designing more effective graph encoders and conducting an in-depth investigation into the learning dynamics of different encoding strategies.

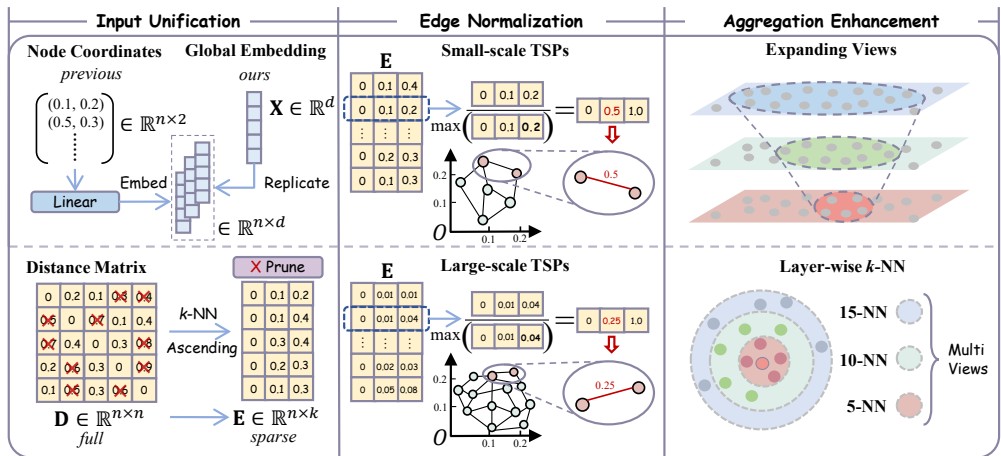

Figure 1: An overview of our **UNE-GCN** framework. **UNE-GCN** improves TSP representation learning via three modules: (1) **Input Unification** with global node embedding and $k$-NN selection; (2) **Edge Normalization** via Min-Max Scaling without reliance on coordinates; (3) **Aggregation Enhancement** with layer-wise expanding views. These components address challenges of generalization, overfitting, flexibility, and scalability in GCN-based TSP solvers, enabling efficient solution search when paired with solvers like LKH-3.

## 3 PRELIMINARIES

For convenience, we denote a two-dimensional symmetric/asymmetric TSP instance with $n$ nodes as STSP/ATSP-$n$. For an instance with $n$ nodes, let the node set be $\mathcal{X} = \{x_i\}_{i=1}^n$ and the distance (edge-length) matrix be $\mathbf{D} \in \mathbb{R}^{n \times n}$, where $\mathbf{D}_{i,j}$ denotes the non-negative distance from $x_i$ to $x_j$. Note that in STSPs, nodes can be described by explicit two-dimensional coordinates, and pairwise distances can be computed directly from these coordinates. In contrast, ATSPs do not necessarily satisfy these conditions; their node sets are used solely to represent the graph structure (*i.e.*, the connectivity and edge direction). Given the above conditions, our goal is to find a permutation $\Pi = [\pi_i]_{i=1}^n$ that minimizes the length $L$ of the TSP tour, which is a Hamiltonian cycle. Note that for ATSPs, the order in $\Pi$ cannot be reversed, *i.e.*, the tour is directed. $L$ is computed as follows:

$$L(\Pi, \mathbf{D}) = \sum_{i=1}^{n-1} (\mathbf{D}_{\pi_i, \pi_{i+1}}) + \mathbf{D}_{\pi_n, \pi_1}. \tag{1}$$

## 4 METHODOLOGY

### 4.1 INPUT UNIFICATION

Let $n$ and $d$ denote the number of nodes and the hidden dimension of the model, respectively. A straightforward approach is to incorporate both node coordinates and edge lengths for embedding. However, coordinates, as scale-dependent features, inevitably introduce biases detrimental to generalization when used for embedding. In addition, ATSPs lack node coordinate attributes. Motivated by the aforementioned issues, we introduce a global embedding of nodes for both STSPs and AT-SPs. As illustrated in the upper panel of *Input Unification* in Figure 1, we first initialize a global learnable tensor $\mathbf{X} \in \mathbb{R}^d$ using Xavier initialization (Glorot & Bengio, 2010), and then replicate it adaptively to $\mathbb{R}^{n \times d}$ according to $n$ of the input instance. In this way, node embeddings across arbitrary scales share a unified representation, without introducing additional biases. Consequently, such coordinate-free inputs significantly improve generalization for general TSPs.

In prior work, the graph sparsification technique based on $k$-NN selection (Joshi et al., 2021; Lischka et al., 2024; Fang et al., 2024; Huang et al., 2025) has been shown to improve generalization. As illustrated in the lower panel of *Input Unification* in Figure 1, we directly prune the full distance matrix $\mathbf{D}$ by $k$-NN selection and obtain the sparse one $\mathbf{E} \in \mathbb{R}^{n \times k}$, where $k$ denotes the fixed degree of each node. The elements in $\mathbf{E}_i$ are arranged in ascending order, where $\mathbf{E}_{i,j}$ denotes the $j$-th shortest edge starting from node $x_i$. Note that $\mathbf{E}_{i,1}$ is always 0, corresponding to the distance from $x_i$ to itself. The sparse distance matrix $\mathbf{E}$ not only stabilizes message aggregation, but also ensures linear scalability in computational complexity across different scales. The unified input prevents

the introduction of biases and irrelevant features, allowing UNE-GCN to focus on learning scale-invariant and coordinate-free representations for TSPs.

## 4.2 EDGE NORMALIZATION

As the most significant scale-dependent feature, edge lengths greatly affect the generalization of GCNs. However, a universal edge normalization applicable to general TSPs has long been lacking. As the number of nodes increases, the denser distribution results in a decreasing magnitude of entries in the sparse distance matrix $\mathbf{E}$. INViT (Fang et al., 2024) and RsGCN (Huang et al., 2025) introduce a two-dimensional uniform scaling for edge lengths based on specific coordinates, this coordinate-based edge normalization effectively aligns the value magnitude of $\mathbf{E}$ across STSPs of different scales. However, ATSPs cannot be naturally represented using coordinates, as their edge lengths lack well-defined proportionality in two-dimensional space. Consequently, RsGCN lacks the ability to normalize edge lengths in ATSPs. Further details are provided in Appendix G.

Accordingly, we employ Min-Max Scaling for edge normalization, ensuring that the approach generalizes to both STSPs and ATSPs. As illustrated in *Edge Normalization* of Figure 1, $\mathbf{E}_i$ represents the edge vector for the subgraph from the perspective of $x_i$. For each such subgraph, we perform independent edge normalization according to the following formula:

$$\tilde{\mathbf{E}}_i = \frac{\mathbf{E}_i - \min(\mathbf{E}_i)}{\max(\mathbf{E}_i) - \min(\mathbf{E}_i)} \xrightarrow{\min(\mathbf{E}_i)=0} \frac{\mathbf{E}_i}{\max(\mathbf{E}_i)}, \ \forall i \in \{1, 2, \dots, n\}, \tag{2}$$

where $\mathbf{E}_i$ contains the distance from $x_i$ to itself, ensuring that $\min(\mathbf{E}_i) = 0$. Min-Max Scaling offers a simpler and more efficient computation and preserves the relative relationships among values. We reduce the two-dimensional relational scaling to a simpler linear scaling, thereby eliminating the reliance on node coordinates.

## 4.3 AGGREGATION ENHANCEMENT

Given an optimal tour $\hat{\Pi}$, heuristic observation suggests that the successor of any node in $\hat{\Pi}$ is more likely to be among its nearest neighbors. Furthermore, a layer-wise edge dropout technique (Rong et al., 2020) has been shown to alleviate over-smoothing and improve training stability for GCNs; INViT (Fang et al., 2024) demonstrates that multi-view encoding can improve the generalization of Transformer for solving STSPs. Motivated by the nearest-neighbor preference of TSPs and above feature enhancement techniques, we adopt layer-wise expanding views to enhance graph convolutions for more effective and robust capture of multi-view features of TSPs.

Specifically, for UNE-GCN with $l$ layers, we define a vector of integers $\mathbf{F} \in \mathbb{Z}^l$ that specifies the view sizes, where $\mathbf{F}_i$ denotes that the $i$-th layer aggregates information only from its $\mathbf{F}_i$-nearest neighbors. We design the constraint that $\mathbf{F}_i \leq \mathbf{F}_j \leq k$ whenever $i < j$. As illustrated in *Aggregation Enhancement* of Figure 1, it presents three-layer expanding views with $\mathbf{F} = [5, 10, 15]$. Under this design, the lower-layer graph convolutions are able to focus on nearer and more promising neighbors, while the higher-layer convolutions with broader views can concentrate on integrating the global information of the entire graph. Overall, the multi-view aggregation prevents UNE-GCN from being restricted to a single-scale TSP pattern, improving generalization on unseen-scale TSPs.

## 4.4 UNE-GCN ARCHITECTURE

For convenience, we define $x_i^j$ as the $j$-th nearest neighbor of node $x_i$, with $x_i^1 = x_i$. Consequently, $\mathbf{E}_{i,j}$ and $\tilde{\mathbf{E}}_{i,j}$ respectively denote the original and normalized length of the edge $x_i \to x_i^j$. Based on the above conditions, we specifically designed the architecture of UNE-GCN as follows.

**Input Embedding** The node embeddings $x^0 \in \mathbb{R}^{n \times d}$ are obtained by replicating the global tensor $\mathbf{X} \in \mathbb{R}^d$ as Equation 3, while the edge embeddings $e^0 \in \mathbb{R}^{n \times k \times d}$ are derived via a learnable linear projection of $\tilde{\mathbf{E}} \in \mathbb{R}^{n \times k}$ as Equation 4. Further node embeddings are presented in Appendix E.

$$x_i^0 = \mathbf{X}, \tag{3}$$

$$e_{i,j}^0 = \text{Linear}(\tilde{\mathbf{E}}_{i,j}) : \mathbb{R}^1 \to \mathbb{R}^d. \tag{4}$$

**Graph Encoder**   Note that throughout the following, $e_{i,j}$ consistently denotes the hidden representation of the $j$-th shortest edge starting from node $\mathrm{x}_i$, *i.e.*, $\mathrm{x}_i \to \mathrm{x}_i^j$. With a predefined view size vector $\mathbf{F} \in \mathbb{R}^l$, the graph convolution in layer $\ell \in \{1, 2, \dots, l\}$ is formulated as:

$$u_{i,j}^\ell = \mathbf{W}_1^\ell e_{i,j}^{\ell-1} + \begin{cases} \mathbf{W}_2^\ell x_i^{\ell-1} + \mathbf{W}_3^\ell x_j^{\ell-1}, & \text{if } j \leq \mathbf{F}_\ell, \\ \mathbf{0}_d, & \text{otherwise}, \end{cases} \tag{5}$$

$$v_{i,j}^\ell = \mathbf{W}_4^\ell x_i^{\ell-1} + \begin{cases} \mathbf{W}_5^\ell x_j^{\ell-1} \odot \sigma(u_{i,j}^\ell), & \text{if } j \leq \mathbf{F}_\ell, \\ \mathbf{0}_d, & \text{otherwise}, \end{cases} \tag{6}$$

$$e_{i,j}^\ell = e_{i,j}^{\ell-1} + \text{GELU}(\text{LN}(u_{i,j}^\ell)), \tag{7}$$

$$x_i^\ell = x_i^{\ell-1} + \text{GELU}(\text{LN}(\sum_{j=1}^k v_{i,j}^\ell)), \tag{8}$$

where $\mathbf{W}_{1\sim5} \in \mathbb{R}^{d \times d}$ are learnable parameter matrices, LN denotes layer normalization (Ba et al., 2016), $\odot$ denotes the Hadamard product, $\sigma(\cdot)$ denotes the logistic sigmoid, and $\text{GELU}(\cdot)$ denotes the Gaussian Error Linear Units (Hendrycks & Gimpel, 2016). The variables $u^\ell$ and $v^\ell$ respectively denote the edge and node features aggregated under the view size $\mathbf{F}_\ell$. In our message aggregation design, messages beyond the view $\mathbf{F}_\ell$ are masked with the zero vector $\mathbf{0}_d$, while the self-transformation is preserved. $\sigma(u_{i,j}) \in [0, 1]$ denotes element-wise gating scores. The variables $e$ and $x$ respectively denote the edge and node features refined by residual connection, normalization, and activation. Note that $\mathbf{W}_2$ and $\mathbf{W}_3$ are tied in STSPs, so as to exploit symmetry and avoid introducing redundant features. The discussion on node aggregation is provided in Appendix F

**Heatmap Decoder**   Finally, the edge-wise probability heatmap is decoded in a non-autoregressive manner as formulated:

$$\tilde{\mathbf{Y}}_{i,j} = \text{MLP}(\text{LN}(e_{i,j}^l + x_i^l + x_j^l)), \tag{9}$$

$$\mathbf{Y} = \text{Scatter}(\tilde{\mathbf{Y}}, \mathcal{X}), \tag{10}$$

$$\mathbf{H} = \sigma(\mathbf{Y}), \tag{11}$$

where $\text{MLP} : \mathbb{R}^d \to \mathbb{R}^1$, $\text{Scatter} : \mathbb{R}^{n \times k} \to \mathbb{R}^{n \times n}$, $e^l$, and $x^l$ denote the features from the last layer of the encoder, MLP is a two-layer multilayer perceptron with the GELU activation. To leverage all learnable parameters of the last-layer encoder, $e^l$ and $x^l$ are jointly decoded as Equation 9. Scatter serves to transform sparse logits $\tilde{\mathbf{Y}} \in \mathbb{R}^{n \times k}$ into full logits $\mathbf{Y} \in \mathbb{R}^{n \times n}$. In addition, the missing entries in $\mathbf{Y}$, as well as those on the main diagonal, are masked with $-100.0$ to avoid loss computation. The procedure of Scatter is presented in Appendix A. Finally, the heatmap $\mathbf{H}$ is obtained by applying a sigmoid activation to $\mathbf{Y}$, where $\mathbf{H}_{i,j} \in [0, 1]$ directly indicates the probability that the edge $\mathrm{x}_i \to \mathrm{x}_j$ is included in the predicted tour.

**Loss Function**   We train UNE-GCN using a binary cross-entropy loss with logits, as formulated:

$$\mathcal{L} = \frac{1}{n} \cdot \sum_{i=1}^n \sum_{j=1}^n \log(1 + \exp(-t_{i,j} \cdot \mathbf{Y}_{i,j})). \tag{12}$$

Let $\hat{\Pi}$ denote the (approximate) optimal tour that serves as the training label for **supervised learning**. The label variable $t \in \mathbb{R}^{n \times n}$ is defined such that $t_{i,j} = 1$ if the directed successor pair $i \to j$ is in $\hat{\Pi}$, and $t_{i,j} = -1$ otherwise. For STSPs, it holds that $t_{i,j} = t_{j,i}$.

## 5 EXPERIMENT

### 5.1 DATASETS

We adopt a mixed-scale dataset for training. Each training set of STSP/ATSP contains one million instances in total, with the proportions across different scales being STSP/ATSP-20:50:100 = 1:3:6. The generated test set consists of STSP/ATSP instances with scales of 20, 50, 100, 200, 500, 1K, 2K, 5K, and 10K. The number of instances is 1024 for scales of 20/50/100, 128 for scales of

200/500/1K, and 16 for scales of 2K/5K/10K. For the generated train and test sets, near-optimal solutions are obtained by LKH-3 with 10K trials and 1 run. Furthermore, ATSPs are transformed from STSPs by applying constrained perturbations to ensure a reasonable distribution of edge lengths. The procedure for generating STSPs and ATSPs is provided in Algorithm 2 of Appendix A.

## 5.2 TRAINING

Based on UNE-GCN, we design four additional ablation variants to evaluate their impact on graph encoding, namely $\sim$ w/o E, $\sim$ w/o U-E, $\sim$ w/o N-E, and $\sim$ w/o U-N-E. Specifically, variants without **Input Unification** (w/o U) adopt a learnable linear embedding ($\mathbb{R}^2 \to \mathbb{R}^d$) of node coordinates; without **Edge Normalization** (w/o N) adopt the unnormalized distance matrix **E**; without **Aggregation Enhancement** (w/o E) adopt a fixed view size. These variants broadly cover the existing GCN-based TSP encoder architectures, making the results more informative. Furthermore, we configure the hidden dimension $d$ of all models to 128, set the number of graph encoder layers $l \in \{6, 12, 18\}$, and fix the maximum view size $k$ at 50 to cover almost all candidate nodes. Based on these predefined $l$ and $k$, the corresponding *expanding* and *fixed* views are summarized in Table 1. We adopt the Adam optimizer (Kingma & Ba, 2015) and train the model for 10 epochs with a batch size of 32, where the learning rate is scheduled to decay from $1 \times 10^{-3}$ to 0 following a cosine annealing schedule (Loshchilov & Hutter, 2017). Note that the models for solving STSPs and ATSPs are trained separately on their respective training sets. All our experiments are conducted on an **Intel Xeon Platinum 8358P** CPU (32 cores, 2.60 GHz) and an **NVIDIA A800** GPU (SXM, 80 GB). In addition, a dual-encoder architecture is discussed in Appendix I.

Table 1: The settings of expanding and fixed view sizes for different numbers of layers.

| Type | #Layers ($l$) | View Sizes ($\mathbf{F} \in \mathbb{Z}^l$) |
|---|---|---|
| Expanding | 6 | [5, 10, 20, 30, 40, 50] |
| | 12 | [5, 5, 10, 10, 20, 20, 30, 30, 40, 40, 50, 50] |
| | 18 | [5, 5, 5, 10, 10, 10, 20, 20, 20, 30, 30, 30, 40, 40, 40, 50, 50, 50] |
| Fixed | 6, 12, 18 | $[50] \times l$ |

## 5.3 TESTING

**Pre-Evaluation** We adopt the metric proposed by Huang et al. (2025), **Missing Rate in Top-5 (MR@5)**, for pre-evaluation on the test sets. MR@5 measures the probability that the optimal successor does not rank among the top-5 candidates. Consequently, the generalization and learning robustness of models can be rapidly evaluated through heatmaps, eliminating the time-consuming post-search process. Intuitively, a smaller MR indicates better performance. The computation procedure of MR is provided in Algorithm 3 of Appendix A.

**Post-Search** We employ LKH-3 to perform post-search. In the LKH-3 settings, the size of candidate sets is fixed to 5 and 6 for STSPs and ATSPs, respectively; the number of runs is set to 1, and the number of trials is set to 10, 100, and 1K, respectively. For plain LKH-3, it employs the built-in Ascent algorithm (Helsgaun, 2000) to construct candidate sets. For two-stage methods, the candidate sets are defined as the top successors with the highest heatmap scores. Since LKH-3 does not support specifying candidate sets for ATSPs, we apply the JV Transformation (Jonker & Volgenant, 1983) to convert ATSPs into STSPs (detailed procedure provided in Appendix B and Algorithm 4 of Appendix A). The solution metrics for each instance include the two-stage solution time (**Time**), the tour length (**Length**), and the optimality gap (**Gap**). Specifically, the two-stage solution time consists of the inference time of the neural models and the runtime of the post-search; the optimality gap is defined as $L/\hat{L} - 1$, where $\hat{L}$ denotes the near-optimal tour length used for reference. Finally, the averages of **Time**, **Length**, and **Gap** are reported.

## 5.4 MAIN RESULTS

**Results** Figures 2 and 3 visualize the MR@5 evolution across training epochs for all variants on unseen-scale STSPs and ATSPs. All models are trained with 18-layer encoders. For ATSPs, variants

relying on node coordinate inputs (*e.g.*, ∼ w/o U-E, ∼ w/o U-N-E) are excluded due to the absence of such attributes. Across both problem types, **UNE-GCN consistently achieves the lowest MR@5**, converges faster, and generalizes more stably as problem scale increases. In addition, the visual evaluation of heatmaps is presented in Appendix J.

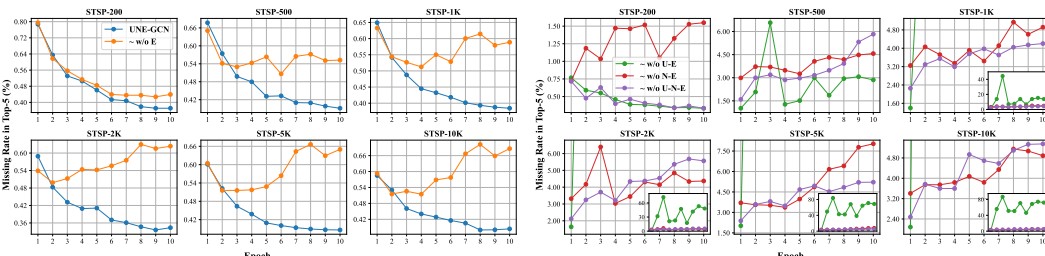

Figure 2: MR@5 (%) variation of ablations on unseen-scale STSPs.

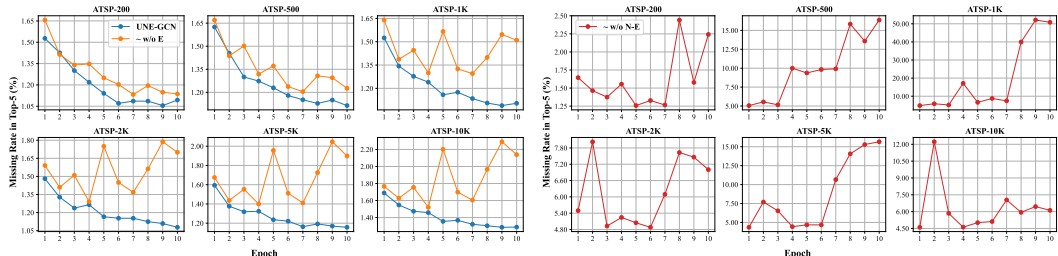

Figure 3: MR@5 (%) variation of ablations on unseen-scale ATSPs.

Tables 2 and 3 report full solution metrics on large-scale instances using model checkpoints from the last epoch. **UNE-GCN + LKH-3 outperforms all baselines** across both STSPs and ATSPs, achieving the lowest Gap values with significantly shorter inference time than plain LKH-3. On AT-SPs in particular, our method offers up to 2× speedup while improving solution quality. Appendix C presents further comparisons with other neural baselines, and Appendix K shows results on TSPLIB.

For massive-scale STSPs (*n* up to 100K), Table 5 shows that our 6-layer UNE-GCN significantly outperforms NeuroLKH (Xin et al., 2021) within the same search time. Despite having 8.5× fewer parameters and being trained on smaller-scale problems, UNE-GCN still generalizes well to very large instances, while NeuroLKH underperforms even plain LKH-3 under the same search time. Moreover, training and model configurations are summarized in Table 4.

**Analysis** The observed MR@5 trends suggest that feature design plays a critical role in generalization. Variants without layer-wise expanding views (∼ w/o E) show degraded performance and limited robustness, confirming the importance of multi-view learning. Models lacking unified representations (*e.g.*, ∼ w/o U-E, ∼ w/o N-E, ∼ w/o U-N-E) suffer from instability on large scales, often overfitting to narrow patterns learned from small instances. In contrast, UNE-GCN maintains stable improvements across scales and epochs, demonstrating both effective representations and strong cross-scale generalization.

On large- and massive-scale STSPs, the consistent advantage of UNE-GCN + LKH-3 confirms that well-encoded prior knowledge significantly reduces search complexity. Its GPU-accelerated inference allows constructing high-quality candidate sets in less time, which is particularly beneficial when scaling beyond 10K nodes. In conclusion, UNE-GCN provides a promising and fast approach for constructing candidate sets for large-scale TSPs.

## 5.5 SCALING LAWS

Under proper encoding, the performance of UNE-GCN steadily improves with increasing parameter scale (as shown in Figure 4). In contrast, other ablation variants, constrained by noise and narrow representations, tend to suffer from overfitting and poor generalization for unseen-scale problems (as shown in Figure 5–8). Moreover, these issues may become even more severe as the parameter scale increases. The complete results of the Scaling Laws are provided in Appendix L.

Table 2: Solution metrics comparison of ablations on large-scale STSPs. The Length of plain LKH-3 are used as near-optimal solutions to compute Gap. The Gap is calculated under the same Trials. For clarity of presentation, the first-, second-, and third-smallest gaps are highlighted in red, blue, and **bold**, respectively.

| Method | Trials | STSP-1K | | | STSP-2K | | | STSP-5K | | | STSP-10K | | |
|---|---|---|---|---|---|---|---|---|---|---|---|---|---|
| | | Length | Gap(%) | Time | Length | Gap(%) | Time | Length | Gap(%) | Time | Length | Gap(%) | Time |
| LKH-3 | 10 | 23.1300 | 0.0000 | 1.17s | 32.4942 | **0.0000** | 4.69s | 51.0049 | **0.0000** | 33.17s | 71.8454 | **0.0000** | 2.65m |
| | 100 | 23.1228 | **0.0000** | 1.70s | 32.4807 | **0.0000** | 6.01s | 50.9813 | **0.0000** | 40.06s | 71.7957 | **0.0000** | 3.08m |
| | 1K | 23.1192 | **0.0000** | 6.45s | 32.4762 | 0.0000 | 17.88s | 50.9724 | **0.0000** | 1.45m | 71.7800 | **0.0000** | 5.44m |
| ∼ w/o U-N-E | | | | 0.02s | | | 0.05s | | | 0.12s | | | 0.33s |
| + LKH-3 | 10 | 23.2291 | 0.4288 | 0.54s | 32.7178 | 0.6878 | 1.87s | 51.2913 | 0.5616 | 9.81s | 72.2587 | 0.5751 | 54.20s |
| | 100 | 23.1686 | 0.1983 | 1.64s | 32.5916 | 0.3414 | 5.93s | 51.1363 | 0.3042 | 17.80s | 72.0000 | 0.2846 | 1.38m |
| | 1K | 23.1402 | 0.0906 | 12.15s | 32.5178 | 0.1281 | 49.92s | 51.0457 | 0.1438 | 1.70m | 71.8734 | 0.1301 | 5.34m |
| ∼ w/o U-E | | | | 0.02s | | | 0.05s | | | 0.12s | | | 0.33s |
| + LKH-3 | 10 | 24.6190 | 6.4368 | 0.52s | 42.2898 | 30.1471 | 1.60s | 72.5796 | 42.2989 | 10.90s | 104.0977 | 44.8925 | 58.48s |
| | 100 | 23.8542 | 3.1643 | 1.03s | 38.9663 | 19.9685 | 2.63s | 66.8056 | 31.0381 | 16.09s | 96.1445 | 33.9144 | 1.34m |
| | 1K | 23.4054 | 1.2384 | 5.38s | 35.0405 | 7.8964 | 11.72s | 58.4045 | 14.5799 | 1.17m | 83.7188 | 16.6312 | 4.48m |
| ∼ w/o N-E | | | | 0.02s | | | 0.05s | | | 0.12s | | | 0.33s |
| + LKH-3 | 10 | 23.2486 | 0.5130 | 0.72s | 32.6494 | 0.4777 | 1.59s | 51.4861 | 0.9434 | 10.85s | 72.3836 | 0.7491 | 55.12s |
| | 100 | 23.1789 | 0.2429 | 2.59s | 32.5713 | 0.2791 | 3.51s | 51.2243 | 0.4766 | 24.78s | 71.9940 | 0.2762 | 1.29m |
| | 1K | 23.1430 | 0.1029 | 22.78s | 32.5095 | 0.1028 | 21.40s | 51.0753 | 0.2019 | 3.14m | 71.8761 | 0.1339 | 4.04m |
| ∼ w/o E | | | | 0.02s | | | 0.05s | | | 0.12s | | | 0.33s |
| + LKH-3 | 10 | 23.1301 | **0.0003** | 0.38s | 32.4931 | -0.0033 | 1.34s | 51.0032 | -0.0033 | 9.90s | 71.8272 | -0.0253 | 54.32s |
| | 100 | 23.1217 | -0.0048 | 0.95s | 32.4803 | -0.0011 | 2.60s | 50.9766 | -0.0091 | 15.90s | 71.7925 | -0.0045 | 1.31m |
| | 1K | 23.1186 | -0.0027 | 6.21s | 32.4777 | **0.0044** | 14.81s | 50.9695 | -0.0057 | 1.13m | 71.7797 | -0.0004 | 4.18m |
| UNE-GCN | | | | 0.02s | | | 0.05s | | | 0.12s | | | 0.31s |
| + LKH-3 | 10 | 23.1263 | -0.0162 | 0.38s | 32.4891 | -0.0157 | 1.37s | 51.0012 | -0.0072 | 9.92s | 71.8277 | -0.0246 | 55.21s |
| | 100 | 23.1201 | -0.0117 | 0.94s | 32.4789 | -0.0056 | 2.60s | 50.9770 | -0.0083 | 17.04s | 71.7883 | -0.0102 | 1.34m |
| | 1K | 23.1184 | -0.0034 | 6.36s | 32.4757 | -0.0015 | 13.68s | 50.9694 | -0.0059 | 1.26m | 71.7779 | -0.0030 | 4.34m |

Table 3: Solution metrics comparison of ablations on large-scale ATSPs.

| Method | Trials | ATSP-1K | | | ATSP-2K | | | ATSP-5K | | | ATSP-10K | | |
|---|---|---|---|---|---|---|---|---|---|---|---|---|---|
| | | Length | Gap(%) | Time | Length | Gap(%) | Time | Length | Gap(%) | Time | Length | Gap(%) | Time |
| LKH-3 | 10 | 20.6466 | **0.0000** | 4.74s | 29.0925 | **0.0000** | 21.49s | 45.8144 | **0.0000** | 2.81m | 64.5529 | **0.0000** | 13.22m |
| | 100 | 20.5081 | **0.0000** | 6.17s | 28.8698 | **0.0000** | 25.16s | 45.5152 | **0.0000** | 3.03m | 64.1310 | **0.0000** | 13.84m |
| | 1K | 20.4698 | **0.0000** | 20.07s | 28.7799 | **0.0000** | 1.00m | 45.3654 | **0.0000** | 5.09m | 63.9332 | **0.0000** | 19.21m |
| ∼ w/o N-E | | | | 0.03s | | | 0.08s | | | 0.18s | | | 0.40s |
| + LKH-3 | 10 | 22.8018 | 10.4407 | 1.60s | 29.2808 | 0.6466 | 6.92s | 51.7784 | 13.0062 | 1.10m | 73.1467 | 13.3294 | 6.45m |
| | 100 | 21.6228 | 5.4377 | 2.60s | 28.9785 | 0.3769 | 9.72s | 47.1358 | 3.5618 | 1.39m | 64.5697 | 0.6848 | 6.69m |
| | 1K | 20.9451 | 2.3221 | 12.40s | 28.8331 | 0.1852 | 37.18s | 46.1316 | 1.6886 | 4.62m | 64.1786 | 0.3844 | 11.50m |
| ∼ w/o E | | | | 0.03 | | | 0.08s | | | 0.18s | | | 0.40s |
| + LKH-3 | 10 | 20.5873 | -0.2863 | 1.56s | 28.9616 | -0.4499 | 6.87s | 45.6025 | -0.4619 | 1.17m | 64.3726 | -0.2779 | 6.26m |
| | 100 | 20.4949 | -0.0640 | 2.42s | 28.8186 | -0.1767 | 9.25s | 45.3875 | -0.2801 | 1.32m | 63.9788 | -0.2366 | 6.96m |
| | 1K | 20.4694 | -0.0020 | 10.75s | 28.7643 | -0.0540 | 32.12s | 45.2874 | -0.1716 | 2.85m | 63.8817 | -0.0799 | 11.19m |
| UNE-GCN | | | | 0.03s | | | 0.07s | | | 0.16s | | | 0.36s |
| + LKH-3 | 10 | 20.5765 | -0.3385 | 1.56s | 28.9178 | -0.6004 | 7.00s | 45.5651 | -0.5429 | 1.16m | 64.2428 | -0.4795 | 6.42m |
| | 100 | 20.4903 | -0.0863 | 2.44s | 28.8134 | -0.1950 | 9.38s | 45.3731 | -0.3121 | 1.32m | 63.9460 | -0.2878 | 6.95m |
| | 1K | 20.4676 | -0.0106 | 10.88s | 28.7657 | -0.0489 | 32.65s | 45.2756 | -0.1978 | 2.88m | 63.8243 | -0.1696 | 11.15m |

Table 4: Comparison of training and model configurations between NeuroLKH and UNE-GCN.

| Model | Problem scale | #Samples | #Parameters | #Layers | Hidden dimension |
|---|---|---|---|---|---|
| NeuroLKH | [101, 500] | 780K | 3.553M | 30 | 128 |
| UNE-GCN | {20, 50, 100} | 1000K | 0.417M | 6 | 128 |

Table 5: Solution metrics comparison on massive-scale STSPs with the same LKH-3 search time (excluding the candidate set construction time). The Trials column shows the average number of trials executed within a given time. The Gap is calculated under the same Time.

| Method | Time (s) | STSP-5K | | | STSP-10K | | | STSP-50K | | | STSP-100K | | |
|---|---|---|---|---|---|---|---|---|---|---|---|---|---|
| | | Length | Gap(%) | Trials | Length | Gap(%) | Trials | Length | Gap(%) | Trials | Length | Gap(%) | Trials |
| LKH-3 | $0.01n$ | 50.9722 | 0.0000 | 946 | 71.7825 | 0.0000 | 588 | 160.1223 | 0.0000 | 205 | 226.2792 | 0.0000 | 147 |
| | $0.02n$ | 50.9700 | 0.0000 | 1931 | 71.7800 | 0.0000 | 1222 | 160.0881 | 0.0000 | 418 | 226.1872 | 0.0000 | 292 |
| | $0.03n$ | 50.9689 | 0.0000 | 2896 | 71.7786 | 0.0000 | 1852 | 160.0708 | 0.0000 | 633 | 226.1673 | 0.0000 | 432 |
| NeuroLKH | | | | | | | | | | | | | |
| + LKH-3 | $0.01n$ | 50.9734 | 0.0025 | 337 | 71.8030 | 0.0284 | 259 | 160.2625 | 0.0876 | 466 | 226.5162 | 0.1048 | 338 |
| | $0.02n$ | 50.9708 | 0.0014 | 701 | 71.7961 | 0.0224 | 532 | 160.1843 | 0.0602 | 967 | 226.3877 | 0.0886 | 705 |
| | $0.03n$ | 50.9703 | 0.0029 | 1071 | 71.7935 | 0.0207 | 815 | 160.1449 | 0.0464 | 1429 | 226.3260 | 0.0702 | 1060 |
| UNE-GCN ($l$=6) | | | | | | | | | | | | | |
| + LKH-3 | $0.01n$ | 50.9695 | -0.0052 | 909 | 71.7793 | -0.0045 | 510 | 160.0491 | -0.0456 | 164 | 226.1802 | -0.0437 | 128 |
| | $0.02n$ | 50.9674 | -0.0052 | 1856 | 71.7776 | -0.0034 | 1038 | 160.0331 | -0.0343 | 326 | 226.1568 | -0.0134 | 256 |
| | $0.03n$ | 50.9673 | -0.0030 | 2316 | 71.7769 | -0.0023 | 1318 | 160.0242 | -0.0290 | 489 | 226.1437 | -0.0104 | 383 |

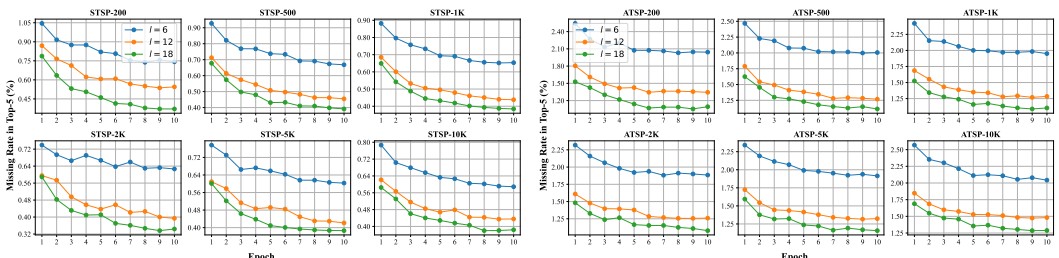

Figure 4: Scaling Laws for UNE-GCN on unseen-scale STSPs and ATSPs.

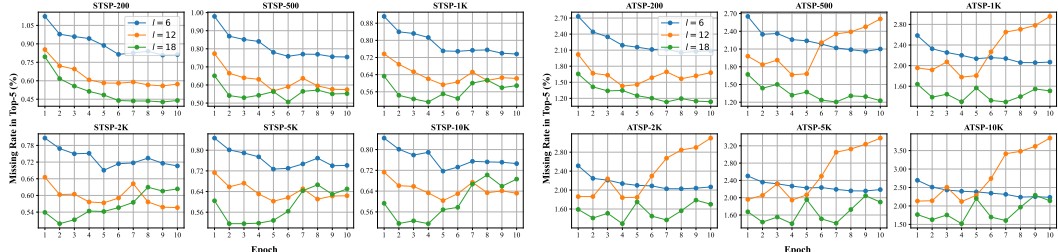

Figure 5: Scaling Laws for ∼ w/o E on unseen-scale STSPs and ATSPs.

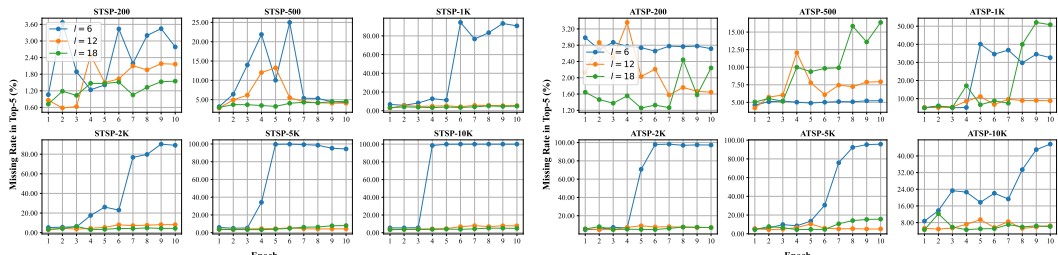

Figure 6: Scaling Laws for ∼ w/o N-E on unseen-scale STSPs and ATSPs.

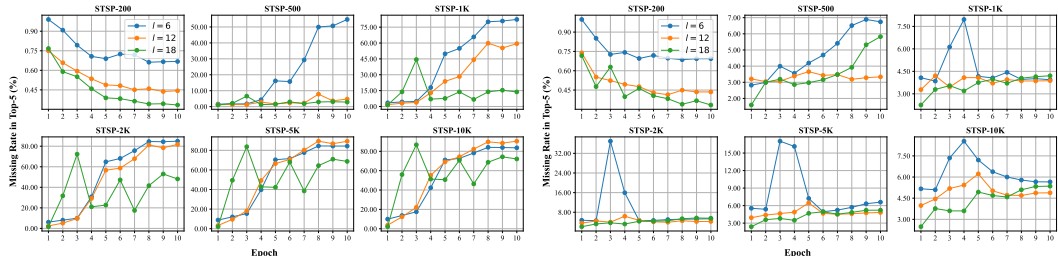

Figure 7: Scaling Laws for ∼ w/o U-E on unseen-scale STSPs.

Figure 8: Scaling Laws for ∼ w/o U-N-E on unseen-scale STSPs.

# 6 CONCLUSION

In this work, we systematically reflect on the key factors that influence the performance of GCN-based TSP solvers, aiming at more effective graph encoding. We address this goal from three perspectives–**Input Unification**, **Edge Normalization**, and **Aggregation Enhancement**. By integrating **U-N-E**, our UNE-GCN achieves more effective, robust, and generalizable representations for both STSPs and ATSPs. In comprehensive and illustrative experiments, we validate the impact of each component of **U-N-E** based on the MR@5 metric. Furthermore, UNE-GCN demonstrates a strong guiding effect on LKH-3, enabling it to obtain superior solutions with less search time, particularly on large-scale ATSPs.

# 7 ETHICS STATEMENT

This research complies with ethical standards. It utilizes datasets that are either synthetic or publicly available, and contains no sensitive or personally identifiable information. The study involves no direct human subjects, nor does it pose any privacy or security concerns. All methodologies and experiments were conducted in accordance with applicable laws and established research integrity practices. There are no conflicts of interest, no undue influence from external sponsorship, and no concerns related to discrimination, bias, or fairness. Moreover, this research does not lead to any harmful insights or applications.

# 8 REPRODUCIBILITY STATEMENT

We have taken steps to ensure the reproducibility of the results presented in this paper. The experimental settings, including datasets and model designs, are thoroughly described in Section 5. Additional details, such as algorithm details and configurations, are provided in Appendix A. Source code will be made publicly available upon acceptance.

# 9 LLM USAGE STATEMENT

In this work, large language models (LLMs) were used exclusively to assist with writing, editing, and LaTeX formatting. Their role was confined to enhancing clarity, grammar, and overall presentation; they had no impact on the design of experiments, data processing, analysis, or the interpretation of results.

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

## A  ALGORITHM DESCRIPTION

---

**Algorithm 1:** Procedure of Scatter

---

**Input:** Sparse logits $\tilde{\mathbf{Y}} \in \mathbb{R}^{n \times k}$, node set $\mathcal{X} = \{x_i\}_{i=1}^n$.
**Output:** A full logits $\mathbf{Y} \in \mathbb{R}^{n \times n}$.
Initialize $\mathbf{Y} \leftarrow (-100) \cdot \mathbf{1}_{n \times n}$;
**for** $i \leftarrow 1$ **to** $n$ **do**
    **for** $j \leftarrow 2$ **to** $k$ **do**
        $x_i^j \leftarrow$ the $j$-th nearest neighbor of $x_i$;
        $k \leftarrow \text{Index}(x_i^j, \mathcal{X})$ ;        `// index of the node `$x_i^j$` in `$\mathcal{X}$
        $\mathbf{Y}_{i,k} \leftarrow \tilde{\mathbf{Y}}_{i,j}$ ;        `// `$x_i^j = x_k$
**return** $\mathbf{Y}$;

---

**Algorithm 2:** Procedure of Generating STSPs and ATSPs

---

**Input:** Problem type, problem scale $n$.
**Output:** An instance of STSP-$n$ or ATSP-$n$.
Generate node coordinates $\mathcal{X} = \{x_i\}_{i=1}^n$, with $x_i \sim \text{Uniform}(0, 1)^2$;
Compute the Euclidean distance matrix $\mathbf{D} \in \mathbb{R}^{n \times n}$ based on $\mathcal{X}$ ;
**if** problem type is ATSP **then**
    **for** $i \leftarrow 1$ **to** $n$ **do**
        **for** $j \leftarrow i + 1$ **to** $n$ **do**
            Sample $\tau \sim \text{Uniform}(-0.4, 0.4)$ ;    `// perturbation factor `$\tau$
            $\mathbf{D}_{i,j} \leftarrow \mathbf{D}_{i,j} \cdot (1 + \tau)$;
            $\mathbf{D}_{j,i} \leftarrow \mathbf{D}_{j,i} \cdot (1 - \tau)$;
**return** $\mathcal{X}, \mathbf{D}$;

---

**Algorithm 3:** Procedure of Computing Missing Rate in Top-$p$

---

**Input:** Problem type, positive integer $p$, near-optimal tour $\hat{\Pi} \in \mathbb{R}^n$, full heatmap $\mathbf{H} \in \mathbb{R}^{n \times n}$.
**Output:** A non-negative scalar $r$ indicating Missing Rate in Top-$p$.
Initialize $c \leftarrow 0$;
**for** $i \leftarrow 1$ **to** $n$ **do**
    $\mathcal{P} \leftarrow \text{Top}(p, \mathbf{H}_i)$ ;    `// indices of the `$p$` largest elements in `$\mathbf{H}_i$
    $k \leftarrow \text{Index}(i, \hat{\Pi})$ ;    `// index of the element `$i$` in `$\hat{\Pi}$
    **if** $\hat{\Pi}_{k+1} \in \mathcal{P}$ **then**
        $c \leftarrow c + 1$;
    **if** problem type is STSP **and** $\hat{\Pi}_{k-1} \in \mathcal{P}$ **then**
        $c \leftarrow c + 1$ ;    `// the undirected nature of tours in STSP`
**if** problem type is STSP **then**
    $r \leftarrow c/(2n)$;
**else**
    $r \leftarrow c/n$;
**return** $r$;

---

As illustrated in Algorithm 1, the full logits $\mathbf{Y}$ are initialized as a matrix filled with $-100$. For each row $\tilde{\mathbf{Y}}_i$ of the sparse logits, we obtain the global index $k$ in $\mathcal{X}$ starting from $x_i^2$ (excluding the self-to-self logits $x_i^1$). Finally, the entries $\tilde{\mathbf{Y}}_{i,j}$ in the sparse logits are scattered to their corresponding positions $\mathbf{Y}_{i,k}$ in the full logits. The purpose of the operation Scatter is to convert local relative in-

dices in the sparse matrix into globally unique indices, thereby allowing for a more concise symbolic representation.

As illustrated in Algorithm 2, we first generate an STSP instance: a node set $\mathcal{X}$ with two-dimensional coordinates and the corresponding Euclidean distance matrix $\mathbf{D}$ derived from $\mathcal{X}$. The nodes are uniformly distributed within the unit square $[0, 1]^2$. Next, if the target problem type is ATSP, we uniformly sample a perturbation factor $\tau \in (-0.4, 0.4)$. Then, for each distance pair $(\mathbf{D}_{i,j}, \mathbf{D}_{j,i})$ with $i < j$, we assign asymmetric values while ensuring that the sum of the new distance pair $(\mathbf{D}_{i,j}, \mathbf{D}_{j,i})$ remains equal to that of the original pair. This procedure transforms STSPs into ATSPs, making the resulting distance matrix closer to real-world distributions rather than purely random ones.

As shown in Algorithm 3, when computing the MR, the undirected nature of the STSP is taken into account, which requires considering the predecessor $\hat{\Pi}_{k-1}$ in addition to the ATSP case. Note that $\hat{\Pi}_{k\pm1}$ are defined with cyclic indexing, since the TSP tour is a cycle, *i.e.*, $\hat{\Pi}_{n+1} = \hat{\Pi}_1$ and $\hat{\Pi}_0 = \hat{\Pi}_n$.

---

**Algorithm 4:** Procedure of JV Transformation

---

**Input:** Asymmetric distance matrix $\mathbf{D} \in \mathbb{R}^{n \times n}$.
**Output:** A symmetric distance matrix $\mathbf{S} \in \mathbb{R}^{2n \times 2n}$.
Initialize $\mathbf{S} \leftarrow \infty \cdot \mathbf{1}_{2n \times 2n}$;
**for** $i \leftarrow n + 1$ **to** $2n$ **do**
    **for** $j \leftarrow 1$ **to** $n$ **do**
        $\mathbf{S}_{i,j} \leftarrow \mathbf{D}_{i-n,j}$;
        $\mathbf{S}_{j,i} \leftarrow \mathbf{D}_{i-n,j}$;

**return** $\mathbf{S}$

---

As shown in Algorithm 4, the new symmetric distance matrix $\mathbf{S}$ is initialized as an all-$\infty$ matrix with doubled dimensions. In practical implementation, $\infty$ is approximated by a sufficiently large constant exceeding any edge length. Then, the asymmetric distance matrix $\mathbf{D}$ and its transpose $\mathbf{D}^\top$ are placed in the lower-left and upper-right blocks of $\mathbf{S} \in \mathbb{R}^{2n \times 2n}$, respectively. A concrete example is shown in Equation 13. The transformation splits each original node $\mathrm{x}_i$ into an in-node $\mathrm{x}_i$ and an out-node $\mathrm{x}_{i+n}$. Consequently, the node set is expanded to $\mathcal{X} = \{\mathrm{x}_1, \mathrm{x}_2, \ldots, \mathrm{x}_{2n}\}$. For any $i \in \{1, 2, \ldots, n\}$ in $\mathbf{S}$, it holds that $\mathbf{S}_{i,i+n} = \mathbf{S}_{i+n,i} = 0$, which imposes a soft constraint that the in- and out-nodes occur as pairs in the tour. In the problem input files of LKH-3, `FIXED_EDGES_SECTION` can be specified to enforce that in- and out-nodes always appear as pairs. Specifically, each pair $(i, i + n)$ is listed in `FIXED_EDGES_SECTION` for $i \in \{1, 2, \ldots, n\}$.

$$\mathbf{D} = \begin{bmatrix} 0 & 1 & 2 \\ 3 & 0 & 4 \\ 5 & 6 & 0 \end{bmatrix} \xrightarrow{\text{JV Transformation}} \mathbf{S} = \begin{bmatrix} \boldsymbol{\infty} & \mathbf{D}^\top \\ \mathbf{D} & \boldsymbol{\infty} \end{bmatrix} = \begin{bmatrix} \infty & \infty & \infty & 0 & 3 & 5 \\ \infty & \infty & \infty & 1 & 0 & 6 \\ \infty & \infty & \infty & 2 & 4 & 0 \\ 0 & 1 & 2 & \infty & \infty & \infty \\ 3 & 0 & 4 & \infty & \infty & \infty \\ 5 & 6 & 0 & \infty & \infty & \infty \end{bmatrix} \quad (13)$$

## B  CANDIDATE SET CONSTRUCTION FOR ATSPs

Since LKH-3 does not support specifying candidate sets for ATSPs, we first convert an ATSP-$n$ into an STSP-$2n$, as shown in Algorithm 4. Next, we define the sparse distance matrices $\tilde{\mathbf{E}}^\mathbb{A}$ and $\tilde{\mathbf{E}}^\mathbb{B}$, which are derived from the full distance matrix $\mathbf{D}$ and its transpose $\mathbf{D}^\top$, respectively, following the procedure described in Section 4.1. Consequently, $\tilde{\mathbf{E}}^\mathbb{A}$ denotes out-distance matrix, while $\tilde{\mathbf{E}}^\mathbb{B}$ denotes in-distance matrix. The corresponding heatmaps $\mathbf{H}^\mathbb{A}$ and $\mathbf{H}^\mathbb{B}$ are computed independently by following the procedure described in Section 4.4.

$$\mathbf{H}^\alpha = \text{UNE-GCN}(\mathcal{X}, \tilde{\mathbf{E}}^\alpha), \ \forall \alpha \in \{\mathbb{A}, \mathbb{B}\}, \quad (14)$$

where $\mathbf{H}^\mathbb{A}$ and $\mathbf{H}^\mathbb{B}$ indicate the candidate sets of nodes $\mathrm{x}_{i+n}$ and $\mathrm{x}_i$ with $i \in \{1, 2, \ldots, n\}$, respectively (*i.e.*, corresponding to the lower-left and upper-right blocks of $\mathbf{S}$). Furthermore, nodes $\mathrm{x}_i$ and

$x_{i+n}$ must be assigned as candidates of each other with the highest priority to guarantee their pairing. This is also why the candidate set size is set to 5 for STSPs, whereas it is set to 6 for ATSPs and for STSPs transformed from ATSPs.

## C  COMPARISON WITH BASELINES

For non-neural baselines, we adopt **1) Exact Solver:** Concorde (Applegate et al., 2006); **2) Heuristic Solvers/Algorithms:** LKH-3 (Helsgaun, 2017), MCTS (Fu et al., 2021), RBS (Huang et al., 2025), and 2-Opt (Croes, 1958). For neural baselines, we adopt **1) Transformer-Based Methods:** H-TSP (Pan et al., 2023), GLOP (Ye et al., 2024), LEHD (Luo et al., 2023), and DRHG (Li et al., 2025a); **2) GCN-Based Methods:** Att-GCN (Fu et al., 2021), DIFUSCO (Sun & Yang, 2023), Fast T2T (Li et al., 2024), RsGCN (Huang et al., 2025), and NeuroLKH (Xin et al., 2021). For our method, we adopt UNE-GCN with 6-layer and 18-layer encoders, denoted as UNE-GCN (6) and UNE-GCN (18), respectively.

We reproduce Concorde, LKH-3, and NeuroLKH, while the other results are taken from the work of Huang et al. (2025). For fairness, we uniformly employ the latest version LKH-*3.0.13*. We note that the performance of GCN-based methods is affected by the post-search algorithm. Here, we focus on the comparison among LKH-3, NeuroLKH + LKH-3, and UNE-GCN + LKH-3, while the other methods are included only as a point of reference. Figure 9 reports the number of learnable parameters for each neural baseline, highlighting that our UNE-GCN is relatively lightweight. Note that for STSP-50K and STSP-100K in Table 5, we disable subgradient optimization (Helsgaun, 2000) in LKH-3 (which calculates more precise node penalties) to avoid excessively long preprocessing times.

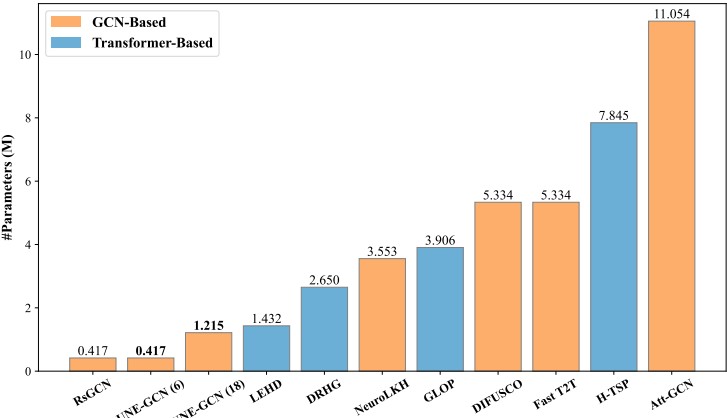

Figure 9: Comparison of the number of learnable parameters (M) among neural baselines.

Tables 6 and 7 show the solution metrics of the baselines on large-scale STSPs and ATSPs, with LKH-3 trials set to 1K. To ensure runtime comparability, we set the time limit of Concorde to $0.1n$ for STSPs and $0.5n$ for ATSPs. Since Concorde does not natively support ATSPs, we solve them by first applying the JV Transformation, as shown in Algorithm 4, to convert ATSPs into STSPs. Note that MatNet (Kwon et al., 2021), MatPOENet, and MatDIFFNet (Pan et al., 2025) have only been applied to small-scale ATSPs, and there is a lack of neural baselines specialized in solving large-scale ATSPs; therefore, we do not include them in the comparison. Overall, UNE-GCN achieves advanced performance on large-scale STSPs and ATSPs with a relatively lightweight design and remarkable generalization.

## D  TRAINING ON LARGE-SCALE STSPS

We employ LKH-3 with 10K trials and 1 run to additionally generate a large-scale STSP training set, which consists of 10K STSP-200, 30K STSP-500, and 60K STSP-1K instances. Based on this dataset, we train another UNE-GCN, using the same training configuration as described in

Table 6: Solution metrics comparison of baselines on large-scale STSPs.

| Method | Type | STSP-1K Length | Gap(%) | Time | STSP-2K Length | Gap(%) | Time | STSP-5K Length | Gap(%) | Time | STSP-10K Length | Gap(%) | Time |
|---|---|---|---|---|---|---|---|---|---|---|---|---|---|
| Concorde | Exact | 23.1198 | 0.0024 | 1.31m | 32.4880 | 0.0363 | 3.46m | 51.0353 | 0.1234 | 8.88m | 71.9349 | 0.2158 | 17.71m |
| LKH-3 | Heuristic | 23.1192 | 0.0000 | 6.45s | 32.4762 | **0.0000** | 17.88s | 50.9724 | 0.0000 | 1.45m | 71.7800 | **0.0000** | 5.44m |
| MCTS | Heuristic | 23.6622 | 2.3487 | 1.81m | 33.3675 | 2.7445 | 3.39m | 52.6930 | 3.3756 | 8.66m | 74.1944 | 3.3636 | 17.03m |
| RBS | Heuristic | 23.2756 | 0.6765 | 52.38s | 32.7452 | 0.8283 | 1.68m | 51.5199 | 1.0741 | 4.18m | 72.8742 | 1.5244 | 8.37m |
| 2-Opt | Heuristic | 24.7392 | 7.0072 | 0.12s | 34.9866 | 7.7300 | 0.49s | 55.1501 | 8.1960 | 7.87s | 77.8939 | 8.5176 | 51.88s |
| H-TSP | Transformer | 24.6679 | 6.6988 | 36.97s | 34.8984 | 7.4584 | 9.42s | 55.0178 | 7.9365 | 20.08s | 77.7446 | 8.3096 | 38.57s |
| GLOP | Transformer | 23.8377 | 3.1078 | 1.51m | 33.6595 | 3.6436 | 1.80m | 53.1567 | 4.2853 | 4.56m | 75.0439 | 4.5471 | 9.01m |
| LEHD | Transformer | 23.6907 | 2.4720 | 1.48m | 34.2119 | 5.3445 | 2.15m | 59.6522 | 17.0284 | 9.59m | 90.5505 | 26.1500 | 54.45m |
| DRHG | Transformer | 23.3217 | 0.8759 | 1.50m | 32.8920 | 1.2803 | 2.01m | 51.6350 | 1.2999 | 5.00m | 72.8973 | 1.5566 | 9.00m |
| Att-GCN + MCTS | GCN Heuristic | 23.6454 | 2.2760 | 0.34s 1.71m | — | | | — | | | 74.3267 | 3.5479 | 15.60s 17.38m |
| DIFUSCO + MCTS | GCN Heuristic | 23.4243 | 1.3197 | 0.12s 1.71m | — | | | — | | | 73.8913 | 2.9413 | 0.14s 17.55m |
| Fast T2T + 2-Opt | GCN Heuristic | 23.3122 | 0.8348 | 2.95m | 32.7940 | 0.9786 | 2.03m | 51.7409 | 1.5077 | 4.35m | 72.9450 | 1.6230 | 17.00m |
| RsGCN + RBS | GCN Heuristic | 23.2210 | 0.4403 | 0.01s 50.40s | 32.6784 | 0.6226 | 0.03s 1.68m | 51.3987 | 0.8363 | 0.18s 4.17m | 72.6335 | 1.1890 | 0.74s 8.36m |
| NeuroLKH + LKH-3 | GCN Heuristic | 23.1182 | -0.0044 | 0.05s 10.10s | 32.4752 | -0.0033 | 0.10s 29.89s | 50.9704 | **-0.0038** | 0.29s 2.37m | 71.7927 | 0.0176 | 0.52s 6.41m |
| UNE-GCN (6) + LKH-3 | GCN Heuristic | 23.1186 | **-0.0026** | 0.01s 5.74s | 32.4763 | 0.0002 | 0.03s 14.34s | 50.9688 | -0.0071 | 0.08s 1.08m | 71.7774 | -0.0036 | 0.26s 4.05m |
| UNE-GCN (18) + LKH-3 | GCN Heuristic | 23.1184 | -0.0034 | 0.02s 6.36s | 32.4757 | -0.0015 | 0.05s 13.68s | 50.9694 | -0.0059 | 0.12s 1.26m | 71.7779 | -0.0030 | 0.31s 4.34m |

Table 7: Solution metrics comparison of baselines on large-scale ATSPs.

| Method | Type | ATSP-1K Length | Gap(%) | Time | ATSP-2K Length | Gap(%) | Time | ATSP-5K Length | Gap(%) | Time | ATSP-10K Length | Gap(%) | Time |
|---|---|---|---|---|---|---|---|---|---|---|---|---|---|
| Concorde | Exact | 20.6862 | 1.0569 | 9.21m | 29.5484 | 2.6713 | 18.00m | 47.2387 | 4.1295 | 43.91m | 67.4584 | 5.5157 | 1.45h |
| LKH-3 | Heuristic | 20.4698 | 0.0000 | 20.07s | 28.7799 | **0.0000** | 1.00m | 45.3654 | **0.0000** | 5.09m | 63.9332 | **0.0000** | 19.21m |
| UNE-GCN (6) + LKH-3 | GCN Heuristic | 20.4723 | **0.0121** | 0.02s 11.14s | 28.7651 | -0.0511 | 0.05s 32.83s | 45.2939 | -0.1575 | 0.10s 2.90m | 63.8859 | -0.0733 | 0.26s 11.27m |
| UNE-GCN (18) + LKH-3 | GCN Heuristic | 20.4676 | -0.0106 | 0.02s 10.88s | 28.7657 | -0.0489 | 0.05s 32.65s | 45.2756 | -0.1978 | 0.12s 2.88m | 63.8243 | -0.1696 | 0.31s 11.15m |

Section 5.2, except that the batch size is set to 16. The training performance on both small-scale and large-scale STSPs is shown in Figure 10. Since STSP-200/500/1K share feature distributions more similar to those of STSP-2K/5K/10K, the large-scale training set requires only one-tenth of the sample size of the small-scale training set to achieve better generalization on STSP-2K/5K/10K. It is worth noting that generating and training on large-scale datasets is more costly, with the time and resource consumption reported in Table 8. To strike a balance between data quality and cost, future work should investigate mixed-scale datasets with more diverse scales and appropriate proportions.

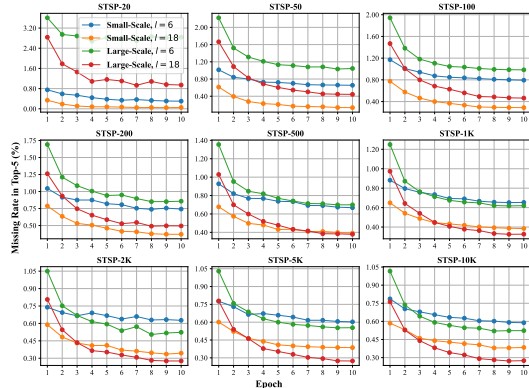

Figure 10: MR@5 (%) variation of training on large-scale STSP.

Table 8: Training cost statistics on small-scale and large-scale datasets with $l = 18$.

| Training set | #Samples | Batch size | Memory usage | Time per epoch |
|---|---|---|---|---|
| STSP-20/50/100 | 1000K | 32 | 6.22GB | 0.77h |
| STSP-200/500/1K | 100K | 16 | 36.62GB | 1.14h |

## E  DISCUSSION ON NODE EMBEDDING

We discuss four types of node embeddings: (a) 2D coordinate linear embedding, (b) 2D random-value linear embedding, (c) zero embedding, and our (d) global learnable embedding, as presented in Equation 15 from top to bottom:

$$x_i^0 = \begin{cases} \text{Linear}(\mathbf{x}_i) & : \mathbb{R}^2 \to \mathbb{R}^d, \\ \text{Linear}(\text{Uniform}(0,1)^2) & : \mathbb{R}^2 \to \mathbb{R}^d, \\ \mathbf{0}_d, \\ \mathbf{X}. \end{cases} \tag{15}$$

2D coordinate embedding is widely adopted in most existing works, but it introduces biases that are detrimental to generalization, corresponding to our ablation variant $\sim$ w/o U-E. 2D random-value embedding is employed in MatDIFFNet (Pan et al., 2025) for node embedding of ATSPs. However, this two-step process can in fact be simplified into a single embedding step, as exemplified by zero embedding and global learnable embedding. Zero embedding is employed in the GAT-based MatPOENet (Pan et al., 2025) as one of the dual node embeddings. From a feasibility perspective, zero embedding can serve as an unbiased node embedding for GCN-based models. However, the representations of the first-layer encoder tend to degenerate with zero values, eventually causing most neurons to become inactive, as shown in Equations 16–19. In contrast, global learnable embedding effectively addresses all these issues: it introduces no bias, eliminates the need for the linear embedding step $\mathbb{R}^2 \to \mathbb{R}^d$, and fully leverages all learnable parameters.

$$u_{i,j}^1 = \mathbf{W}_1^1 e_{i,j}^0 + \mathbf{W}_2^1 x_i^0 + \mathbf{W}_3^1 x_j^0 \xrightarrow{x^0=\mathbf{0}_d} u_{i,j}^1 = \mathbf{W}_1^1 e_{i,j}^0, \tag{16}$$

$$v_{i,j}^1 = \mathbf{W}_4^1 x_i^0 + \mathbf{W}_5^1 x_j^0 \odot \sigma(u_{i,j}^1) \xrightarrow{x^0=\mathbf{0}_d} v_{i,j}^1 = \mathbf{0}_d, \tag{17}$$

$$e_{i,j}^1 = e_{i,j}^0 + \text{GELU}(\text{LN}(u_{i,j}^1)) \xrightarrow{x^0=\mathbf{0}_d} e_{i,j}^1 = e_{i,j}^0 + \text{GELU}(\text{LN}(\mathbf{W}_1^1 e_{i,j}^0)), \tag{18}$$

$$x_i^1 = x_i^0 + \text{GELU}(\text{LN}(\sum_{j=1}^{k} v_{i,j}^1)) \xrightarrow{x^0=\mathbf{0}_d} x_i^1 = \mathbf{0}_d. \tag{19}$$

## F  DISCUSSION ON NODE AGGREGATION

For node aggregation of GCNs, NeuroLKH (Xin et al., 2021) adopts a channel-dependent softmax scheme, while our UNE-GCN uses a channel-independent sigmoid scheme. The feature weight computations for these two schemes are given in Equation 20.

$$w_{i,j}^\ell = \begin{cases} \sigma(u_{i,j}^\ell), & \text{sigmoid}, \\ \exp(u_{i,j}^\ell) \oslash \sum_{j=1}^{\mathbf{F}_\ell} \exp(u_{i,j}^\ell), & \text{softmax}, \end{cases} \tag{20}$$

$$v_{i,j}^\ell = \mathbf{W}_4^\ell x_i^{\ell-1} + \begin{cases} \mathbf{W}_5^\ell x_j^{\ell-1} \odot w_{i,j}^\ell, & \text{if } j \leq \mathbf{F}_\ell, \\ \mathbf{0}_d, & \text{otherwise}, \end{cases} \tag{21}$$

where $\oslash$ and $\odot$ denote element-wise division and element-wise multiplication, respectively. The sigmoid scheme corresponds to a sum aggregation, whereas the softmax scheme corresponds to a weighted mean aggregation. A prior study by Xu et al. (2019) has shown that sum aggregation is more advantageous for GNN representations than mean aggregation, as it preserves richer neighborhood information and offers stronger discriminative capacity. Furthermore, our experiments demonstrate that sum aggregation with the sigmoid scheme is more effective for UNE-GCN in learning TSPs. As shown in Figure 11, node aggregation under the softmax scheme suffers from overfitting and struggles to converge, consistently maintaining a relatively high MR, and performs significantly worse than the sigmoid scheme.

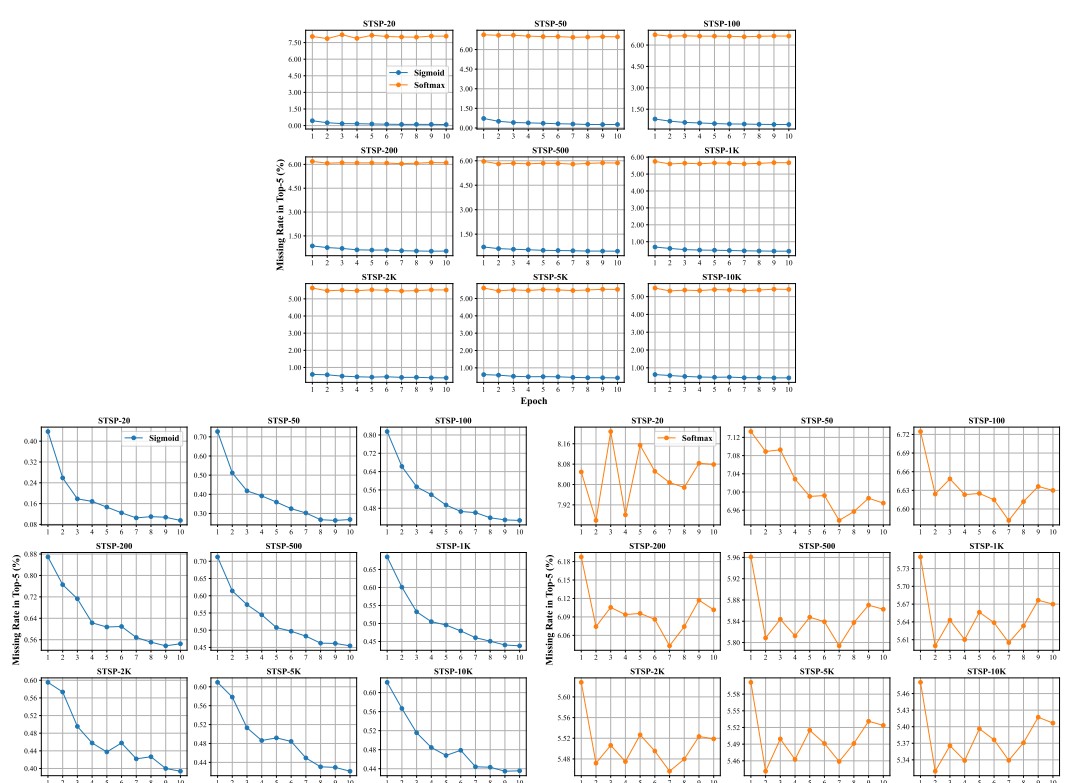

Figure 11: MR@5 (%) variation of node aggregation with sigmoid and softmax ($l = 12$).

# G  DISCUSSION ON EDGE NORMALIZATION

INViT (Fang et al., 2024) and RsGCN (Huang et al., 2025) introduce 2D-Uniform Scaling for edge lengths based on 2D coordinates. Specifically, from the perspective of subgraph $\mathcal{N}_i = \{x_i^j\}_{j=1}^k$ with respect to $x_i$, we first compute the maximum spans $\mathbf{P}_i$ and $\mathbf{Q}_i$ of $\mathcal{N}_i$ along the two axes, respectively:

$$\mathbf{P}_i = \max(\{x_{i,1}^j\}_{j=1}^k) - \min(\{x_{i,1}^j\}_{j=1}^k), \tag{22}$$

$$\mathbf{Q}_i = \max(\{x_{i,2}^j\}_{j=1}^k) - \min(\{x_{i,2}^j\}_{j=1}^k), \tag{23}$$

where $x_{i,1}^j$ and $x_{i,2}^j$ denote the coordinates of node $x_i^j$ along the first and second axes, respectively. Next, we compute the scaling factor $\mu_i$ and apply it to each row of the sparse distance matrix $\mathbf{E}$, obtaining the normalized distance matrix $\tilde{\mathbf{E}}$:

$$\mu_i = \frac{1}{\max(\mathbf{P}_i, \mathbf{Q}_i)}, \tag{24}$$

$$\tilde{\mathbf{E}}_i = \mu_i \cdot \mathbf{E}_i, \ \forall i \in \{1, 2, \ldots, n\}. \tag{25}$$

Figure 12 visually compares the effects of 2D-Uniform and Min-Max Scaling. 2D-Uniform Scaling preserves the exact two-dimensional proportion, but it relies on explicit coordinates and thus cannot be applied to edge normalization in ATSPs. Figure 13 illustrates the MR@5 variation of 2D-Uniform and Min-Max Scaling on STSPs. As can be observed, 2D-Uniform Scaling achieves better performance on the seen scales of STSP-20/50/100. When generalized to STSP-200 and larger scales, however, the performance gap becomes negligible, which corroborates that Min-Max Scaling as an edge normalization method can also effectively capture the relative relationships among edge lengths.

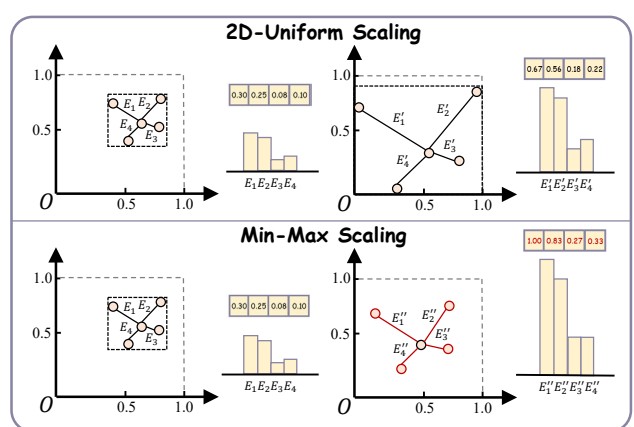

Figure 12: Comparison of 2D-Uniform and Min-Max Scaling.

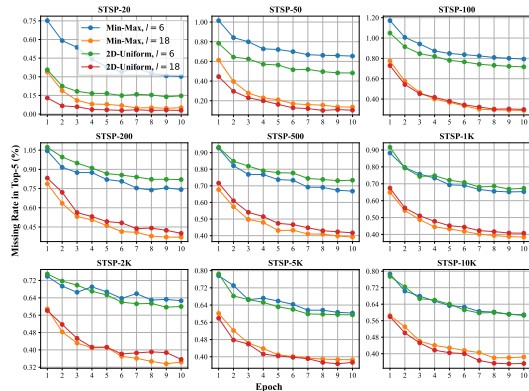

Figure 13: MR@5 (%) variation of edge normalization with 2D-Uniform and Min-Max Scaling.

## H    SCALABILITY OF INFERENCE

We perform single-batch inference with UNE-GCN under the default *float-32* precision, and record the time and GPU memory required to complete inference on a single STSP instance, as shown in Table 9. The scalability of GPU memory is visualized in Figure 14. It can be seen that UNE-GCN exhibits good linear scalability in memory, demonstrating its potential for solving massive-scale TSPs.

Table 9: Scalability of inference in time and memory.

| STSP- | | 100 | 500 | 1K | 5K | 10K | 50K | 100K | 500K |
|---|---|---|---|---|---|---|---|---|---|
| **Time (s)** | $l = 6$ | 0.01 | 0.01 | 0.01 | 0.01 | 0.01 | 0.08 | 0.26 | 1.70 |
| | $l = 12$ | 0.02 | 0.02 | 0.02 | 0.02 | 0.02 | 0.23 | 0.57 | 3.24 |
| | $l = 18$ | 0.02 | 0.02 | 0.02 | 0.02 | 0.02 | 0.39 | 0.88 | 4.82 |
| **Memory (GB)** | $l \in \{6, 12, 18\}$ | 0.03 | 0.09 | 0.16 | 0.74 | 1.47 | 7.27 | 14.53 | 72.54 |

## I    DUAL-ENCODER ARCHITECTURE

In prior work, dual-encoder architectures have been introduced to better handle the asymmetric structure of ATSPs. For example, MatNet (Kwon et al., 2021) employs a dual graph attentional encoder, while MatDIFFNet (Pan et al., 2025) adopts a dual graph convolutional encoder. We also

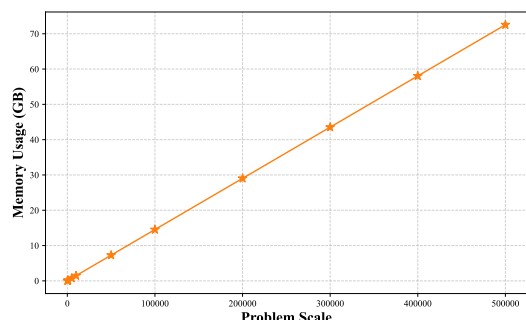

Figure 14: Linear scalability of GPU memory usage for UNE-GCN.

design a dual-encoder variant of UNE-GCN to verify its effect, and re-implement the network architecture following the procedure described in Section 4.4.

**Dual Input Embedding** We initialize two independent vectors $\mathbf{X}^{\mathbb{A}}$ and $\mathbf{X}^{\mathbb{B}}$ using Xavier initialization, which serve as the global embeddings of out-nodes $x^{\mathbb{A},0}$ and in-nodes $x^{\mathbb{B},0}$, respectively. The out-distance and in-distance matrices $\tilde{\mathbf{E}}^{\mathbb{A}}$ and $\tilde{\mathbf{E}}^{\mathbb{B}}$ are obtained as described in Appendix B. Note that in STSPs we have $\tilde{\mathbf{E}}^{\mathbb{A}} = \tilde{\mathbf{E}}^{\mathbb{B}}$, whereas this property does not hold in ATSPs. Subsequently, the embeddings of out-edges $e^{\mathbb{A},0}$ and in-edges $e^{\mathbb{B},0}$ are obtained through the following linear projection:

$\forall \alpha \in \{\mathbb{A}, \mathbb{B}\} :$

$$x_i^{\alpha,0} = \mathbf{X}^{\alpha}, \tag{26}$$

$$e_{i,j}^{\alpha,0} = \text{Linear}(\tilde{\mathbf{E}}_{i,j}^{\alpha}) : \mathbb{R}^1 \to \mathbb{R}^d, \tag{27}$$

**Dual Graph Encoder** For the intermediate edge and node features, we also apply dual encoding $(u^{\mathbb{A}/\mathbb{B}}, v^{\mathbb{A}/\mathbb{B}})$. Note that for the learnable matrices $\mathbf{W}_4$ and $\mathbf{W}_5$ associated with the intermediate node feature $v^{\mathbb{A}/\mathbb{B}}$, we separate them into $\mathbf{W}_4^{\mathbb{A}/\mathbb{B}}$ and $\mathbf{W}_5^{\mathbb{A}/\mathbb{B}}$. This design not only aligns with that of MatDIFFNet, but also enables differentiated processing of the intermediate features of out-nodes $v^{\mathbb{A}}$ and in-nodes $v^{\mathbb{B}}$. Finally, the refined edge features $e^{\mathbb{A}/\mathbb{B}}$ and node features $x^{\mathbb{A}/\mathbb{B}}$ are also represented in the dual manner.

$\forall (\alpha, \beta) \in \{(\mathbb{A}, \mathbb{B}), (\mathbb{B}, \mathbb{A})\} :$

$$u_{i,j}^{\alpha,\ell} = \mathbf{W}_1^{\ell} e_{i,j}^{\alpha,\ell-1} + \begin{cases} \mathbf{W}_2^{\ell} x_i^{\alpha,\ell-1} + \mathbf{W}_3^{\ell} x_j^{\beta,\ell-1}, & \text{if } j \leq \mathbf{F}_{\ell}, \\ \mathbf{0}_d, & \text{otherwise}, \end{cases} \tag{28}$$

$$v_{i,j}^{\alpha,\ell} = \mathbf{W}_4^{\alpha,\ell} x_i^{\alpha,\ell-1} + \begin{cases} \mathbf{W}_5^{\beta,\ell} x_j^{\beta,\ell-1} \odot \sigma(u_{i,j}^{\beta,\ell}), & \text{if } j \leq \mathbf{F}_{\ell}, \\ \mathbf{0}_d, & \text{otherwise}, \end{cases} \tag{29}$$

$\forall \alpha \in \{\mathbb{A}, \mathbb{B}\} :$

$$e_{i,j}^{\alpha,\ell} = e_{i,j}^{\alpha,\ell-1} + \text{GELU}(\text{LN}(u_{i,j}^{\alpha,\ell})), \tag{30}$$

$$x_i^{\alpha,\ell} = x_i^{\alpha,\ell-1} + \text{GELU}(\text{LN}(\sum_{j=1}^{k} v_{i,j}^{\alpha,\ell})), \tag{31}$$

**Dual Heatmap Decoder** Similarly, the dual logits $\mathbf{Y}^{\mathbb{A}/\mathbb{B}}$ and the dual heatmaps $\mathbf{H}^{\mathbb{A}/\mathbb{B}}$ are obtained as follows:

$\forall (\alpha, \beta) \in \{(\mathbb{A}, \mathbb{B}), (\mathbb{B}, \mathbb{A})\} :$

$$\tilde{\mathbf{Y}}_{i,j}^{\alpha} = \text{MLP}(\text{LN}(e_{i,j}^{\alpha,l} + x_i^{\alpha,l} + x_j^{\beta,l})), \tag{32}$$

$$\mathbf{Y}^{\alpha} = \text{Scatter}(\tilde{\mathbf{Y}}^{\alpha}, \mathcal{X}), \tag{33}$$

$$\mathbf{H}^{\alpha} = \sigma(\mathbf{Y}^{\alpha}), \tag{34}$$

**Dual Loss Function** The loss function also takes into account the dual logits $\mathbf{Y}^{\mathbb{A}/\mathbb{B}}$. Since $\mathbf{Y}^{\mathbb{B}}$ corresponds to the transposed distance matrix $\mathbf{D}^{\top}$, it has the opposite direction. Therefore, the indices of the label variable $t$ are reversed for $\mathbf{Y}^{\mathbb{B}}$, *i.e.*, $t_{i,j} \to t_{j,i}$.

$$\mathcal{L}^{\mathbb{AB}} = \frac{1}{2n} \cdot \sum_{i=1}^{n} \sum_{j=1}^{n} \left[ \log(1 + \exp(-t_{i,j} \cdot \mathbf{Y}_{i,j}^{\mathbb{A}})) + \log(1 + \exp(-t_{j,i} \cdot \mathbf{Y}_{i,j}^{\mathbb{B}})) \right], \quad (35)$$

Following the settings in the main text and the dual convolution in this section, we train dual UNE-GCN with $l = 6, 18$ for both STSPs and ATSPs. Based on the dual logits $\mathbf{Y}^{\mathbb{A}/\mathbb{B}}$, we also computed two separate MR@5, as shown in Figures 15 and 16. In the legends, **Single** and **Dual-A/B** denote the MR@5 computed from $\mathbf{Y}$ of UNE-GCN and $\mathbf{Y}^{\mathbb{A}/\mathbb{B}}$ of dual UNE-GCN, respectively. It can be observed that the dual convolution brings certain improvements for learning STSPs, but its benefit for ATSPs is less evident. Moreover, the dual convolution exhibits a logit imbalance issue on ATSPs: the quality of $\mathbf{Y}^{\mathbb{B}}$ is noticeably inferior to that of $\mathbf{Y}^{\mathbb{A}}$, even falling behind the single-convolution UNE-GCN. Furthermore, dual convolution doubles both the number of parameters and the computational cost, as shown in Table 10. Based on these results, we generally recommend using the single-convolution architecture, which offers a better balance between performance and cost.

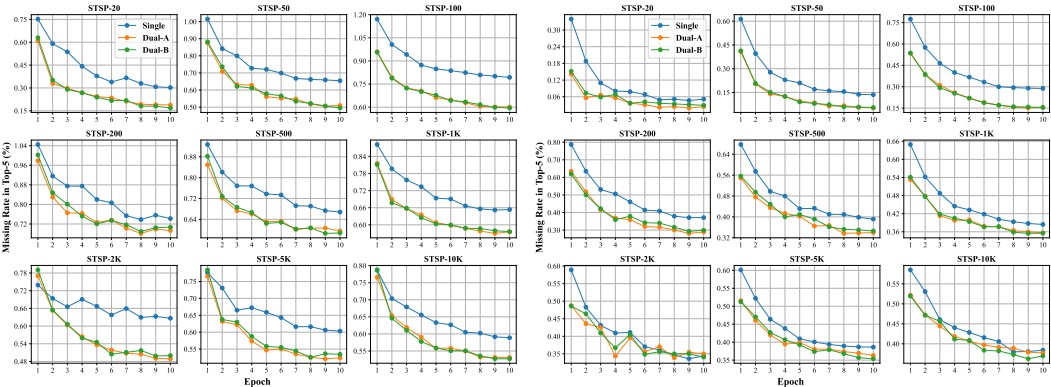

Figure 15: MR@5 (%) variation of dual convolution on STSPs (left: $l = 6$, right: $l = 18$).

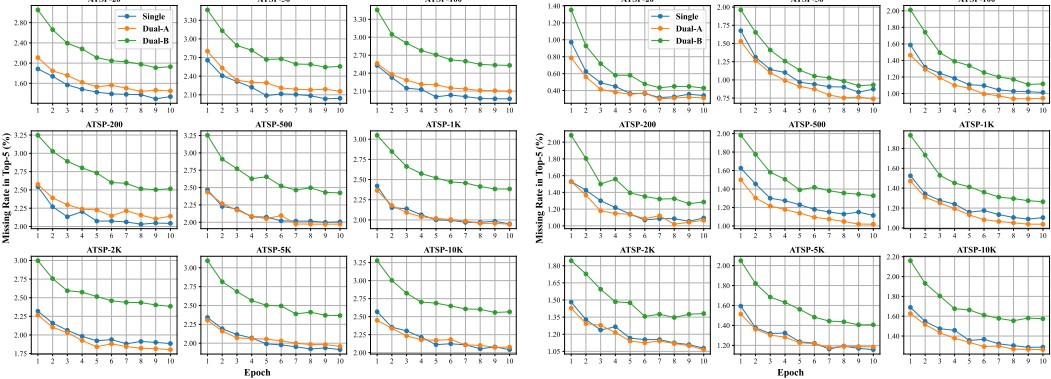

Figure 16: MR@5 (%) variation of dual convolution on ATSPs (left: $l = 6$, right: $l = 18$).

## J VISUAL EVALUATION OF HEATMAPS

For STSPs, we adopt the ordered heatmap proposed by Huang et al. (2025) to visually evaluate the quality of the heatmaps generated by 18-layer UNE-GCN and its ablation variants. The ordered heatmap is obtained by sorting according to the near-optimal tour. The more concentrated the heat values are around the neighboring cells of the main diagonal, the more indicative the heatmap is. We select the ordered heatmaps of identical large-scale STSP instances and crop a 100×100 local

Table 10: Training cost statistics of single and dual convolution with $l = 18$.

| Training set | Batch size | Convolution | #Parameters | Memory usage | Time per epoch |
|---|---|---|---|---|---|
| STSP-20/50/100 | 32 | Single | 1.215M | 6.22GB | 0.77h |
| | | Dual | 2.112M | 12.24GB | 2.20h |
| ATSP-20/50/100 | 32 | Single | 1.513M | 6.22GB | 1.00h |
| | | Dual | 2.112M | 12.25GB | 2.19h |

region at the same location, which are shown in Figures 17–21. Clearly, the heatmaps of variants $\sim$ w/o U-E, $\sim$ w/o N-E, and $\sim$ w/o U-N-E are of low quality and fail to provide effective indications.

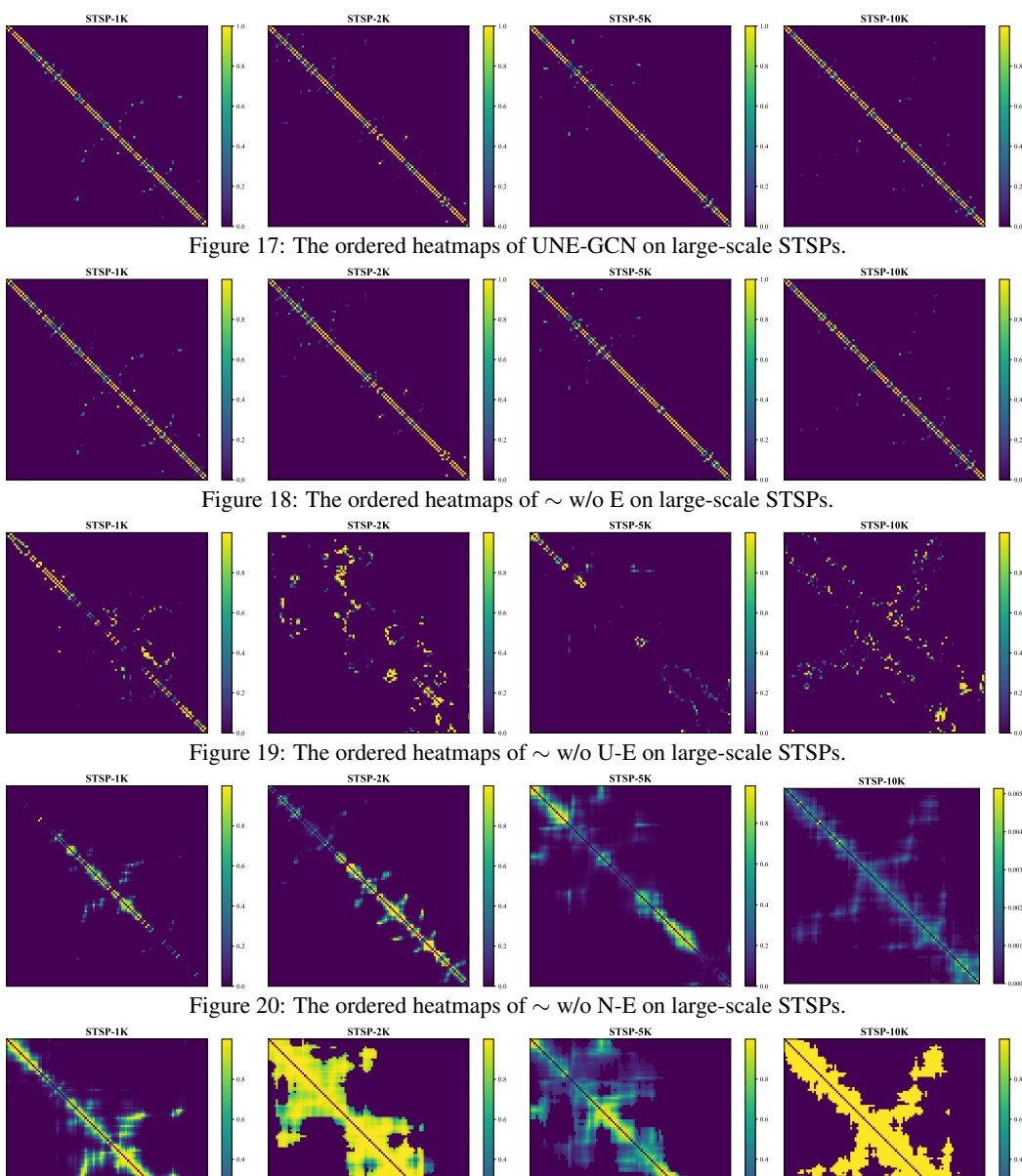

Figure 17: The ordered heatmaps of UNE-GCN on large-scale STSPs.

Figure 18: The ordered heatmaps of $\sim$ w/o E on large-scale STSPs.

Figure 19: The ordered heatmaps of $\sim$ w/o U-E on large-scale STSPs.

Figure 20: The ordered heatmaps of $\sim$ w/o N-E on large-scale STSPs.

Figure 21: The ordered heatmaps of $\sim$ w/o U-N-E on large-scale STSPs.

## K    RESULTS ON TSPLIB

The real-world test set from TSPLIB (Reinelt, 1991), contains 78 STSP instances with scales up to 18,512 and 27 ATSP instances with scales up to 443. The best known solutions are taken directly from TSPLIB. We set the number of trails in LKH-3 to $n$, equal to the problem scale, and the number of runs to 10. To investigate the cross-distribution generalization of NeuroLKH and UNE-GCN, we conduct experiments using NeuroLKH and a 6-layer UNE-GCN trained on uniformly distributed STSPs. Table 11 reports the solution metrics averaged over runs. It can be observed that UNE-GCN still has room for improvement in cross-distribution generalization, but it significantly outperforms NeuroLKH. NeuroLKH performs poorly on large-scale real-world STSP instances, especially beyond the 10K scale, where it falls into excessively long ineffective searches. Figure 22 presents three STSP instances in which UNE-GCN is less effective, due to the particular real-world distributions. In addition, Tables 12 and 13 present the solution metrics for each STSP and ATSP instance, where the optimal solution is shown below each instance name, and the **Trials** column reports the average number of trials required to obtain the optimum.

Table 11: Average solution metrics across different scale intervals on TSPLIB instances.

| Type | Scale | LKH-3 | | NeuroLKH | | UNE-GCN | |
|---|---|---|---|---|---|---|---|
| | | Gap(‰) | Time | Gap(‰) | Time | Gap(‰) | Time |
| STSP | <100 | 0.0000 | 0.01s | 0.0000 | 0.01s | 0.0000 | 0.01s |
| | [100, 1K) | 0.0568 | 0.24s | 1.1711 | 1.16s | 0.2918 | 1.90s |
| | [1K, 10K) | 0.3316 | 1.12m | 1.2751 | 14.20m | 0.7001 | 2.26m |
| | ≥10K | 0.0482 | 1.26h | 2.9637 | 21.39h | 0.0728 | 1.71h |
| | All | 0.1364 | 5.20m | 1.2279 | 1.44h | 0.3809 | 7.27m |
| ATSP | <50 | 0.0549 | 0.01s | — | | 1.6371 | 0.01s |
| | [50, 100) | 0.0000 | 0.01s | — | | 3.0787 | 0.01s |
| | [100, 200) | 0.0166 | 0.01s | — | | 0.2755 | 0.01s |
| | ≥200 | 0.0000 | 0.82s | — | | 4.9590 | 10.30s |
| | All | 0.0218 | 0.13s | — | | 1.9957 | 1.53s |

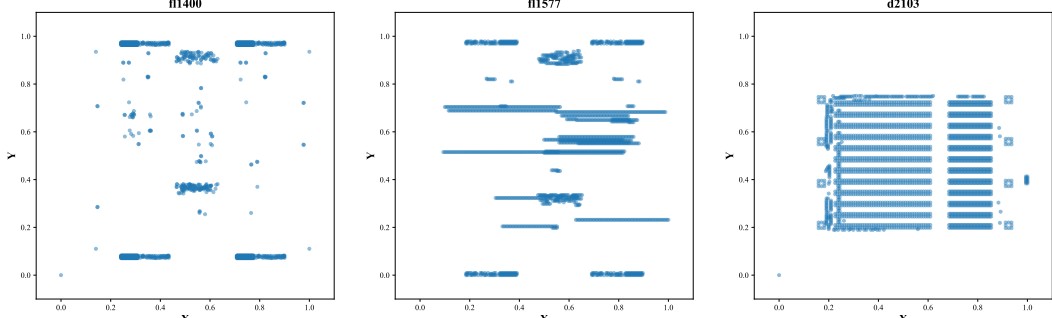

Figure 22: The real-world distributions of three STSP instances from TSPLIB.

## L    COMPLETE RESULTS OF THE SCALING LAWS

To explore the MR@5 lower bounds of the listed models, we select the lowest MR@5 achieved by each model across all scales and visualize the results in comparative heatmaps as Figure 23. Note that here $-\mathbf{Dm}$ is obtained by masking the diagonal entries of $\mathbf{D}$ to $+\infty$ and subsequently negating the entire matrix. $-\mathbf{Dm}$ serves as the heatmap for computing a MR@5 upper bound. The results show that UNE-GCN achieves a better MR@5 lower bound on large-scale STSPs and ATSPs. Although its MR@5 is slightly inferior to other ablation variants on the seen scales of 20/50/100, the solving complexity and cost of small-scale TSPs are relatively low. Moreover, the slight gap in MR@5 has little impact on post-search for small-scale TSPs. The complete results are shown in Figures 24–28. For the seen scales of 20/50/100, all models perform well. However, for unseen scales, the variants ∼ w/o E, ∼ w/o U-E, ∼ w/o N-E, and ∼ w/o U-N-E quickly fall into local optima

due to their narrow representations, making further training counterproductive. In contrast, UNE-GCN achieves stable convergence, demonstrating remarkable generalization and superior learning dynamics.

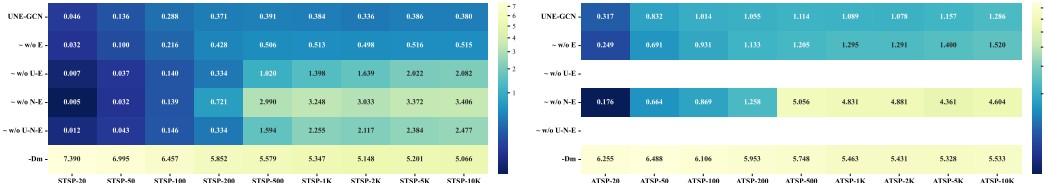

Figure 23: MR@5 (%) lower bounds of ablations.

Table 12: Solution metrics for STSP instances in TSPLIB.

| Method | Instance | Length | Time (s) | Trials | Instance | Length | Time (s) | Trials |
|---|---|---|---|---|---|---|---|---|
| LKH-3 | a280 | 2579.0 | 0.01 | 1 | berlin52 | 7542.0 | 0.01 | 0 |
| NeuroLKH | 2579 | 2579.0 | 0.01 | 1 | 7542 | 7542.0 | 0.01 | 1 |
| UNE-GCN | | 2579.0 | 0.01 | 1 | | 7542.0 | 0.01 | 1 |
| LKH-3 | bier127 | 118282.0 | 0.01 | 1 | brd14051 | 469401.0 | 4086.19 | 14051 |
| NeuroLKH | 118282 | 118313.0 | 0.08 | 127 | 469385 | 473022.2 | 45491.24 | 14051 |
| UNE-GCN | | 118282.0 | 0.02 | 1 | | 469408.8 | 5629.19 | 14051 |
| LKH-3 | ch130 | 6110.0 | 0.01 | 2 | ch150 | 6530.0 | 0.06 | 100 |
| NeuroLKH | 6110 | 6110.0 | 0.01 | 1 | 6528 | 6528.0 | 0.03 | 41 |
| UNE-GCN | | 6110.0 | 0.01 | 2 | | 6528.5 | 0.04 | 41 |
| LKH-3 | d198 | 15780.0 | 0.12 | 11 | d493 | 35003.4 | 2.85 | 393 |
| NeuroLKH | 15780 | 15879.6 | 0.33 | 198 | 35002 | 35044.8 | 2.87 | 461 |
| UNE-GCN | | 15803.7 | 0.91 | 197 | | 35002.0 | 0.47 | 50 |
| LKH-3 | d657 | 48913.0 | 2.50 | 657 | d1291 | 50814.5 | 16.91 | 755 |
| NeuroLKH | 48912 | 48913.6 | 1.88 | 657 | 50801 | 50879.2 | 55.12 | 1291 |
| UNE-GCN | | 48913.0 | 2.29 | 657 | | 50860.8 | 18.74 | 1291 |
| LKH-3 | d1655 | 62128.0 | 1.85 | 153 | d2103 | 80495.4 | 65.66 | 2103 |
| NeuroLKH | 62128 | 62130.0 | 6.06 | 635 | 80450 | 80460.3 | 69.65 | 1959 |
| UNE-GCN | | 62128.0 | 9.49 | 775 | | 80571.6 | 40.91 | 2103 |
| LKH-3 | d15112 | 1573166.7 | 5540.00 | 15112 | d18512 | 645255.9 | 8207.25 | 18512 |
| NeuroLKH | 1573084 | 1574932.3 | 18375.51 | 15112 | 645238 | 645954.9 | 21329.51 | 18512 |
| UNE-GCN | | 1573159.3 | 7072.53 | 15112 | | 645302.3 | 12914.63 | 18512 |
| LKH-3 | eil51 | 426.0 | 0.01 | 1 | eil76 | 538.0 | 0.01 | 1 |
| NeuroLKH | 426 | 426.0 | 0.01 | 1 | 538 | 538.0 | 0.01 | 1 |
| UNE-GCN | | 426.0 | 0.01 | 1 | | 538.0 | 0.01 | 1 |
| LKH-3 | eil101 | 629.0 | 0.01 | 1 | fl417 | 11866.1 | 1.50 | 72 |
| NeuroLKH | 629 | 629.0 | 0.01 | 1 | 11861 | 11969.7 | 6.14 | 417 |
| UNE-GCN | | 629.0 | 0.01 | 1 | | 11900.3 | 12.74 | 375 |
| LKH-3 | fl1400 | 20166.1 | 104.53 | 1400 | fl1577 | 22260.7 | 107.85 | 1480 |
| NeuroLKH | 20127 | 20248.0 | 3297.82 | 1400 | 22249 | 22254.8 | 114.58 | 1465 |
| UNE-GCN | | 20198.8 | 800.22 | 1400 | | 22411.4 | 111.83 | 1577 |
| LKH-3 | fl3795 | 28799.8 | 557.55 | 3413 | fnl4461 | 182566.5 | 57.07 | 1793 |
| NeuroLKH | 28772 | 29020.5 | 15510.99 | 3795 | 182566 | 182569.0 | 123.94 | 3677 |
| UNE-GCN | | 28772.0 | 1356.61 | 1467 | | 182568.7 | 90.42 | 3070 |
| LKH-3 | gil262 | 2378.0 | 0.14 | 42 | kroA100 | 21282.0 | 0.01 | 1 |
| NeuroLKH | 2378 | 2378.0 | 0.03 | 7 | 21282 | 21282.0 | 0.01 | 1 |
| UNE-GCN | | 2378.0 | 0.04 | 11 | | 21282.0 | 0.01 | 1 |

Table 12 – continued from previous page

| Method | Instance | Length | Time (s) | Trials | Instance | Length | Time (s) | Trials |
|---|---|---|---|---|---|---|---|---|
| LKH-3 | kroB100 | 22141.0 | 0.01 | 1 | kroC100 | 20749.0 | 0.01 | 1 |
| NeuroLKH | 22141 | 22141.0 | 0.01 | 6 | 20749 | 20749.0 | 0.01 | 1 |
| UNE-GCN | | 22141.0 | 0.01 | 2 | | 20749.0 | 0.01 | 1 |
| LKH-3 | kroD100 | 21294.0 | 0.01 | 1 | kroE100 | 22068.0 | 0.02 | 12 |
| NeuroLKH | 21294 | 21294.0 | 0.01 | 1 | 22068 | 22068.0 | 0.01 | 9 |
| UNE-GCN | | 21294.0 | 0.01 | 1 | | 22068.0 | 0.01 | 3 |
| LKH-3 | kroA150 | 26524.0 | 0.01 | 1 | kroB150 | 26130.8 | 0.07 | 80 |
| NeuroLKH | 26524 | 26524.0 | 0.01 | 1 | 26130 | 26130.4 | 0.04 | 84 |
| UNE-GCN | | 26524.0 | 0.01 | 1 | | 26131.2 | 0.11 | 104 |
| LKH-3 | kroA200 | 29368.0 | 0.02 | 1 | kroB200 | 29437.0 | 0.01 | 1 |
| NeuroLKH | 29368 | 29368.0 | 0.02 | 2 | 29437 | 29437.0 | 0.01 | 1 |
| UNE-GCN | | 29368.0 | 0.02 | 1 | | 29437.0 | 0.01 | 3 |
| LKH-3 | lin105 | 14379.0 | 0.01 | 1 | lin318 | 42080.0 | 0.26 | 209 |
| NeuroLKH | 14379 | 14379.0 | 0.01 | 1 | 42029 | 42051.8 | 0.22 | 101 |
| UNE-GCN | | 14379.0 | 0.01 | 1 | | 42043.7 | 0.40 | 260 |
| LKH-3 | linhp318 | 41345.0 | 0.02 | 6 | nrw1379 | 56638.6 | 4.22 | 775 |
| NeuroLKH | 41345 | 41345.0 | 0.02 | 8 | 56638 | 56638.7 | 6.87 | 889 |
| UNE-GCN | | 41345.0 | 0.01 | 1 | | 56638.7 | 9.26 | 847 |
| LKH-3 | p654 | 34643.0 | 0.74 | 22 | pcb442 | 50778.0 | 0.21 | 32 |
| NeuroLKH | 34643 | 35674.0 | 35.92 | 654 | 50778 | 50778.0 | 0.05 | 4 |
| UNE-GCN | | 34830.2 | 59.87 | 320 | | 50778.0 | 0.11 | 12 |
| LKH-3 | pcb1173 | 56893.0 | 1.47 | 418 | pcb3038 | 137716.0 | 63.32 | 1903 |
| NeuroLKH | 56892 | 56892.0 | 0.85 | 185 | 137694 | 137707.4 | 72.64 | 2162 |
| UNE-GCN | | 56892.0 | 1.58 | 232 | | 137704.8 | 62.21 | 1837 |
| LKH-3 | pr76 | 108159.0 | 0.01 | 1 | pr107 | 44303.0 | 0.01 | 1 |
| NeuroLKH | 108159 | 108159.0 | 0.01 | 1 | 44303 | 44303.0 | 0.02 | 1 |
| UNE-GCN | | 108159.0 | 0.01 | 1 | | 44303.0 | 0.21 | 9 |
| LKH-3 | pr124 | 59030.0 | 0.01 | 1 | pr136 | 96772.0 | 0.02 | 1 |
| NeuroLKH | 59030 | 59030.0 | 0.01 | 1 | 96772 | 96772.0 | 0.03 | 10 |
| UNE-GCN | | 59066.8 | 0.04 | 99 | | 96772.0 | 0.06 | 36 |
| LKH-3 | pr144 | 58559.6 | 0.01 | 46 | pr152 | 73682.0 | 0.04 | 35 |
| NeuroLKH | 58537 | 58552.9 | 0.09 | 73 | 73682 | 73838.9 | 0.08 | 125 |
| UNE-GCN | | 58558.2 | 0.08 | 76 | | 73682.0 | 0.14 | 46 |
| LKH-3 | pr226 | 80369.0 | 0.01 | 1 | pr264 | 49135.0 | 0.05 | 4 |
| NeuroLKH | 80369 | 80398.1 | 0.18 | 206 | 49135 | 49135.0 | 0.17 | 40 |
| UNE-GCN | | 80419.0 | 0.59 | 206 | | 49135.0 | 0.07 | 2 |
| LKH-3 | pr299 | 48191.0 | 0.12 | 8 | pr439 | 107217.0 | 0.09 | 22 |
| NeuroLKH | 48191 | 48191.0 | 0.08 | 9 | 107217 | 107245.8 | 0.37 | 311 |
| UNE-GCN | | 48191.0 | 0.13 | 6 | | 107217.0 | 0.11 | 22 |
| LKH-3 | pr1002 | 259045.0 | 0.58 | 312 | pr2392 | 378032.0 | 3.03 | 190 |
| NeuroLKH | 259045 | 259045.3 | 0.56 | 306 | 378032 | 378032.0 | 4.42 | 299 |
| UNE-GCN | | 259128.3 | 1.90 | 497 | | 378032.0 | 5.04 | 200 |
| LKH-3 | rat99 | 1211.0 | 0.01 | 1 | rat195 | 2323.0 | 0.03 | 2 |
| NeuroLKH | 1211 | 1211.0 | 0.01 | 1 | 2323 | 2323.0 | 0.03 | 4 |
| UNE-GCN | | 1211.0 | 0.01 | 1 | | 2323.0 | 0.03 | 3 |
| LKH-3 | rat575 | 6773.1 | 0.67 | 184 | rat783 | 8806.0 | 0.08 | 4 |
| NeuroLKH | 6773 | 6773.1 | 0.35 | 136 | 8806 | 8806.0 | 0.09 | 30 |
| UNE-GCN | | 6774.2 | 2.04 | 502 | | 8806.0 | 0.07 | 12 |
| LKH-3 | rd100 | 7910.0 | 0.01 | 1 | rd400 | 15281.0 | 0.05 | 8 |
| NeuroLKH | 7910 | 7910.0 | 0.01 | 1 | 15281 | 15281.0 | 0.03 | 7 |
| UNE-GCN | | 7910.0 | 0.01 | 1 | | 15281.0 | 0.05 | 9 |

Table 12 – continued from previous page

| Method | Instance | Length | Time (s) | Trials | Instance | Length | Time (s) | Trials |
|---|---|---|---|---|---|---|---|---|
| LKH-3 | rl1304 | 253085.9 | 3.30 | 660 | rl1323 | 270241.6 | 5.06 | 1099 |
| NeuroLKH | 252948 | 253771.9 | 7.07 | 1304 | 270199 | 270217.9 | 6.85 | 1062 |
| UNE-GCN | | 252948.0 | 0.52 | 20 | | 270231.6 | 4.21 | 1323 |
| LKH-3 | rl1889 | 316625.5 | 13.00 | 1280 | rl5915 | 565633.9 | 214.24 | 5915 |
| NeuroLKH | 316536 | 316905.4 | 48.50 | 1889 | 565530 | 566725.9 | 337.57 | 5915 |
| UNE-GCN | | 316925.1 | 38.37 | 1789 | | 565668.1 | 434.61 | 5915 |
| LKH-3 | rl5934 | 556546.1 | 295.97 | 5934 | rl11849 | 923367.2 | 1776.17 | 10502 |
| NeuroLKH | 556045 | 558081.0 | 441.40 | 5934 | 923288 | 924355.7 | 10707.82 | 11849 |
| UNE-GCN | | 556052.4 | 174.61 | 2757 | | 923413.6 | 1901.99 | 10136 |
| LKH-3 | st70 | 675.0 | 0.01 | 1 | ts225 | 126643.0 | 0.02 | 1 |
| NeuroLKH | 675 | 675.0 | 0.01 | 1 | 126643 | 126643.0 | 0.01 | 1 |
| UNE-GCN | | 675.0 | 0.01 | 1 | | 126643.0 | 0.02 | 1 |
| LKH-3 | tsp225 | 3916.0 | 0.02 | 1 | u159 | 42080.0 | 0.01 | 1 |
| NeuroLKH | 3916 | 3916.0 | 0.02 | 1 | 42080 | 42080.0 | 0.01 | 1 |
| UNE-GCN | | 3916.0 | 0.03 | 1 | | 42080.0 | 0.01 | 1 |
| LKH-3 | u574 | 36905.0 | 0.23 | 59 | u724 | 41910.0 | 0.44 | 59 |
| NeuroLKH | 36905 | 36905.0 | 0.14 | 15 | 41910 | 41910.0 | 0.62 | 124 |
| UNE-GCN | | 36906.8 | 0.29 | 73 | | 41910.0 | 0.54 | 74 |
| LKH-3 | u1060 | 224112.9 | 20.86 | 831 | u1432 | 152970.0 | 0.62 | 18 |
| NeuroLKH | 224094 | 224306.1 | 15.59 | 935 | 152970 | 152970.0 | 0.61 | 20 |
| UNE-GCN | | 224097.6 | 5.35 | 382 | | 152970.0 | 1.18 | 14 |
| LKH-3 | u1817 | 57239.3 | 28.91 | 1817 | u2152 | 64282.1 | 31.83 | 1632 |
| NeuroLKH | 57201 | 57240.1 | 27.60 | 1817 | 64253 | 64260.9 | 24.96 | 1074 |
| UNE-GCN | | 57249.8 | 44.51 | 1817 | | 64261.0 | 23.30 | 1019 |
| LKH-3 | u2319 | 234256.0 | 0.80 | 3 | usa13509 | 19983672.4 | 3120.32 | 13509 |
| NeuroLKH | 234256 | 234256.0 | 0.74 | 5 | 19982859 | 20055339.0 | 289087.57 | 13509 |
| UNE-GCN | | 234256.0 | 0.70 | 3 | | 19983449.7 | 3172.30 | 13509 |
| LKH-3 | vm1084 | 239339.8 | 7.72 | 853 | vm1748 | 336572.0 | 8.58 | 902 |
| NeuroLKH | 239297 | 239351.7 | 5.59 | 1084 | 336556 | 337099.4 | 274.36 | 1748 |
| UNE-GCN | | 239349.6 | 6.47 | 976 | | 336556.0 | 10.05 | 659 |

Table 13: Solution metrics for ATSP instances in TSPLIB.

| Method | Instance | Length | Time (s) | Trials | Instance | Length | Time (s) | Trials |
|--------|----------|--------|----------|--------|----------|--------|----------|--------|
| LKH-3 | br17 | 39.0 | 0.01 | 1 | ft53 | 6905.0 | 0.01 | 1 |
| UNE-GCN | 39 | 39.0 | 0.01 | 1 | 6905 | 6905.0 | 0.01 | 42 |
| LKH-3 | ft70 | 38673.0 | 0.01 | 4 | ftv33 | 1286.0 | 0.01 | 1 |
| UNE-GCN | 38673 | 39091.3 | 0.06 | 140 | 1286 | 1286.0 | 0.01 | 1 |
| LKH-3 | ftv35 | 1473.0 | 0.01 | 15 | ftv38 | 1530.4 | 0.01 | 28 |
| UNE-GCN | 1473 | 1474.0 | 0.01 | 53 | 1530 | 1531.8 | 0.01 | 74 |
| LKH-3 | ftv44 | 1613.0 | 0.01 | 2 | ftv47 | 1776.0 | 0.01 | 15 |
| UNE-GCN | 1613 | 1613.0 | 0.01 | 3 | 1776 | 1776.0 | 0.01 | 4 |
| LKH-3 | ftv55 | 1608.0 | 0.01 | 2 | ftv64 | 1839.0 | 0.01 | 1 |
| UNE-GCN | 1608 | 1608.0 | 0.01 | 1 | 1839 | 1839.0 | 0.01 | 7 |
| LKH-3 | ftv70 | 1950.0 | 0.01 | 1 | ftv90 | 1579.0 | 0.01 | 5 |
| UNE-GCN | 1950 | 1950.0 | 0.01 | 2 | 1579 | 1583.8 | 0.02 | 165 |
| LKH-3 | ftv100 | 1788.0 | 0.01 | 17 | ftv110 | 1958.0 | 0.01 | 6 |
| UNE-GCN | 1788 | 1792.2 | 0.04 | 164 | 1958 | 1961.6 | 0.03 | 152 |
| LKH-3 | ftv120 | 2166.0 | 0.02 | 37 | ftv130 | 2307.0 | 0.01 | 4 |
| UNE-GCN | 2166 | 2166.0 | 0.01 | 18 | 2307 | 2307.0 | 0.01 | 22 |
| LKH-3 | ftv140 | 2420.0 | 0.01 | 3 | ftv150 | 2611.0 | 0.01 | 2 |
| UNE-GCN | 2420 | 2422.4 | 0.04 | 167 | 2611 | 2611.0 | 0.01 | 1 |
| LKH-3 | ftv160 | 2683.4 | 0.05 | 85 | ftv170 | 2755.0 | 0.01 | 7 |
| UNE-GCN | 2683 | 2683.0 | 0.01 | 7 | 2755 | 2755.0 | 0.03 | 51 |
| LKH-3 | kro124p | 36230.0 | 0.01 | 2 | p43 | 5621.0 | 0.04 | 86 |
| UNE-GCN | 36230 | 36231.1 | 0.01 | 46 | 5620 | 5626.9 | 0.04 | 86 |
| LKH-3 | rbg323 | 1326.0 | 0.73 | 139 | rbg358 | 1163.0 | 1.34 | 178 |
| UNE-GCN | 1326 | 1331.5 | 12.10 | 646 | 1163 | 1170.1 | 15.04 | 716 |
| LKH-3 | rbg403 | 2465.0 | 0.56 | 35 | rbg443 | 2720.0 | 0.66 | 20 |
| UNE-GCN | 2465 | 2467.6 | 6.38 | 704 | 2720 | 2721.8 | 4.40 | 685 |
| LKH-3 | ry48p | 14422.0 | 0.01 | 1 | | | | |
| UNE-GCN | 14422 | 14422.0 | 0.01 | 1 | | | | |

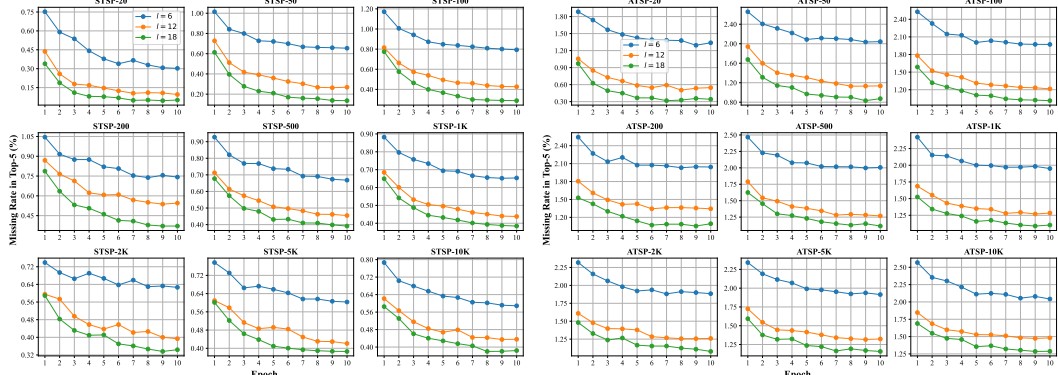

Figure 24: Scaling Laws for UNE-GCN on STSPs and ATSPs.

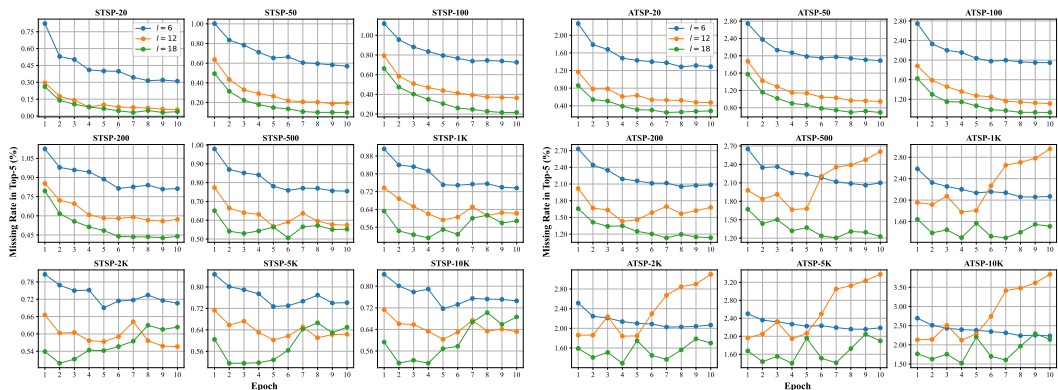

Figure 25: Scaling Laws for $\sim$ w/o E on STSPs and ATSPs.

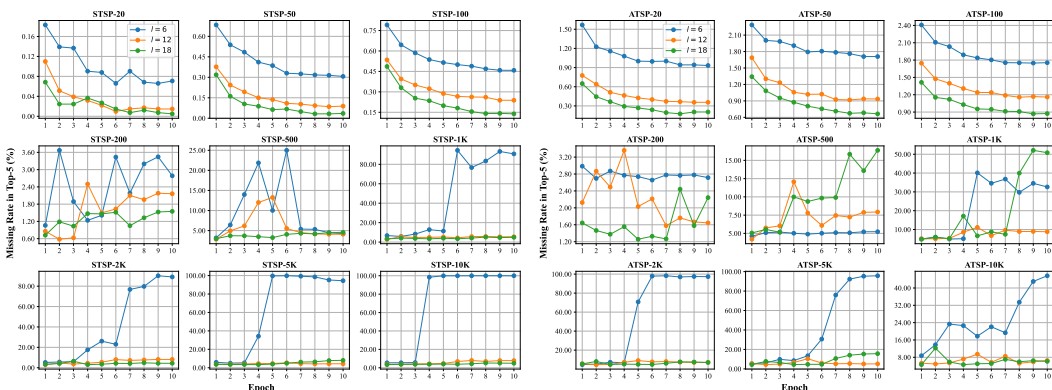

Figure 26: Scaling Laws for $\sim$ w/o N-E on STSPs and ATSPs.

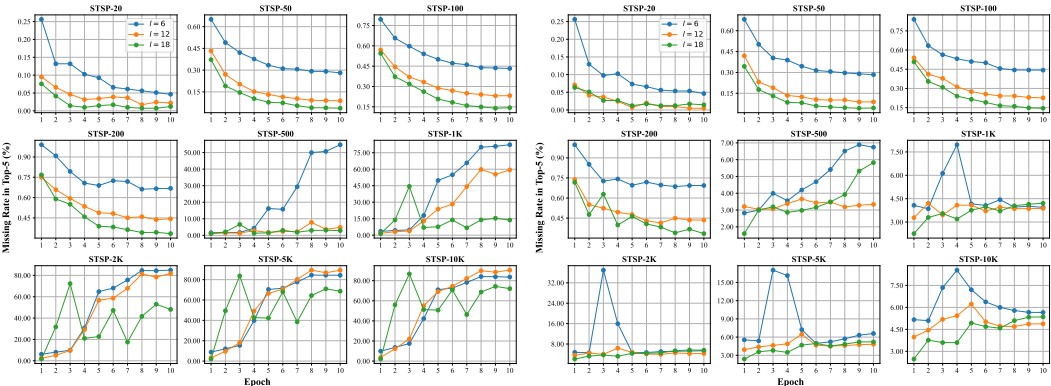

Figure 27: Scaling Laws for $\sim$ w/o U-E on STSPs.     Figure 28: Scaling Laws for $\sim$ w/o U-N-E on STSPs.

## M    LIMITATIONS AND FUTURE WORK

Although our UNE-GCN shows promising performance in cross-scale generalization, its generalization across different distributions (*e.g.*, uniform distribution and real-world distribution) still requires further investigation and improvement. Future work should focus on uncovering generalization patterns from general TSPs of diverse distributions, rather than merely increasing the diversity of training data distributions. Moreover, the design philosophy of U-N-E remains to be applied to different model backbones (*e.g.*, GAT, Transformer) and to a broader range of routing problems (*e.g.*, k-TSP, mTSP, CVRP).

# N ADDITIONAL MATERIAL FOR REBUTTAL

## N.1 ARCHITECTURAL COMPARISON OF GCNS

First, we would like to emphasize that the GCN-based baselines mentioned in our paper–Att-GCN (Fu et al., 2021), DIFUSCO (Sun & Yang, 2023), Fast T2T (Li et al., 2024), RsGCN (Huang et al., 2025), and NeuroLKH (Xin et al., 2021)–use standard Gated GCN (Joshi et al., 2019) or its slight variants as encoders. In fact, the standard Gated GCN is essentially equivalent to our ablation variant $\sim$ w/o U-N-E. We apologize for not making this explicit in the main text, which may have caused some confusion. Table 14 clearly shows the component composition of these GCN-based baselines and their correspondence to the ablation variants in our main text. Furthermore, the table also provides evidence for the novelty that distinguishes our work from prior methods.

Our systematic study of UNE-GCN and its ablation variants ($\sim$ w/o E, $\sim$ w/o N-E, $\sim$ w/o U-E, and $\sim$ w/o U-N-E) is essentially a systematic rethinking of prior GCN encoders. Here, LKH-3 serves primarily a validation role, while MR@5 is sufficient to clearly pre-evalute the model's generalization performance. In our ablation experiments, we compare GCN encoders individually under fair and consistent conditions, thereby eliminating the influence of decoders, training data, loss function, and other factors.

Table 14: Architectural comparison of GCN-based methods.

| Model | Equivalent Variant | Input Unification | Edge Normalization | Aggregation Enhancement |
|---|---|---|---|---|
| Att-GCN | $\sim$ w/o U-N-E | ✗ | ✗ | ✗ |
| NeuroLKH | $\sim$ w/o U-N-E | ✗ | ✗ | ✗ |
| DIFUSCO | $\sim$ w/o U-N-E | ✗ | ✗ | ✗ |
| Fast T2T | $\sim$ w/o U-N-E | ✗ | ✗ | ✗ |
| RsGCN | $\sim$ w/o U-E | ✗ | ✓ | ✗ |
| UNE-GCN | — | ✓ | ✓ | ✓ |

## N.2 RESULTS WITHOUT RELIANCE ON LKH-3

Here we clarify an empirical observation from our experiments: when a heatmap performs well in guiding greedy search, it typically also performs well when guiding LKH-3. In the following, we provide additional results based on greedy search (with 2-Opt).

We use greedy search (G) to obtain **one** solution and apply 2-Opt for further improvement, which is aligned with the standard setup of heatmap-based methods. For comparison, we reproduced the state-of-the-art GCN-based method NeuroLKH and Fast T2T. To ensure fairness and accurately assess generalization, we used the weights of Fast T2T trained on 1,502K STSP-100 instances. Additionally, our 2-Opt implementation is adopted from Fast T2T, with the maximum number of 2-Opt iterations uniformly set to 5K.

Table 15 reports the optimal gaps (%) of different methods and variants. As shown, when trained solely on STSP-100, Fast T2T exhibits limited generalization ability on large-scale STSPs. NeuroLKH, whose training set covers node sizes from 100 to 500, shows a slight advantage on STSP-1K and STSP-2K. However, its performance also degrades when the scale increases to STSP-5K and STSP-10K. In other words, the improvement of NeuroLKH largely stems from the increased node scales in its training data rather than from the efficiency of its GCN architecture.

It is worth noting that, in terms of parameter count, UNE-GCN (**1.215M**) has only around one-third of NeuroLKH (3.553M) and one-fourth of Fast T2T (5.334M). Because NeuroLKH and Fast T2T serve as advanced GCN-based solvers, the resulting comparisons are representative and meaningful.

Table 15: Results on large-scale STSPs with greedy search and 2-Opt.

| Method | STSP-1K | | STSP-2K | | STSP-5K | | STSP-10K | |
| | G | G + 2-Opt | G | G + 2-Opt | G | G + 2-Opt | G | G + 2-Opt |
|---|---|---|---|---|---|---|---|---|
| ∼ w/o U-N-E | 23.9952 | 4.9385 | 24.3169 | 5.4521 | 23.5369 | 5.4831 | 23.0728 | 5.4483 |
| ∼ w/o U-E | 75.7699 | 6.5615 | 235.2008 | 8.9647 | 311.8848 | 10.1151 | 324.9831 | 27.7476 |
| ∼ w/o N-E | 24.5721 | 5.1382 | 23.4649 | 5.0482 | 23.4917 | 6.0011 | 23.4764 | 5.4342 |
| ∼ w/o E | 18.5359 | 3.1360 | 17.6326 | 3.4647 | 19.3529 | 3.5568 | 19.6614 | 3.5837 |
| Fast T2T | 27.6901 | 5.4985 | 44.1805 | 7.4812 | 74.1795 | 9.4983 | 101.6159 | 10.1587 |
| NeuroLKH | 13.1563 | 1.8764 | 15.6162 | 2.3832 | 19.6152 | 3.4222 | 22.8993 | 4.4584 |
| UNE-GCN | 18.0433 | 2.9071 | 18.1637 | 2.9841 | 18.2597 | 3.2155 | 17.8744 | 3.1708 |

## N.3 RESULTS ON TMAT-CLASS ATSPs

For fairness, we include UniCO's (Pan et al., 2025) GCN-based method, MatDIFFNet, as a baseline. We use test sets publicly released in the UniCO repository, containing 2,500 ATSP-50 and 2,500 ATSP-100 instances in the TMAT class. Meanwhile, the weights for MatDIFFNet are taken from the checkpoints released by UniCO. Both UNE-GCN and MatDIFFNet adopt greedy search followed by 2-Opt as the post-search strategy. In addition, greedy search is restricted to sampling a single solution, and the 2-Opt implementation is adopted from UniCO.

As shown in Table 16, our UNE-GCN uses only about one-sixth of the parameters of MatDIFFNet, and requires merely one-tenth of its training epochs. Despite being significantly more lightweight, as shown in Table 17, UNE-GCN outperforms MatDIFFNet on both ATSP-50 and ATSP-100, while also achieving shorter inference time. These results collectively demonstrate that UNE-GCN achieves better performance with a much more lightweight architecture.

Table 16: Comparison with MatDIFFNet on training efficiency.

| Method | #Params | #Training Instances | #Training Epochs |
|---|---|---|---|
| MatDIFFNet | 6.986M | 1.28M | 100 |
| ∼ w/o N | 1.215M | 1.00M | 10 |
| UNE-GCN | 1.215M | 1.00M | 10 |

Table 17: Comparison with MatDIFFNet on TMAT-class ATSPs.

| Method | Search | ATSP-50 | | ATSP-100 | |
| | | Length | Time | Length | Time |
|---|---|---|---|---|---|
| MatDIFFNet | G | 6.6611 | 0.47s | 12.5671 | 0.43s |
| | G + 2-Opt | 3.2918 | 0.48s | 4.3353 | 0.43s |
| ∼ w/o N | G | 3.6545 | 0.03s | 4.6009 | 0.04s |
| | G + 2-Opt | 3.1687 | 0.03s | 3.9669 | 0.05s |
| UNE-GCN | G | 3.4989 | 0.03s | 4.5056 | 0.04s |
| | G + 2-Opt | 3.0883 | 0.03s | 3.9429 | 0.05s |

