# OpenReview forum: "Rethinking GCNs for the Traveling Salesman Problem: Are We Encoding Effectively?"
_ICLR.cc/2026/Conference — Submitted to ICLR 2026_

### Official Review · Reviewer_N9Zi · 2025-10-17

**Soundness:** 2
**Presentation:** 2
**Contribution:** 1
**Rating:** 2
**Confidence:** 4

**Summary:**

This paper re-evaluates Graph Convolutional Network (GCN) design for solving TSP, addressing limitations of poor generalization, overfitting, and inability to handle asymmetric TSP (ATSP) in existing solvers. It proposes UNE-GCN, a framework integrating three encoding strategies: Input Unification, Edge Normalization, and Aggregation Enhancement. UNE-GCN is tested in a two-stage pipeline with LKH-3, showing potential towards a unified GCN design for STSP/ATSP and insights into scalable TSP representation learning.

**Strengths:**

1. The paper focuses on the gap in prior GCN-based TSP solvers (lack of unified STSP/ATSP handling and poor cross-scale generalization), which is important for practical TSP solving by neural methods.
2. The proposed three encoding strategies are described with clarity. And the layer-wise expanding views are relatively novel adaptations to TSP's structural properties.

**Weaknesses:**

In general, I appreciate the research targets that have motivated the authors as well as the substantial works accomplished and reported thus far in their submission. However, there exist major concerns regarding narrative clarity, technical contribution, and the empirical evaluations.

**1. The novelty and technical contribution are limited.**
- The k-NN sparsification, learnable node embeddings, and min-max normalization for edge weights seem quite straightforward, already mentioned, or widely applied, which might appear as supplementary introduction or background setting in much of the literature rather than specific components as a primary contribution. The main contribution of this work may lie in proposing a new GCN backbone architecture tailored for TSP solving with a multi-view message-passing mechanism (Sec. 4.3-4.4). I suggest the authors emphasize this part with more explicit comparisons against previous GCNs. It is difficult to identify what's innovative about this work from the current version.
- As a submission claiming to address flaws in neural architectures, "*UNE-GCN can guide LKH-3 to perform more efficient searches, and the two-stage framework of UNE-GCN + LKH-3 achieves superior solutions with less search time on both symmetric and asymmetric TSPs*" is not supposed to be sufficient as a summarized outcome (detailed in point 3 below).

**2. The clarity and readability need to be improved.**
- The paper organization is not quite reasonable. Substantial critical information is piled in the appendix, e.g, the comparison with neural baselines (Appendix C), distinctive discussions (Appendix E/F/G), etc., while explanations for some straightforward tricks take up much space in Sec. 4.

**3. The experimental evaluations are limited.**
- Most importantly, the main attention should be on demonstrating the advantage of the proposed enhanced GCN backbone against other neural encoders that produce similar heatmaps with subpar quality.
    - So the authors need to provide results where solution tours are ***greedily constructed* under the guidance of the heatmaps predicted by baselines with different backbone models, without any post-inference improving techniques (e.g., LKH, MCTS, etc.)**
    - Currently, the main empirical comparison centers on ablation variants and against LKH, which is somewhat too narrow and deviates from the main targets. I do not see the point involving LKH as a stage-2 partner, because LKH-3 itself is well-known to be powerful enough and shall disguise the (minor) quality divergence of the raw neural output (e.g., the heatmap) by different baseline networks.
    - I.e., Table 6 and Table 7 should be placed in the main context with fairer comparisons. MCTS, 2-Opt, and LKH-3 have no comparability, and it does not make sense to show that UNE-GCN + LKH3 outperforms methods like DIFUSCO+MCTS or even pure neural approaches like the Transformer-based ones. **A possible and mainstream assessment might be as simple and clear as: 1) UNE-GCN + greedy > GCN + greedy; and 2) UNE-GCN + greedy > baseline (Att-GCN/DIFUSCO/T2T/...) + greedy.** Of course, you are still encouraged to keep comparisons like UNE-GCN + LKH > baselines + LKH, and so on.
- The evaluation protocol and test datasets are not standard.
    - Generalization is important, and so are the in-distribution parts. I wish to see ***clear and comparable*** results that the models are trained and tested on uniform (S)TSP instances that have been conventionally adopted by numerous works (e.g., Att-GCN, DIMES, DIFUSCO, Fast-T2T, etc. 1280 instances for TSP-50/100, 128 for TSP-500, etc.), with clear lengths, gaps, and solving time reported.
    - Could you please provide results on ATSP-[20/50/100/200/500] instances that are generated in a conventional TMAT class (by MatNet, UniCO, GOAL, etc) to ensure the best ***comparability***? Note you may still keep the results on test sets that are perturbed from TSP ones as cross-distribution generalization.
- Evaluating a GCN backbone only on TSP is somewhat limited. Consider testing on at least more similar graph-based combinatorial tasks is better, e.g., CVRP, MIS, etc.

**Questions:**

For clarity and convenient interaction, please refer to Weaknesses for all concerns.

---

> ### Author Response · Authors · 2025-11-17
>
> ## **Important Clarifications**
>
> First, we would like to emphasize that the GCN-based baselines mentioned in our paper—Att-GCN, DIFUSCO, Fast T2T, RsGCN, and NeuroLKH—use standard Gated GCN or its slight variants as encoders. In fact, the standard Gated GCN is essentially equivalent to our ablation variant ~ w/o U-N-E. We apologize for not making this explicit in the main text, which may have caused some confusion. The table below clearly shows the component composition of these GCN-based baselines and their correspondence to the ablation variants in our paper. Furthermore, this table also provides evidence for the novelty that distinguishes our work from prior methods.
>
> Our systematic study of UNE-GCN and its ablation variants (~ w/o E, ~ w/o N-E, ~ w/o U-E, and ~ w/o U-N-E) is essentially a systematic rethinking of prior GCN encoders. **Here, LKH serves primarily a validation role, while MR@5 is sufficient to clearly pre-evalute the model's generalization performance.** In contrast, comparisons with other baselines (e.g., Transformer-based methods) are less important. In our ablation experiments, we compare GCN encoders individually under fair and consistent conditions, thereby eliminating the influence of decoders, training data, loss function, and other factors.
>
> |  Model   | Corresponding Variant | Input Unification | Edge Normalization | Aggregation Enhancement |
> | :------: | :-------------------: | :---------------: | :----------------: | :---------------------: |
> | Att-GCN  |      ~ w/o U-N-E      |         ❌         |         ❌          |            ❌            |
> | NeuroLKH |      ~ w/o U-N-E      |         ❌         |         ❌          |            ❌            |
> | DIFUSCO  |      ~ w/o U-N-E      |         ❌         |         ❌          |            ❌            |
> | Fast T2T |      ~ w/o U-N-E      |         ❌         |         ❌          |            ❌            |
> |  RsGCN   |       ~ w/o U-E       |         ❌         |         ✅          |            ❌            |
> | UNE-GCN  |                       |         ✅         |         ✅          |            ✅            |
>
> ------
>
> ## **Q1. The Novelty and Technical Contribution**
>
> As noted in **Important Clarifications**, we have conducted a thorough comparison of the architectural distinctions between UNE-GCN and the GCN-based encoder baselines. To the best of our knowledge, our UNE-GCN is the first GCN-based TSP solver that jointly integrates **Input Unification**, **Edge Normalization**, and **Aggregation Enhancement**. If you are aware of more advanced GCN-based baselines, we would be happy to include them for comparison.
>
> Importantly, in terms of **cross-scale generalization**, UNE-GCN significantly outperforms all the GCN-based baselines discussed. The additional experiments provided in our later responses (**Q3**) further support this conclusion.
>
> ------
>
> ## **Q2. The Clarity and Readability**
>
> Thank you very much for pointing this out. We agree that the structural organization of our paper can be further improved. We will carefully incorporate your suggestions and enhance the clarity and readability in the subsequent versions.

---

> ### Author Response · Authors · 2025-11-17
>
> ## **Q3. The Experimental Evaluations**
>
> ### **(1) Results with Greedy Search**
>
> Your concern is valid and scientifically rigorous. Here we clarify an empirical observation from our experiments: when a heatmap performs well in guiding greedy search, it typically also performs well when guiding LKH-3. Below, we provide additional results based on greedy search (with 2-Opt).
>
> We use greedy search (G) to obtain **one** solution and apply 2-Opt for further improvement, which is aligned with the standard setup of heatmap-based methods. As a comparison, we reproduced the state-of-the-art GCN-based method NeuroLKH and Fast T2T. To ensure fairness and accurately assess generalization, we used the Fast T2T weights trained on 1,502K STSP-100 instances. Additionally, the 2-Opt procedure was also implemented using the Fast T2T version, with the maximum number of 2-Opt iterations uniformly set to 5,000.
>
> The table below reports the optimal gaps (%) of different methods and variants. As shown, when trained solely on STSP-100, Fast T2T exhibits limited generalization ability on large-scale STSPs. NeuroLKH, whose training set covers node sizes from 100 to 500, shows a slight advantage on STSP-1K and STSP-2K. However, its performance also degrades when the scale increases to STSP-5K and STSP-10K. In other words, the improvement of NeuroLKH largely stems from the increased node scales in its training data rather than from the efficiency of its GCN architecture.
>
> It is worth noting that, in terms of parameter count, UNE-GCN (**1.215M**) has only around one-third of NeuroLKH (3.553M) and one-fourth of Fast T2T (5.334M). Because NeuroLKH and Fast T2T serve as advanced GCN-based solvers, the resulting comparisons are representative and meaningful.
>
> | Method/STSP- | 1K (G)      | 1K (G+2-Opt) | 2K (G)      | 2K (G+2-Opt) | 5K (G)      | 5K (G+2-Opt) | 10K (G)     | 10K (G+2-Opt) |
> | :------: | :------: | :-----: | :------: | :-----: | :------: | :-----: | :------: | :------: |
> | ~ w/o U-N-E  | 23.9952 | 4.9385    | 24.3169 |    5.4521    | 23.5369     |    5.4831    | 23.0728     |    5.4483     |
> | ~ w/o U-E | 75.7699 | 6.5615 | 235.2008 |    8.9647    | 311.8848    |   10.1151    | 324.9831    |    27.7476    |
> | ~ w/o N-E | 24.5721 | 5.1382 | 23.4649 |    5.0482    | 23.4917 | 6.0011 | 23.4764     |    5.4342     |
> | ~ w/o E  | 18.5359|    3.1360    | *17.6326*  |  3.4647 | *19.3529*   |    3.5568    | *19.6614*   |   *3.5837*    |
> | Fast T2T | 27.6901 |    5.4985    | 44.1805 | 7.4812    | 74.1795 |    9.4983    | 101.6159    |    10.1587    |
> | NeuroLKH | **13.1563** |  **1.8764**  | **15.6162** |  **2.3832**  | 19.6152     |   *3.4222*   | 22.8993     |    4.4584     |
> | UNE-GCN  | *18.0433*   |   *2.9071*   | 18.1637     |   *2.9841*   | **18.2597** |  **3.2155**  | **17.8744** |  **3.1708**   |
>
>
>
> ### **(2) Evaluation Protocol**
>
> As stated in the **Important Clarifications**, our primary focus is to evaluate the ability of GCN-based solvers to generalize from **small-scale** to **large-scale** instances. Therefore, we mainly present results on large-scale instances, which also reflects the current trend in this research area.
>
> Moreover, our earlier experiments already show that UNE-GCN generalizes very well to smaller instances such as 200 and 500 nodes, as demonstrated in Figures 2 and 3 of the main paper. Given the already substantial amount of experimental data, we were concerned that including the small-scale results would make the presentation redundant.
>
> We sincerely apologize for the inconvenience caused by the current layout. In the subsequent version, we will include the complete results on small-scale instances as well.
>
> Intuitively, since UNE-GCN has not seen ATSPs of the TMAT class, its **cross-distribution** generalization is expected to be limited. To properly evaluate UNE-GCN's **cross-scale** generalization on ATSPs of the TMAT class, we would need to generate TMAT-class training data and retrain the model. Given that this process is quite time-consuming, we regret that we are unable to provide these results immediately.
>
> ------
>
> ## **Remarks**
>
> **We sincerely thank you for your critical review. We have made every effort to clarify the novelty and significance of our work, emphasizing that our focus lies in the design of the GCN architecture. We believe this work provides valuable insights and prior knowledge in the field of GCN-based TSP solvers, especially regarding the design of the backbone.**
>
> However, we are disappointed that the contribution score was rated as **[1: poor]**, which we feel does not fully reflect the scope and effort of our work. We hope that our current responses address your concerns and contribute to an improvement in the overall rating.
>
> If additional results are particularly requested to further support your assessment, please let us know in the next round of comments. We would be very happy to provide them and look forward to further discussion with you.

---

> ### Author Response · Authors · 2025-11-18
>
> ## **Q3. Supplementary Results for TMAT ATSP**
>
> For fairness, we include UniCO's GCN-based method, MatDIFFNet, as a baseline. We use the 2,500 ATSP-50 and 2,500 ATSP-100 instances publicly released in the UniCO repository as our test sets. Meanwhile, the weights for MatDIFFNet are taken from the checkpoints released by UniCO, namely `MatDIFFNet-ATSP50` and `MatDIFFNet-ATSP100`. Both UNE-GCN and MatDIFFNet adopt greedy search followed by 2-Opt as the post-search strategy. In addition, greedy search is restricted to sampling a single solution, and 2-Opt is implemented using UniCO's official code.
>
> As shown in the table below, our UNE-GCN uses only about one-sixth of the parameters of MatDIFFNet, and requires merely one-tenth of its training epochs. Despite being significantly more lightweight, UNE-GCN outperforms MatDIFFNet on both ATSP-50 and ATSP-100, while also achieving shorter inference time. These results collectively demonstrate that UNE-GCN achieves high efficiency with a much more lightweight architecture.
>
> We hope that these additional experiments address your concerns and provide positive clarification. If you have further questions or would like to see additional results, please feel free to let us know — we will do our best to respond.
>
> |   Method   |  #Params   | #Training instances | #Training Epochs |
> | :--------: | :--------: | :-----------------: | :--------------: |
> | MatDIFFNet |   6.986M   |        1.28M        |       100        |
> |  ~ w/o N   |   1.215M   |        1.00M        |        10        |
> |  UNE-GCN   | **1.215M** |        1.00M        |      **10**      |
>
> |     Method     |  ATSP-50   |       |  ATSP-100  |       |
> | :------------: | :--------: | :---: | :--------: | :---: |
> | Greedy |   Length   | Time  |   Length   | Time  |
> | MatDIFFNet  |   6.6611   | 0.47s |  12.5671   | 0.43s |
> |  ~ w/o N   |   3.6545   | 0.03s |   4.6009   | 0.04s |
> |  UNE-GCN    | **3.4989** | 0.03s | **4.5056** | 0.04s |
>
> |        Method         |  ATSP-50   |       |  ATSP-100  |       |
> | :-------------------: | :--------: | :---: | :--------: | :---: |
> | Greedy + 2-Opt |   Length   | Time  |   Length   | Time  |
> |      MatDIFFNet       |   3.2918   | 0.48s |   4.3353   | 0.43s |
> |        ~ w/o N        |   3.1687   | 0.03s |   3.9669   | 0.05s |
> |        UNE-GCN        | **3.0883** | 0.03s | **3.9429** | 0.05s |

---

> > ### Comment · Reviewer_N9Zi · 2025-11-27
> >
> > I thank the authors for the rebuttal and additional experimental clarifications. The updated results with greedy decoder (and 2-opt) are indeed much more comparable than those relying on LKH as a post-processor, which makes the overall empirical evidence **sounder**.
> >
> > However, I remain concerned that the technical contribution of this work is still somewhat limited. While I acknowledge the authors’ effort in explicitly presenting a comparative table showing these components (U/N/E) first appear jointly in this work, my concerns are the following.
> >
> > First, it remains unclear how substantial the performance gains brought by these additional elements would be relative to the original contributions of prior methods, under otherwise identical settings. Concretely, an ideal way to assess the true incremental value of UNE-GCN would be to replace the original GCN backbone in representative prior works (e.g., Att-GCN, NeuroLKH, DIFUSCO, Fast-T2T) with the proposed UNE-GCN, while keeping all other components like training protocols (e.g., the discrete diffusion mechanism), data scale, etc., strictly unchanged. I understand that such a broad retraining effort is beyond the scope and time budget of a rebuttal, and I do not expect the authors to conduct it at this stage. **My hypothesis, however, is that while the orthogonal, plug-in style application of UNE would very much likely bring consistent improvements, these gains may be incremental when compared against the main methodological novelties of those original works.**
> >
> > Second, several of the proposed design choices appear to be relatively mild extensions or consolidations of existing ideas. E.g., **learnable node** embeddings as an alternative to one-hot or coordinate-based inputs have already been discussed in MatNet and related works; **normalization** of edge weights (distances) or scale-dependent features has been adopted: MatNet adopts a global scaling factor; more recent works (e.g., arXiv:2408.16717, 2024) normalize distances instance-wise; and normalizing node coordinates to $[0,1]^2$ is even a common preprocessing step, especially when assessing a solver on real-world benchmarks such as TSPLIB. I acknowledge that your version of edge normalization does incorporate some distinct considerations and non-trivial design choices, but it somehow still appears more aligned with a refined engineering practice rather than introducing a fundamentally new modeling principle. In my view, the **multi-view GCN** idea is, the most promising architectural direction here. However, in its current form with manually chosen, heuristic k-NN views, it remains somewhat preliminary. A deeper or more principled study of multi-view aggregation could potentially stand as a strong backbone contribution on its own, closer to how papers proposing new GNN architectures are typically positioned and organized.
> >
> > In summary, after considering the authors’ clarifications, I see this work as making a **solid empirical and engineering contribution**: the proposed U/N/E components are **very likely effective, broadly applicable, and helpful for improving the robustness and generalization of GCN-based TSP solvers**. And some findings are interesting and practically useful. Nevertheless, I still find it difficult to view UNE-GCN as a sufficiently strong standalone architectural or conceptual contribution. Instead, I believe these insights would be most impactful if **positioned as supplementary or enhancing components that strengthen (maybe your) concurrent or future methods, rather than as the primary novelty here.** In this light, I decide to adjust the soundness and contribution scores, while maintaining my overall assessment.

---

### Official Review · Reviewer_2vXk · 2025-10-19

**Soundness:** 2
**Presentation:** 3
**Contribution:** 2
**Rating:** 4
**Confidence:** 4

**Summary:**

The paper addresses the challenges of generalization, overfitting, and difficulty in adapting GCN-based solvers to the TSP and ATSP. The authors propose the UNE-GCN model, which incorporates three strategies for improving performance: Input Unification (global node embedding), Edge Normalization (Min-Max scaling), and Aggregation Enhancement (layer-wise expanding views). Experimental results demonstrate that these strategies contribute to the generalization ability of the model, enhancing both the accuracy and efficiency of the GCN-based solver.

**Strengths:**

1. The paper includes thorough experimental analysis with extensive ablation studies, providing clear evidence for the effectiveness of each proposed strategy.
2. The proposed methods are easy to follow.

**Weaknesses:**

1. The individual techniques used—Min-Max scaling, edge aggregation, and pruning—are not entirely new and have been employed in previous works, which limits the overall novelty of the approach.
2. The paper should demonstrate that the generalization improvement is independent of the LKH solver used, as the observed improvements may be a result of enhancing LKH rather than improving generalization per se.
3. The mainly compared baseline model (Neurolkh) is from 2021 (although exists some baselines without combining LKH in Appendix). It would be beneficial to compare the UNE-GCN with more recent models based on LKH, such as VSR-LKH, DualOpt, and Select and Optimize, to strengthen the comparative analysis.
4. While the paper focuses on TSP, it remains unclear whether the proposed approach can be generalized to other combinatorial optimization problems like CVRP or FJSP, which could broaden the impact of the proposed model.
5. The adopted MR@5 metric cannot fully reflect the effectiveness of the model, because combinatorial optimization problems are not classification tasks.

**Questions:**

Could the authors provide results without the use of LKH to better isolate the contribution of UNE-GCN itself, separate from the post-processing improvements contributed by LKH?

---

> ### Author Response · Authors · 2025-11-16
>
> ## **1. Important Clarifications**
>
> First, we would like to emphasize that the GCN-based baselines mentioned in our paper—Att-GCN, DIFUSCO, Fast T2T, RsGCN, and NeuroLKH—use standard Gated GCN or its slight variants as encoders. In fact, the standard Gated GCN is essentially equivalent to our ablation variant ~ w/o U-N-E. We apologize for not making this explicit in the main text, which may have caused some confusion. The table below clearly shows the component composition of these GCN-based baselines and their correspondence to the ablation variants in our paper. Furthermore, this table also provides evidence for the novelty that distinguishes our work from prior methods.
>
> Our systematic study of UNE-GCN and its ablation variants (~ w/o E, ~ w/o N-E, ~ w/o U-E, and ~ w/o U-N-E) is essentially a systematic rethinking of prior GCN encoders. **Here, LKH serves primarily a validation role, while MR@5 is sufficient to clearly pre-evalute the model's generalization performance.** In contrast, comparisons with other baselines (e.g., Transformer-based methods) are less important. In our ablation experiments, we compare GCN encoders individually under fair and consistent conditions, thereby eliminating the influence of decoders, training data, loss function, and other factors.
>
> |  Model   | Corresponding Variant | Input Unification | Edge Normalization | Aggregation Enhancement |
> | :------: | :-------------------: | :---------------: | :----------------: | :---------------------: |
> | Att-GCN  |      ~ w/o U-N-E      |         ❌         |         ❌          |            ❌            |
> | NeuroLKH |      ~ w/o U-N-E      |         ❌         |         ❌          |            ❌            |
> | DIFUSCO  |      ~ w/o U-N-E      |         ❌         |         ❌          |            ❌            |
> | Fast T2T |      ~ w/o U-N-E      |         ❌         |         ❌          |            ❌            |
> |  RsGCN   |       ~ w/o U-E       |         ❌         |         ✅          |            ❌            |
> | UNE-GCN  |                       |         ✅         |         ✅          |            ✅            |
>
> ------
>
> ## **2. Results with Greedy Search**
>
> We use greedy search (G) to obtain **one** solution and apply 2-Opt for further improvement. As a comparison, we reproduced the state-of-the-art GCN-based method NeuroLKH and Fast T2T. To ensure fairness and accurately assess generalization, we used the Fast T2T weights trained on 1,502K STSP-100 instances. Additionally, the 2-Opt procedure was also implemented using the Fast T2T version, with the maximum number of 2-Opt iterations uniformly set to 5,000.
>
> The table below reports the optimal gaps (%) of different methods and variants. As shown, when trained solely on STSP-100, Fast T2T exhibits limited generalization ability on large-scale STSPs. NeuroLKH, whose training set covers node sizes from 100 to 500, shows a slight advantage on STSP-1K and STSP-2K. However, its performance also degrades when the scale increases to STSP-5K and STSP-10K. In other words, the improvement of NeuroLKH largely stems from the increased node scales in its training data rather than from the efficiency of its GCN architecture.
>
> It is worth noting that, in terms of parameter count, UNE-GCN (1.215M) has only around one-third of NeuroLKH (3.553M) and one-fourth of Fast T2T (5.334M).
>
> | Method/STSP- | 1K (G)      | 1K (G+2-Opt) | 2K (G)      | 2K (G+2-Opt) | 5K (G)      | 5K (G+2-Opt) | 10K (G)     | 10K (G+2-Opt) |
> | :----------: | :------: | :----------: | :------: | :----------: | :------: | :----------: | :------: | :-----------: |
> | ~ w/o U-N-E  | 23.9952     |    4.9385    | 24.3169     |    5.4521    | 23.5369     |    5.4831    | 23.0728     |    5.4483     |
> |  ~ w/o U-E   | 75.7699     |    6.5615    | 235.2008    |    8.9647    | 311.8848    |   10.1151    | 324.9831    |    27.7476    |
> |  ~ w/o N-E   | 24.5721     |    5.1382    | 23.4649     |    5.0482    | 23.4917     |    6.0011    | 23.4764     |    5.4342     |
> |   ~ w/o E    | 18.5359     |    3.1360    | *17.6326*   |    3.4647    | *19.3529*   |    3.5568    | *19.6614*   |   *3.5837*    |
> |   Fast T2T   | 27.6901     |    5.4985    | 44.1805     |    7.4812    | 74.1795     |    9.4983    | 101.6159    |    10.1587    |
> |   NeuroLKH   | **13.1563** |  **1.8764**  | **15.6162** |  **2.3832**  | 19.6152     |   *3.4222*   | 22.8993     |    4.4584     |
> |   UNE-GCN    | *18.0433*   |   *2.9071*   | 18.1637     |   *2.9841*   | **18.2597** |  **3.2155**  | **17.8744** |  **3.1708**   |

---

> ### Author Response · Authors · 2025-11-16
>
> ## **3. Comparison with DualOpt**
>
> In the publication of NeuroLKH, prior experiments have already demonstrated that NeuroLKH outperforms VSR-LKH in overall performance. In addition, we found that the code repository of Select and Optimize is no longer available. Therefore, we focus on reproducing DualOpt and include it as the comparison on large-scale STSPs. The table below presents the average tour length and solution time, where both plain LKH and UNE-GCN + LKH are evaluated with `MAX_TRIALS=10`. To ensure fairness, we disabled the `SPECIAL` option in DualOpt' LKH to obtain better solutions and align the setting with ours. It can be observed that UNE-GCN consistently delivers better solution quality and less solution time.
>
> |    Method/STSP-     |     1K      |        |     2K      |       |     5K      |        |     10K     |        |
> | :-----------------: | :---------: | ------ | :---------: | ----- | :---------: | ------ | :---------: | ------ |
> |                     |   Length    | Time   |   Length    | Time  |   Length    | Time   |   Length    | Time   |
> |         LKH         |   23.1300   | 1.17s  |   32.4942   | 4.69s |   51.0049   | 33.17s |   71.8454   | 2.65m  |
> |       DualOpt       |   23.2847   | 18.30s |   32.5930   | 1.12m |   51.4501   | 3.31m  |   72.1765   | 15.59m |
> | UNE-GCN | **23.1263** | 0.40s  | **32.4891** | 1.42s | **51.0012** | 10.04s | **71.8277** | 55.52s |
>
> ------
>
> ## **4. Clarification of MR@5**
>
> In fact, MR@5, as a discrete metric, reflects the model performance more effectively than the loss and enables fast evaluation without performing any search. Furthermore, our extensive experiments demonstrate a consistent correlation between MR@5 and the solution quality: a lower MR@5 typically corresponds to higher-quality solutions obtained by LKH-3 and greedy search.
>
> ------
>
> ## **5. Remarks**
>
> Thank you for your valuable review comments! We have made every effort to include additional experiments, and the new results further validate the strong generalization ability of UNE-GCN. Meanwhile, we will also include the new experimental results in subsequent versions of the paper. If you have any remaining concerns, please feel free to let us know. We hope our responses address your questions and contribute to an improved rating.

---

> > ### Comment · Reviewer_2vXk · 2025-11-19
> >
> > 1. Although the authors emphasize the innovativeness of their work, I consider the contribution of the integrated technologies to the field to be negligible. Methods adopting these technologies are essentially ubiquitous, such as INVIT, ICAM, L2R, GELD, etc. Indeed, the models you listed (e.g., NeuroLKH, DIFUSCO) do not employ these specific technologies, but their advantages lie in the novelty of their workflows rather than the encoding approach for TSP. In its current form, I believe the technical innovativeness of the proposed method is insufficient.
> >
> > 2. Still no results for other COPs.
> >
> > 3. I do not consider that MR@5 has a fully consistent correspondence with the solution quality of the NCO model. In extreme cases, the optimal solution will necessarily correspond to an MR@5 of 0%, but an MR@5 of 0% is highly likely to be associated with poor solution quality
> >
> > In summary, I maintain my original evaluation score.

---

> ### Author Response · Authors · 2025-11-19
>
> ## **Response to Comment 1**
>
> We do not follow your view that “methods adopting these technologies are essentially ubiquitous.”
>
> First, we carefully reviewed the four Transformer-based works you cited (INViT, ICAM, L2R, GELD). Among them, only INViT has been formally published. We do not find comparisons with unpublished works or with methods built on fundamentally different backbones to be persuasive. More importantly, while we acknowledge that some of these works apply **Edge Normalization**, we could not find clear evidence that they implement **Input Unification** or **Aggregation Enhancement** as we define them. If you have concrete pointers showing where those two components appear in those papers, please let us know.
>
> Crucially, all four methods are Transformer-based and therefore differ substantially from GCNs in both architecture and prediction manner. Moreover, GCNs are edge-aware, whereas Transformers are node-aware. This raises concrete questions: can Transformers naturally leverage global node vectors the way GCNs do? Where would Input Unification be realized in a Transformer architecture?
>
> GCNs have long suffered from overfitting and oversmoothing in cross-scale generalization, a problem that has not been adequately identified or addressed in prior work. Our contribution goes beyond introducing the U-N-E components to address this issue. More importantly, we provide a detailed investigation of how different ablation variants affect the backbone's cross-scale generalization, highlighting the importance of balancing performance and computational cost. The compact parameter size and stable learning dynamics of UNE-GCN further reinforce these insights.
>
> We agree that novel workflows are important. However, improvements to the underlying neural architecture are equally valuable. These are complementary contributions at different levels — both deserve recognition. UNE-GCN achieves better performance and cross-scale generalization at a substantially lower cost. We should not focus solely on developing new workflows; equal attention should be given to the efficiency and performance of the underlying backbone models.
>
> ------
>
> ## **Response to Comment 2**
>
> Our title states our scope: **"Rethinking GCNs for the Traveling Salesman Problem"**. Extensive experimental results on both STSP and ATSP sufficiently demonstrate the robust representational capacity of UNE-GCN, particularly on cross-scale generalization.  To the best of our knowledge, UNE-GCN achieves SOTA cross-scale generalization and the advantage on large-scale ATSP are significant compared to prior GCN-based methods.
>
> Therefore, our UNE-GCN focuses on cross-scale generalization in solving two typical type of TSP (STSP and ATSP) and extensive experiment are conducted. Further experiments for the generalization to other COPs may deviate the focus of work.
>
> In addition, NeuroLKH has shown that the architecture of GCN + LKH is also capable for CVRP, indicating the transferability of GCN-based backbones across COPs. Thus, the UNE-GCN is also promising in dealing with other COPs. Given that experiments on other COPs require substantial computational cost and time, we would like to inquire whether their inclusion is considered essential and can lead to a higher rating.
>
> ------
>
> ## **Response to Comment 3**
>
> We would like to clarify that the phrase "**fully consistent**" does not appear in our manuscript, and it does not reflect the claim we intended to make. Instead, our claim concerns the **overall trend**, which is supported by extensive empirical evidence—given the same post-search method, a model with better MR@5 result typically yields overall better solution quality. The extreme cases you mentioned are theoretically possible but may not be practically meaningful—we have not encountered situations where MR@5 is 0 yet the solution quality is still poor.
>
> We fully understand your point.  We would like to note that **our performance evaluation is multifaceted, not solely rely on MR@5.** Results based on LKH-3 search, greedy search, and 2-Opt are also presented, all of which serve as complementary assessments. Taken together, this evaluation framework is both reasonable and convincing.
>
>
>
> Thanks for your further comments. We hope our responses address your concerns. If you have any further questions, please feel free to let us know.

---

### Official Review · Reviewer_SU2T · 2025-10-28

**Soundness:** 2
**Presentation:** 2
**Contribution:** 2
**Rating:** 4
**Confidence:** 4

**Summary:**

This paper rethinks the use of Graph Convolutional Networks (GCNs) in solving the Traveling Salesman Problem (TSP). It introduces three encoder strategies: input unification, edge normalization, and aggregation enhancement, to improve representation learning. Extensive ablation studies are conducted to demonstrate the effectiveness of these strategies.

**Strengths:**

The proposed input unification strategy is well designed, enabling the model to handle asymmetric TSP (ATSP) instances effectively. The experiments on the three encoder strategies are comprehensive, and the model demonstrates strong generalization ability across different size instances.

**Weaknesses:**

1. This paper claims to be a rethinking of GCNs for the TSP. However, the experiments do not demonstrate how this rethinking applies to other GCN-based models. Such experiments are necessary; otherwise, the work reads more like the proposal of a new model rather than a true rethinking. Moreover, the experiments are limited to applications within the LKH framework only.

2. DIFUSCO and T2T are GNN-based models, not standard GCNs. The paper’s description of these two methods is therefore inaccurate and should be corrected.

3. Regarding LKH, it is recommended to use version 3.0.6, since different LKH versions vary in efficiency and implementation details. NeuroLKH was trained based on version 3.0.6, under which its runtime is shorter than that of LKH. However, the paper reports that NeuroLKH is slower, which contradicts the original paper. Comparing under version 3.0.6 would be a fairer and more consistent setup. It would also be preferable to compare under the same iteration count rather than same runtime limits.

4. For DIFUSCO, the original work used a 2-opt search as the default configuration, whereas this paper reports results under MCTS. To ensure consistency, it is recommended to evaluate DIFUSCO using the default 2-opt search, since the model was trained and tuned under that setting.

5. Including comparison with baseline on the 200-node and 500-node datasets is more easier for a more comprehensive evaluation of model scalability and generalization.

6. The paper states that the candidate set size for ATSP is set to 6, but the default in LKH should be 5. Unless there is a specific reason for this adjustment, it is suggested to follow the default setting.

7. The main text contains a large number of tables without corresponding discussion, and readers are forced to consult the Appendix, which is not ideal. It is recommended to move the key descriptions from the Appendix into the main body, possibly moving some tables to the Appendix instead. Furthermore, comparisons with baseline models are particularly important and should be presented in the main text.

**Questions:**

1. Whether this paper is truly a rethinking or simply a new model remains unclear. Although the title claims it to be a rethinking, the content reads more like the proposal of a new LKH-based model.

2. The others are shown in weakness.

---

> ### Author Response · Authors · 2025-11-15
>
> ## **Q1. Important Clarifications**
>
> First, we would like to emphasize that the GCN-based baselines mentioned in our paper—Att-GCN, DIFUSCO, Fast T2T, RsGCN, and NeuroLKH—use standard Gated GCN or its slight variants as encoders. In fact, the standard Gated GCN is essentially equivalent to our ablation variant ~ w/o U-N-E. We apologize for not making this explicit in the main text, which may have caused some confusion. The table below clearly shows the component composition of these GCN-based baselines and their correspondence to the ablation variants in our paper. Furthermore, this table also provides evidence for the novelty that distinguishes our work from prior methods.
>
> Our systematic study of UNE-GCN and its ablation variants (~ w/o E, ~ w/o N-E, ~ w/o U-E, and ~ w/o U-N-E) is essentially a systematic rethinking of prior GCN encoders. **Here, LKH serves primarily a validation role, while MR@5 is sufficient to clearly pre-evalute the model's generalization performance.** In contrast, comparisons with other baselines (e.g., Transformer-based methods) are less important. In our ablation experiments, we compare GCN encoders individually under fair and consistent conditions, thereby eliminating the influence of decoders, training data, loss function, and other factors.
>
> |  Model   | Corresponding Variant | Input Unification | Edge Normalization | Aggregation Enhancement |
> | :------: | :-------------------: | :---------------: | :----------------: | :---------------------: |
> | Att-GCN  |      ~ w/o U-N-E      |         ❌         |         ❌          |            ❌            |
> | NeuroLKH |      ~ w/o U-N-E      |         ❌         |         ❌          |            ❌            |
> | DIFUSCO  |      ~ w/o U-N-E      |         ❌         |         ❌          |            ❌            |
> | Fast T2T |      ~ w/o U-N-E      |         ❌         |         ❌          |            ❌            |
> |  RsGCN   |       ~ w/o U-E       |         ❌         |         ✅          |            ❌            |
> | UNE-GCN  |                       |         ✅         |         ✅          |            ✅            |
>
> ------
>
> ## **Q2. The Backbone of DIFUSCO and Fast T2T**
>
> We have thoroughly reviewed the original papers and source code of DIFUSCO and Fast T2T. As stated in our response to Q1, the encoders of DIFUSCO and Fast T2T are standard Gated GCNs, corresponding to our ablation variant ~ w/o U-N-E. We are confident in our understanding of their encoder architecture.
>
> ---
>
> ## **Q3. Clarification of LKH and NeuroLKH**
>
> Through extensive experiments, we found that LKH-3.0.6 and 3.0.13 have almost identical efficiency. We note that NeuroLKH uses LKH-3.0.6 for graph sparsification. We keep this part unchanged, but unify the search phase to LKH-3.0.13 to ensure fairness. Regarding your comment that “the paper reports that NeuroLKH is slower,” we also observed this behavior on TSPLIB.
>
> When solving TSPLIB instances, NeuroLKH only provides the `CANDIDATE_FILE` without a `PI_FILE`, and uses LKH's **Subgradient Optimization** to compute pi values. However, since `MAX_CANDIDATES` is not set to 0, **Subgradient Optimization** modifies the contents of the `CANDIDATE_FILE`, which can obscure the true evaluation of the model's performance. Overall, when a `CANDIDATE_FILE` is already provided and the **Subgradient Optimization** is enabled, setting `MAX_CANDIDATES` to 0 is the more appropriate choice. Only under this setting do the results reflect the true performance of NeuroLKH.
>
> In the main paper (Table 5), we compare with NeuroLKH under the same runtime limits, while in Appendix (Table 6), the comparison is conducted under the same number of iterations. Here we provide additional results on **tour length** for NeuroLKH with LKH-3.0.6 and LKH-3.0.13, respectively (`MAX_TRIALS=1000`). As shown in the table below, the search results of NeuroLKH under the two versions are very similar.
>
> |   Method    |   STSP-1K    |   STSP-2K    |     STSP-5K      |     STSP-10K     |
> | :---------------: | :-----: | :-----: | :---------: | :---------: |
> | NeuroLKH (3.0.6)  | 23.1183 | 32.4755 |   50.9723   |   71.7909   |
> | NeuroLKH (3.0.13) | 23.1182 | 32.4752 |   50.9704   |   71.7927   |

---

> ### Author Response · Authors · 2025-11-17
>
> ## **Q4. Post-Search of DIFUSCO**
>
> In the publication of DIFUSCO, MCTS is also used as a post-search and achieves better results. Therefore, evaluating DIFUSCO with MCTS does not compromise fairness; if anything, it favors DIFUSCO's reported results.
>
> ---
>
> ## **Q5. Evaluation on Small-Scale TSPs**
>
> This work focuses on the ability of GCN to generalize from small-scale to large-scale TSPs, which is why we emphasize results on large-scale TSPs. Moreover, the experimental scope of this paper is already substantial, and presenting small-scale TSP results would be somewhat redundant. Nevertheless, we can still quickly evaluate the performance of UNE-GCN and its ablation variants on small-scale TSPs using MR@5, as shown in Figure 23 of Appendix L.
>
> ---
>
> ## **Q6. The Candidate Size of ATSP**
>
> Setting the candidate set size to 6 for ATSP is more appropriate. When LKH-3 handles ATSP-$n$, it first transforms it into STSP-$2n$ via the JV Transformation. As shown in Appendix B, one element of the candidate set is occupied by the fixed edge $x_i \to x_{n+i}$, reducing the candidate set by one. Therefore, we set the candidate set size to 6 for ATSP, making the actual search space equivalent to candidate sets of size 5, which aligns with the default settings of STSP.
>
> ---
>
> ## **Q7. Presentation Problem**
>
> Thank you for pointing this out. We agree that the structure and formatting of our paper can be further improved. We will carefully incorporate your suggestions and refine the paper in the subsequent revision.
>
> ---
>
> ## **Remarks**
>
> Thank you for raising these detailed questions, and we look forward to further discussion with you. We hope our responses address your concerns and contribute to an improvement in rating.

---

> > ### Comment · Reviewer_SU2T · 2025-11-27
> >
> > Thank you for the revisions and clarifications. I would like to reiterate a few important points that are essential for a paper positioned as a rethinking work.
> >
> > First, if the paper aims to rethink existing paradigms, then demonstrating the applicability and generalizability of the proposed perspective is important. Showing results only on LKH is not sufficient. Even if previous works often adopt the w/o U-N-E setting, their key contributions are not limited to the network architecture but also include the training strategies and methodological design. Therefore, I strongly believe that demonstrating applicability to other representative methods is necessary.
> >
> > Second, the evaluation should follow the standardized setup of NeuroLKH. Deviating from its experimental protocol makes the comparison unfair and weakens the validity of the conclusions.
> >
> > Third, DIFUSCO is trained with 2-opt, and thus should also be evaluated with 2-opt at test time. Ensuring methodological consistency is important for interpreting the results correctly.
> >
> > At this stage, my assessment focuses primarily on these three aspects. I hope the authors can strengthen the experiments accordingly. I will maintain my current score for now and look forward to seeing the improved version.

---

> > > ### Author Response · Authors · 2025-12-02
> > >
> > > We sincerely appreciate your feedback, which is highly important for enhancing our work. We would be delighted to strengthen the experiments based on your suggestions.
> > >
> > > ---
> > >
> > > ## **1. Validation of Applicability to Methods other than LKH**
> > >
> > > To eliminate the influence of the LKH, we use greedy search (G) to obtain **one** solution and apply 2-Opt for further improvement (G+2-Opt). As a comparison, we reproduce the state-of-the-art GCN-based method NeuroLKH and Fast T2T. To ensure fairness and accurately assess generalization, we use the Fast T2T weights trained on 1,502K STSP-100 instances. Additionally, the 2-Opt procedure is also implemented using the Fast T2T version, with the maximum number of 2-Opt iterations uniformly set to 5,000.
> > >
> > > The table below reports the optimal gaps (%) of different methods and variants. As shown, when trained solely on STSP-100, Fast T2T exhibits limited generalization ability on large-scale STSPs. NeuroLKH shows a slight advantage on STSP-1K and STSP-2K and  but  its performance also degrades when the scale increases to STSP-5K and STSP-10K. Considering that the training set of NeuroLKH covers node sizes from 100 to 500 while the training set of UNE-GCN contains STSPs with at most 100 nodes, the improvement of NeuroLKH largely stems from the increased node scales in its training data rather than from the efficiency of its architecture.
> > >
> > > It is worth noting that, in terms of parameter count, UNE-GCN (1.215M) has only around one-third of NeuroLKH (3.553M) and one-fourth of Fast T2T (5.334M).
> > >
> > > |Method/STSP-|1K (G)|1K (G+2-Opt)|2K (G)|2K (G+2-Opt)|5K (G)|5K (G+2-Opt)|10K (G)|10K (G+2-Opt)|
> > > |-|:-:|:-:|:-:|:-:|:-:|:-:|:-:|:-:|
> > > |~w/o U-N-E|23.9952|4.9385|24.3169|5.4521|23.5369|5.4831|23.0728|5.4483|
> > > |~w/o U-E|75.7699|6.5615|235.2008|8.9647|311.8848|10.1151|324.9831|27.7476|
> > > |~w/o N-E|24.5721|5.1382|23.4649|5.0482|23.4917|6.0011|23.4764|5.4342|
> > > |~w/o E|18.5359|3.1360|*17.6326*|3.4647|*19.3529*|3.5568|*19.6614*|*3.5837*|
> > > |Fast T2T|27.6901|5.4985|44.1805|7.4812|74.1795|9.4983|101.6159|10.1587|
> > > |NeuroLKH|**13.1563**|**1.8764**|**15.6162**|**2.3832**|19.6152|*3.4222*|22.8993|4.4584|
> > > |UNE-GCN|*18.0433*|*2.9071*|18.1637|*2.9841*|**18.2597**|**3.2155**|**17.8744**|**3.1708**|
> > >
> > > ---
> > >
> > > ## **2. Results with the Standardized Setup of NeuroLKH**
> > >
> > > We have re-run the experiments based on the **LKH-3.0.6** version and parameter settings adopted by NeuroLKH. The results are presented in the table below. As shown, UNE-GCN still demonstrates superior performance in overall compared to NeuroLKH. The results based on **LKH-3.0.6** further strengthen our experiments and conclusions.
> > >
> > > ### **A. Uniform-distribution STSPs**
> > >
> > > |   Method/STSP-   |   1K    |             |   2K    |             |   5K    |             |   10K   |             |
> > > | :--------------: | :-----: | :---------: | :-----: | :---------: | :-----: | :---------: | :-----: | :---------: |
> > > |                  | Length  |   Gap (%)   | Length  |   Gap (%)   | Length  |   Gap (%)   | Length  |   Gap (%)   |
> > > |      LKH-3       | 23.1192 |   0.0000    | 32.4762 |   0.0000    | 50.9724 |   0.0000    | 71.7800 |   0.0000    |
> > > |     NeuroLKH     | 23.1183 |   -0.0040   | 32.4755 | **-0.0022** | 50.9723 |   -0.0002   | 71.7909 |   0.0152    |
> > > | UNE-GCN ($l$=6)  | 23.1184 |   -0.0037   | 32.4765 |   0.0009    | 50.9688 | **-0.0071** | 71.7788 |   -0.0016   |
> > > | UNE-GCN ($l$=18) | 23.1182 | **-0.0043** | 32.4757 |   -0.0017   | 50.9693 |   -0.0060   | 71.7784 | **-0.0023** |
> > >
> > > ### **B. Real-world (TSPLIB) STSPs**
> > >
> > > | Scale/Method |  NeuroLKH  |       |  UNE-GCN   |        |
> > > | :----------: | :--------: | :---: | :--------: | :----: |
> > > |              |  Gap (‰ )  | Time  |  Gap (‰ )  |  Time  |
> > > |    ＜100     | **0.0000** | 0.01s | **0.0000** | 0.01s  |
> > > |  [100, 1K)   |   0.3706   | 0.35s | **0.2139** | 0.28s  |
> > > |  [1K, 10K)   |   0.6730   | 1.24m | **0.6469** | 1.15m  |
> > > |     ≥10K     |   0.6392   | 1.65h | **0.0545** | 2.91h  |
> > > |     All      |   0.4524   | 6.74m | **0.3205** | 11.56m |
> > >
> > > ------
> > >
> > > ## **3. Results of DIFUSCO with 2-Opt**
> > >
> > > Due to reproduction issues, we directly utilize the STSP-1K/10K heatmaps provided by DIFUSCO's repository. We also adopt the 2-Opt implementation in DIFUSCO, and maintain the maximum number of 2-Opt iterations at the default value of 5000. The specific results are shown in the table below, which further enriches our experimental results and shows that DIFUSCO + MCTS is superior to DIFUSCO + 2-Opt.
> > >
> > > |Method/STSP-| 1K  |   | 10K  |   |
> > > | --- | ------- | ------- | ------- | ------- |
> > > |    | Length  | Gap (%) | Length  | Gap (%) |
> > > | DIFUSCO + MCTS  | 23.4243 | 1.3197  | 73.8913 | 2.9413  |
> > > | DIFUSCO + 2-Opt | 24.4113 | 5.5942  | 76.3170 | 6.3261  |
> > >
> > > ---
> > >
> > > ## **Remarks**
> > >
> > > We sincerely thank you once again for your solid suggestions on our work. We will incorporate the relevant descriptions and supplemental experimental results in the subsequent version.

---

### Official Review · Reviewer_2Lz4 · 2025-10-30

**Soundness:** 3
**Presentation:** 3
**Contribution:** 3
**Rating:** 6
**Confidence:** 5

**Summary:**

This paper presents UNE-GCN, a novel approach to solving the Traveling Salesman Problem (TSP) using Graph Convolutional Networks (GCNs). The authors introduce three key encoding strategies—Input Unification, Edge Normalization, and Aggregation Enhancement (U-N-E)—to address the challenges of generalization, overfitting, and scalability in GCN-based solvers. The experimental results show that UNE-GCN outperforms traditional TSP solvers like LKH-3, particularly in handling large-scale instances (up to 100K nodes). The paper makes a significant contribution by improving the generalization ability of GCNs for combinatorial optimization problems.

**Strengths:**

1. The three encoding techniques—Input Unification, Edge Normalization, and Aggregation Enhancement—are well-justified and provide a unified approach to graph encoding that helps overcome the limitations of previous GCN-based solvers. This innovation is a key strength of the paper.

2. The paper presents a thorough set of experiments comparing UNE-GCN with baseline methods, including LKH-3. The results demonstrate a clear advantage in performance, especially on large TSP instances. The ablation studies add depth to the findings and validate the effectiveness of each encoding strategy.

3. UNE-GCN is shown to scale efficiently with larger problem sizes, with performance improvements observed even on instances with up to 100,000 nodes. This scalability is essential for real-world applications where TSP problems can grow quite large.

4. The insights into encoding strategies and generalization in TSP solvers can have broader implications for future research in combinatorial optimization and graph-based neural networks. The framework is easy to extend to other similar problems.

**Weaknesses:**

1.Although UNE-GCN achieves good performance, the added complexity of the model (with respect to parameters and training time) could be a barrier to its practical deployment, particularly for large-scale instances. A more thorough discussion on the computational efficiency and trade-offs involved would be helpful.
2. While the paper compares UNE-GCN to traditional TSP solvers like LKH-3, it doesn't sufficiently compare it to other state-of-the-art neural network-based methods (e.g., GATs, graph attention networks, or reinforcement learning-based solvers). A more comprehensive comparison would help position UNE-GCN more clearly in the context of neural optimization approaches.
3. The paper focuses primarily on benchmark datasets, but it could provide more insight into how the model performs in real-world TSP instances, which may have more irregular or noisy data. Further discussion on the generalizability of the model to diverse TSP variants would strengthen the paper.

**Questions:**

1.How does UNE-GCN compare to simpler models in terms of memory usage and inference time for large-scale TSP instances? Could the model be optimized for better computational efficiency?

2.the authors provide a more in-depth comparison with other neural approaches, particularly reinforcement learning-based solvers or GCN variants? How does UNE-GCN's performance compare in terms of optimization stability and convergence speed?

3.The paper provides experimental evidence of the approach's success, but it would be valuable to include some theoretical discussion or guarantees regarding the improvements brought by the proposed encoding strategies. Could the authors provide a theoretical analysis to justify the benefits of the three encoding techniques?

4. While the paper shows great results on benchmark datasets, how does UNE-GCN generalize to more complex real-world TSP instances (e.g., with noisy or incomplete data)? Can the authors show the robustness of the model in more varied settings?

---

> ### Author Response · Authors · 2025-11-14
>
> Thank you for your valuable review comments. We address each of your questions one by one below.
>
> ## **Q1. Computational Efficiency on Large-Scale TSPs**
>
> Since the GCN is a non-autoregressive encoder with a full-graph receptive field, it naturally incurs a relatively higher computational cost. Nevertheless, UNE-GCN remains more efficient and lightweight compared with GCN-Diffusion-based solvers (e.g., DIFUSCO and Fast T2T). For further improving efficiency, we evaluated inference in float-16 precision, and the results show almost no loss in accuracy—likely due to the robustness of the GCN’s full-graph representation.
>
> For future work, a [subgraph-wise independent encoding] → [subgraphs merging] strategy may offer a promising direction.
>
> ------
>
> ## **Q2. Additional Comparison with Neural Methods**
>
> It is important to emphasize that the ablation variant **~ w/o U-N-E** is essentially identical to the GCN-based encoder used in Att-GCN, DIFUSCO and Fast T2T. Figures 2–3 in the paper clearly demonstrate the superior optimization stability and faster convergence of UNE-GCN on unseen-scale TSPs.
>
> UNE-GCN adopts supervised learning (SL) and performs full-graph optimization, leading to substantially higher training efficiency compared with reinforcement learning (RL)–based solvers. The table below summarizes the training efficiency of several neural methods. As shown, SL methods typically converge faster; however, we argue that SL and RL are fundamentally different paradigms and therefore not directly comparable.
>
> | Paradigm |  Method  |              #Parameters              | Training Epochs |
> | :------: | :------: | :-----------------------------------: | :-------------: |
> |    SL    | UNE-GCN  | 0.417M ~ 1.215M |       10        |
> |    SL    |   LEHD   |                 1.43M                 |       150       |
> |    SL    |   DRHG   |                 2.65M                 |       100       |
> |    SL    | DIFUSCO  |                 5.33M                 |       50        |
> |    SL    | Fast T2T |                 5.33M                 |       50        |
> |    SL    | Att-GCN  |                 11.1M                 |       15        |
> |    RL    |   GLOP   |               1.30M × 3               |      > 500      |
> |    RL    |  H-TSP   |             5.34M + 2.51M             |      > 500      |
>
> ------
>
> ## **Q3. Theoretical Analysis of the Encoding Techniques**
>
> Due to space limitations, please refer to the next page.
>
> ------
>
> ## **Q4. More Complex Real-World TSP Instances**
>
> We present the results of UNE-GCN and NeuroLKH on TSPLIB in the appendix, where UNE-GCN still performs well on unseen real-world distributions. Regarding your concerns about noisy or incomplete data, we currently do not have access to such datasets or baselines; we would like to refer to any specific literature you could point us to for reproducing.
>
> ------
>
> ## **Remarks**
>
> We sincerely thank you for your professional and insightful questions, which have been extremely helpful for improving our work. We look forward to further discussions and hope that our responses address your concerns and lead to a positive impact on the rating.

---

> ### Author Response · Authors · 2025-11-14
>
> ## **Q3. Theoretical Analysis of the Encoding Techniques**
>
> **Edge Normlization** is fairly intuitive, as it enhances the numerical stability of the input data and prevents extremely small edge lengths in large-scale TSPs from causing degradation of the neural network's learned experience.
>
> **Aggregation Enhancement**, as a form of data augmentation and regularization, helps mitigate overfitting and oversmoothing in deep GCNs, thereby promoting stable convergence and improved generalization. The theoretical foundation is discussed in prior work, DropEdge [1]. For TSPs in Our UNE-GCN, Aggregation Enhancement is implemented as a heuristic DropEdge rather than a random one, preferentially retaining edge features that are more promising.
>
> Regarding **Input Unification**, we provide here a concise theoretical justification demonstrating that global node embedding is more effective than coordinate-based embedding:
>
> ### **(1) Problem Setup**
>
> For simplicity, we focus only on node inputs and ignore the effect of the distance matrix.
>
> Consider two TSP instances of different scales, with number of nodes $n$ and $m$, where $n < m$.
>
> Node coordinates:
> $$
> \\mathbf{x}\_i \\in [0,1]^2, \\quad i \\in \\{1, \\dots, n \\text{ or } m\\}
> $$
>
> Node embeddings are first constructed for each node and then aggregated (simplified mean aggregation) to form a fixed-dimensional representation:
>
> **Coordinate Linear Embedding with Aggregation:**
> $$
> X\_i = W \\mathbf{x}\_i + b, \\quad
> \\bar{\\mathbf{X}}\_n = \\frac{1}{n}\sum\_{i=1}^n X\_i, \\quad
> \\bar{\\mathbf{X}}\_m = \\frac{1}{m}\sum\_{i=1}^m X\_i
> $$
>
> **Global Embedding with Aggregation:**
> $$
> X\_i = e, \\quad
> \\bar{\\mathbf{X}}\_n = \\frac{1}{n}\\sum\_{i=1}^n e = e, \\quad
> \\bar{\\mathbf{X}}\_m = \\frac{1}{m}\\sum\_{i=1}^m e = e
> $$
>
> We focus on how the aggregated node features affect generalization. Let the model be $f\_\theta$, with expected risk:
> $$
> R(\\theta; \\bar{\\mathbf{X}}) = \\mathbb{E}[\\ell(f\_\theta(\\bar{\\mathbf{X}}))]
> $$
>
> The generalization gap between small and large scales is:
> $$
> \\|\\;R(\\theta; \\bar{\\mathbf{X}}\_m) - R(\\theta; \\bar{\\mathbf{X}}\_n)\\;\\|
> $$
>
> ---
>
> ### **(2) Lipschitz Assumption**
>
> Assume the model is Lipschitz continuous w.r.t. the aggregated input embeddings:
> $$
> \\|\\;f\_\theta(\\bar{\\mathbf{X}}\_n) - f\_\theta(\\bar{\\mathbf{X}}\_m)\\;\\| \le L \cdot \\|\\bar{\\mathbf{X}}\_n - \\bar{\\mathbf{X}}\_m\\|
> $$
>
> where $L$ is the Lipschitz constant.
>
> ---
>
> ### **(3) Analysis Based on Aggregated Embedding Norm Differences**
>
> **Coordinate Linear Embedding:**
>
> The embedding is linear:
> $$
> X\_i = W \\mathbf{x}\_i + b
> $$
>
> By linearity, for any two sets of node coordinates:
> $$
> \\|\\bar{\\mathbf{X}}\_m - \\bar{\\mathbf{X}}\_n\\| \le \\|W\\| \cdot \\Big\\| \frac{1}{m}\sum\_{i=1}^m \\mathbf{x}\_i - \frac{1}{n}\sum\_{i=1}^n \\mathbf{x}\_i \\Big\\|
> $$
>
> Combining with the Lipschitz assumption:
> $$
> \\|f\_\theta(\\bar{\\mathbf{X}}\_m) - f\_\theta(\\bar{\\mathbf{X}}\_n)\\| \le L \cdot \\|\\bar{\\mathbf{X}}\_m - \\bar{\\mathbf{X}}\_n\\| \le L \cdot \\|W\\| \cdot \\Big\\| \frac{1}{m}\sum\_{i=1}^m \\mathbf{x}\_i - \frac{1}{n}\sum\_{i=1}^n \\mathbf{x}\_i \\Big\\|
> $$
>
> As the number of nodes increases from $n$ to $m$, the coordinate distribution difference grows, and the linear mapping $W$ amplifies this difference.
>
> ---
>
> **Global Embedding:**
>
> All node embeddings are identical:
> $$
> X\_i = e
> $$
>
> Therefore, the difference is zero:
> $$
> \\|\\bar{\\mathbf{X}}\_m - \\bar{\\mathbf{X}}\_n\\| = 0
> $$
>
> By Lipschitz continuity:
> $$
> \\|f\_\theta(\\bar{\\mathbf{X}}\_m) - f\_\theta(\\bar{\\mathbf{X}}\_n)\\| \le L \cdot 0 = 0
> $$
>
> The input embedding distribution does not change with scale, so the model output is stable across scales.
>
> ---
>
> ### **(4) Conclusion**
>
> $$
> \begin{aligned}
> \text{Coordinate Embedding: } & \\|\\;R(\\theta;\\bar{\\mathbf{X}}\_m) - R(\\theta;\\bar{\\mathbf{X}}\_n)\\;\\| \le L \cdot \\|W\\| \cdot \\Big\\| \frac{1}{m}\sum\_{i=1}^m \\mathbf{x}\_i - \frac{1}{n}\sum\_{i=1}^n \\mathbf{x}\_i \\Big\\| \\\\
> \text{Global Embedding: } & \\|\\;R(\\theta;\\bar{\\mathbf{X}}\_m) - R(\\theta;\\bar{\\mathbf{X}}\_n)\\;\\| = 0
> \end{aligned}
> $$
>
> Coordinate embeddings amplify differences in input distributions as the scale increases.
>
> Global embeddings eliminate input variation, keeping the model output consistent and improving generalization from small to large scale.
>
> ### **Reference**
>
> [1] Yu Rong, Wenbing Huang, Tingyang Xu, and Junzhou Huang. "DropEdge: Towards deep graph convolutional networks on node classification." International Conference on Learning Representations (2020).
>
> [2] Virmaux, Aladin, and Kevin Scaman. "Lipschitz regularity of deep neural networks: Analysis and efficient estimation." Advances in Neural Information Processing Systems 31 (2018).

---

### Official Review · Reviewer_eBvv · 2025-10-31

**Soundness:** 3
**Presentation:** 3
**Contribution:** 3
**Rating:** 6
**Confidence:** 4

**Summary:**

The paper proposes **UNE-GCN**, a set of three encoding strategies (Input Unification — global node embeddings; Edge Normalization — min–max scaling; Aggregation Enhancement — multi-scale / layered neighborhood aggregation) to improve GCN-based tour-generation for TSP variants. The authors evaluate the approach on a broad suite of experiments (cross-scale synthetic STSP/ATSP, TSPLIB instances) and combine the learned components with LKH-3 in a two-stage pipeline. Results include ablations, scaling-law observations, and comparisons showing time-quality tradeoffs and gains over several neural baselines and, in some settings, over plain LKH-3.

**Strengths:**

* Clear motivation: focuses on how input/edge/aggregation encoding choices affect cross-scale generalization and ATSP/STSP differences.
* Simple, implementable modules (global node embedding, row-wise min–max scaling, layered aggregation) with individual contributions validated via ablation.
* Broad experimental coverage: large-scale synthetic instances, TSPLIB comparisons, and combination with LKH-3 showing practical time/quality tradeoffs.

**Weaknesses:**

* **Limited theoretical justification.** The paper lacks deeper analysis explaining why a global node embedding does not discard essential geometric information (especially for coordinate-based STSP).
* **Generalization to real-world distributions under-explored.** Training and evaluation are heavily based on synthetic/mixed-scale samples; coverage of realistic instance distributions is limited.
* **Instability on some TSPLIB cases.** Appendix tables show instances where performance/time gaps are worse — these failure modes are not analyzed in depth.
* **Reproducibility details missing.** Essential items to reproduce experiments are incomplete (data-generation scripts, exact LKH-3 and JV parameter settings, sampling strategy, random seeds, hardware/parallel setup).

**Questions:**

* Please provide a deeper, quantitative justification for Input Unification (global node embeddings). How and why does replacing coordinate-based embeddings with a global node embedding retain sufficient geometric/topological information? Can you show per-instance comparisons (coordinate embedding vs. global embedding) on a held-out STSP validation set?
* How robust is min–max edge normalization to outliers or extreme edge-length distributions? Have you tried robust alternatives (e.g., quantile-based scaling, clipping) and how do they compare?
* Clarify training details: sampling strategy for the stated 1M training instances (are all seen each epoch?), full training curves (loss, validation metrics) and total training time. This helps assess whether models were trained sufficiently.
* For TSPLIB instances where UNE-GCN performs worse or inconsistently, can you provide qualitative/quantitative analysis (instance topology, degree distributions, coordinate patterns) to explain failure modes?
* For ATSP + two-stage LKH-3 pipeline: provide precise details of the Johnson-Ven der (JV) transformation implementation and examine whether JV or any preprocessing changes candidate structure in a way that biases comparisons.

---

> ### Author Response · Authors · 2025-11-14
>
> Thank you for your valuable and insightful review comments. We address each question in detail below.
>
> ## **Q1. Brief Theoretical Analysis on Global Node Embedding**
>
> Due to the rebuttal length limit, we provide a concise version of the theoretical analysis, demonstrating that using a global node embedding is more effective than coordinate-based embeddings.
>
> First, it is important to clarify that discarding node coordinates and using only the distance matrix does not impair the geometric or topological information of the graph. In the TSP, the tour length is ultimately computed solely from edge lengths. Therefore, the distance matrix fully preserves the geometric structure required for the task.
>
> Next, we provide a theoretical justification—based on the Lipschitz condition—showing why the global node embedding leads to better generalization.
>
> ### **(1) Problem Setup**
>
> For simplicity, we focus only on node inputs and ignore the effect of the distance matrix.
>
> Consider two TSP instances of different scales, with number of nodes $n$ and $m$, where $n < m$.
>
> Node coordinates:
> $$
> \\mathbf{x}\_i \\in [0,1]^2, \\quad i \\in \\{1, \\dots, n \\text{ or } m\\}
> $$
>
> Node embeddings are first constructed for each node and then aggregated (simplified mean aggregation) to form a fixed-dimensional representation:
>
> **Coordinate Linear Embedding with Aggregation:**
> $$
> X\_i = W \\mathbf{x}\_i + b, \\quad
> \\bar{\\mathbf{X}}\_n = \\frac{1}{n}\sum\_{i=1}^n X\_i, \\quad
> \\bar{\\mathbf{X}}\_m = \\frac{1}{m}\sum\_{i=1}^m X\_i
> $$
>
> **Global Embedding with Aggregation:**
> $$
> X\_i = e, \\quad
> \\bar{\\mathbf{X}}\_n = \\frac{1}{n}\\sum\_{i=1}^n e = e, \\quad
> \\bar{\\mathbf{X}}\_m = \\frac{1}{m}\\sum\_{i=1}^m e = e
> $$
>
> We focus on how the aggregated node features affect generalization. Let the model be $f\_\theta$, with expected risk:
> $$
> R(\\theta; \\bar{\\mathbf{X}}) = \\mathbb{E}[\\ell(f\_\theta(\\bar{\\mathbf{X}}))]
> $$
>
> The generalization gap between small and large scales is:
> $$
> \\|\\;R(\\theta; \\bar{\\mathbf{X}}\_m) - R(\\theta; \\bar{\\mathbf{X}}\_n)\\;\\|
> $$
>
> ---
>
> ### **(2) Lipschitz Assumption**
>
> Assume the model is Lipschitz continuous w.r.t. the aggregated input embeddings:
> $$
> \\|\\;f\_\theta(\\bar{\\mathbf{X}}\_n) - f\_\theta(\\bar{\\mathbf{X}}\_m)\\;\\| \le L \cdot \\|\\bar{\\mathbf{X}}\_n - \\bar{\\mathbf{X}}\_m\\|
> $$
>
> where $L$ is the Lipschitz constant.
>
> ---
>
> ### **(3) Analysis Based on Aggregated Embedding Norm Differences**
>
> **Coordinate Linear Embedding:**
>
> The embedding is linear:
> $$
> X\_i = W \\mathbf{x}\_i + b
> $$
>
> By linearity, for any two sets of node coordinates:
> $$
> \\|\\bar{\\mathbf{X}}\_m - \\bar{\\mathbf{X}}\_n\\| \le \\|W\\| \cdot \\Big\\| \frac{1}{m}\sum\_{i=1}^m \\mathbf{x}\_i - \frac{1}{n}\sum\_{i=1}^n \\mathbf{x}\_i \\Big\\|
> $$
>
> Combining with the Lipschitz assumption:
> $$
> \\|f\_\theta(\\bar{\\mathbf{X}}\_m) - f\_\theta(\\bar{\\mathbf{X}}\_n)\\| \le L \cdot \\|\\bar{\\mathbf{X}}\_m - \\bar{\\mathbf{X}}\_n\\| \le L \cdot \\|W\\| \cdot \\Big\\| \frac{1}{m}\sum\_{i=1}^m \\mathbf{x}\_i - \frac{1}{n}\sum\_{i=1}^n \\mathbf{x}\_i \\Big\\|
> $$
>
> As the number of nodes increases from $n$ to $m$, the coordinate distribution difference grows, and the linear mapping $W$ amplifies this difference.
>
> ---
>
> **Global Embedding:**
>
> All node embeddings are identical:
> $$
> X\_i = e
> $$
>
> Therefore, the difference is zero:
> $$
> \\|\\bar{\\mathbf{X}}\_m - \\bar{\\mathbf{X}}\_n\\| = 0
> $$
>
> By Lipschitz continuity:
> $$
> \\|f\_\theta(\\bar{\\mathbf{X}}\_m) - f\_\theta(\\bar{\\mathbf{X}}\_n)\\| \le L \cdot 0 = 0
> $$
>
> The input embedding distribution does not change with scale, so the model output is stable across scales.
>
> ---
>
> ### **(4) Conclusion**
>
> $$
> \begin{aligned}
> \text{Coordinate Embedding: } & \\|\\;R(\\theta;\\bar{\\mathbf{X}}\_m) - R(\\theta;\\bar{\\mathbf{X}}\_n)\\;\\| \le L \cdot \\|W\\| \cdot \\Big\\| \frac{1}{m}\sum\_{i=1}^m \\mathbf{x}\_i - \frac{1}{n}\sum\_{i=1}^n \\mathbf{x}\_i \\Big\\| \\\\
> \text{Global Embedding: } & \\|\\;R(\\theta;\\bar{\\mathbf{X}}\_m) - R(\\theta;\\bar{\\mathbf{X}}\_n)\\;\\| = 0
> \end{aligned}
> $$
>
> Coordinate embeddings amplify differences in input distributions as the scale increases.
>
> Global embeddings eliminate input variation, keeping the model output consistent and improving generalization from small to large scale.
>
> ### **Reference**
> [1] Virmaux, Aladin, and Kevin Scaman. "Lipschitz regularity of deep neural networks: analysis and efficient estimation." Advances in Neural Information Processing Systems 31 (2018).

---

> ### Author Response · Authors · 2025-11-14
>
> ## **Q2. Discussion on Edge Normalization**
>
> In Appendix G, we provide a detailed discussion of the effects of 2D-Uniform normalization and Min-Max Scaling. Beyond the paper, we also experimented with Z-Score normalization and Mean Normalization. The results show that these alternatives perform similarly to, or worse than, Min-Max Scaling. After graph sparsification, extreme edge lengths are substantially reduced and the relative distance distribution within subgraphs becomes more stable, which explains why Min-Max Scaling works well in most cases.
>
> ------
>
>
>
> ## **Q3. Clarification of Training Details**
>
> In fact, we have provided detailed and comprehensive descriptions of the training setup in both the main paper and the appendix.
>
> > Whether the 1M training instances are seen in each epoch?
>
> Yes.
>
> > Full training curves (loss, validation metrics) and total training time.
>
> We use MR@5 as a surrogate for the training loss. In practice, MR@5 provides a more faithful reflection of the optimization progress for discrete problems, serving as a discrete form of both the training loss and the validation metric. Moreover, our MR@5 curves clearly indicate that the model has been sufficiently trained.
>
> Many MR@5 curves are shown in Figures 3–8, and we will include the full training curves in the updated version of the paper. The detailed training time is reported in Tables 8 and 10 of the appendix; please refer to them for more details.
>
> ------
>
>
>
> ## **Q4. Case Study on TSPLIB Instances**
>
> In Figure 22 of the appendix, we present three STSP instances whose node distributions are highly irregular. Their spatial layouts deviate substantially from the uniform distribution that UNE-GCN was trained on, causing the normalized distance matrices to fall outside the model's prior experience. Despite this mismatch, UNE-GCN still achieves significantly better performance than NeuroLKH on TSPLIB instances in overall.
>
> ------
>
>
>
> ## **Q5. Clarification of JV Transformation**
>
> In fact, our implementation of the JV Transformation **does not** introduce any additional parameters (addressing Weaknesses 4). Using the JV Transformation within the ATSP + LKH-3 framework is fair, as LKH-3 itself applies this transformation when solving ATSP. **We have verified that the outputs of our implementation match those produced by LKH-3’s built-in JV Transformation exactly.**
>
> ------
>
> ## **Remarks**
> In summary, we have provided extensive experimental results and detailed setup descriptions in the paper, and we have taken measures to ensure the fairness of our evaluation. Once again, we sincerely thank you for your valuable review comments. We hope that our responses have addressed your concerns and will lead to a positive improvement in the final rating.

---

### Meta-Review · Area_Chair_63yZ · 2026-01-03

**Summary:**

This work investigates and rethinks the efficiency of graph convolutional networks (GCNs) for solving the traveling salesman problem (TSP), with particular focus on challenge of generalization, overfitting, and extension to asymmetric TSP (ATSP). It proposes a new GCN model called UNE-GCN, which integrates three encoding strategies, namely Input Unification, Edge Normalization, and Aggregation Enhancement, for effective graph encoding. Experimental results show that UNE-GCN, when combined with LKH-3, can outperform the original LKH-3 on different TSP and ATSP instances with up to 10K nodes.

The reviewers have mixed scores (6,6,4,4,2) on this work and raised many concerns regarding novelty, contribution, the perspective of a rethinking paper, theoretical justification, computational efficiency, generalization to real-world distributions, method details, experimental settings, comparisons (especially those without LKH-3), and clarifications. The authors have provided a detailed rebuttal along with new results to address these concerns.

However, none of the reviewers decided to increase their score. Reviewer eBvv and Reviewer 2Lz4 keep their borderline acceptance scores (6, 6) without further response. In their responses, Reviewer SU2T, Reviewer 2vXk, and Reviewer N9Zi indicated their concerns have not yet been properly addressed and therefore kept their negative scores (4, 4, 2). The authors have provided follow-up rebuttals for Reviewer 2vXk/Reviewer SU2T before/after the OpenReview incident, but did not receive further response.

I have read this paper in detail and find this work interesting and meaningful. I also agree with the reviewers that this work is clearly motivated, the proposed modules are simple and implementable, and the experimental results are promising. However, I believe some major concerns raised by the reviewers (novelty, contributions, and applicability) have not been adequately addressed. Therefore, the Reviewer SU2T/2vXk/N9Zi might keep their negative scores even if they had been able to participate fully in the discussion. In addition, no reviewer voted to strongly support the acceptance of this work. The possible final score of this work could be (6, 6, 6, 4, 2) or (6, 6, 4, 4, 2). Therefore, I have to recommend rejecting this work and strongly encourage the authors to submit a carefully revised paper to the next venue.

**Reviewer Concerns:**

Based on the reviewers' responses, I believe that many concerns have been appropriately addressed through the rebuttal, including those related to theoretical justification, computational efficiency, generalization to real-world distributions, methodological details, experimental settings, and clarifications. However, several major concerns still remain.

**Technical Novelty (Reviewer 2vXk, Reviewer N9Zi):** Reviewers think that the proposed encoding components, namely Input Unification (U), Edge Normalization (N), and Aggregation Enhancement (E), appear to be relatively mild extensions or consolidations of existing ideas. Therefore, their novelty is still somewhat limited.

**Contribution and Applicability to Other Representative Methods (Reviewer SU2T, Reviewer 2vXk, Reviewer N9Zi):** The reviewers find it remains unclear how substantial the performance gains brought by the integrated UNE-GCN would be relative to the original contributions of prior methods. They believe that while the orthogonal, plug-in style application of UNE would very likely bring consistent improvements, these gains may be incremental when compared against the main methodological novelties of those original works. The reviewers believe demonstrating the applicability of UNE-GCN to other representative methods is necessary. An ideal way to assess the true incremental value of UNE-GCN would be to replace the original GCN backbone in representative prior works (e.g., Att-GCN, NeuroLKH, DIFUSCO, Fast-T2T) with the proposed UNE-GCN, while keeping all other components like training protocols (e.g., the discrete diffusion mechanism), data scale, etc., strictly unchanged.

**(At Least Partially Addressed) Performance without LKH and Generalization to Other Combinatorial Optimization Problems (Reviewer SU2T, Reviewer 2vXk, Reviewer N9Zi):** The reviewers argue that this work should demonstrate that the generalization improvement is independent of post-inference improving techniques (e.g., LKH-3, MCTS, etc). They also ask for comparisons on other combinatorial optimization problems (e.g., CVRP).

With the authors' follow-up response, I believe some of these concerns have been (at least partially) addressed, such as the performance without LKH-3. However, a major revision is still needed to thoroughly tackle the remaining concerns. In particular, I recommend:

- Contextualizing and comparing this work's contributions with other recent rethinking studies in the field, such as [1] and [2].

- Adding carefully designed experiments to analyze the impact and applicability of UNE when integrated into other representative methods with proper ablation studies.

- Carefully rewriting the current manuscript to shift focus away from LKH-3 and toward the basic contributions of UNE-GCN. On the other hand, the combination with 2-opt/MCTS/LKH-3 could be a plus when the contribution of UNE-GCN itself has been fully supported.

Addressing these points would substantially strengthen the submission and make it a strong candidate to the next venue.

[1] Position: Rethinking Post-Hoc Search-Based Neural Approaches for Solving Large-Scale Traveling Salesman Problems. ICML 2024.

[2] Rethinking the "Heatmap + Monte Carlo Tree Search" Paradigm for Solving Large Scale TSP. arXiv:2411.09238.

**Reviewer Scores:**

Based on the rebuttal and reviewers' responses, if the reviewers had been able to participate fully in the discussion, I believe Reviewer eBvv (6), Reviewer 2Lz4 (6), Reviewer 2vXk (4) and Reviewer N9Zi (2) would likely maintain their initial scores. Reviewer SU2T may or may not raise their score, as some of their concerns have been addressed while others remain.

Therefore, the final score of this work could be (6, 6, 6, 4, 2) or (6, 6, 4, 4, 2).

---

### Decision · Program_Chairs · 2026-01-26

Reject